# TSI-BENCH: BENCHMARKING TIME SERIES IMPUTATION

## ABSTRACT

Effective imputation is a crucial preprocessing step for time series analysis. Despite the development of numerous deep learning algorithms for time series imputation, the community lacks standardized and comprehensive benchmark platforms to effectively evaluate imputation performance across different settings. Moreover, although many deep learning forecasting algorithms have demonstrated excellent performance, whether their modelling achievements can be transferred to time series imputation tasks remains unexplored. To bridge these gaps, we develop TSI-Bench, the first (to our knowledge) comprehensive benchmark suite for time series imputation utilizing deep learning techniques. The TSI-Bench pipeline standardizes experimental settings to enable fair evaluation of imputation algorithms and identification of meaningful insights into the influence of domain-appropriate missing rates and patterns on model performance. Furthermore, TSI-Bench innovatively provides a systematic paradigm to tailor time series forecasting algorithms for imputation purposes. Our extensive study across 34,804 experiments, 28 algorithms, and 8 datasets with diverse missingness scenarios demonstrates TSI-Bench's effectiveness in diverse downstream tasks and potential to unlock future directions in time series imputation research and analysis. All source code and experiment logs are released.

## 1 INTRODUCTION

Time series data is widely used in many application domains (Zhang et al., 2024; Jin et al., 2024; Hamilton, 2020; Liu et al., 2023a), such as air quality monitoring (Liang et al., 2023), energy systems (Zhu et al., 2023), traffic (Zeng et al., 2022), and healthcare (Ibrahim et al., 2021). However, due to sensor malfunctions, environmental interference, privacy considerations, and other factors, the time series data available for analysis may be incomplete. The absence of complete data can greatly undermine a model's ability to accurately capture the domain for downstream tasks. By imputing missing values in time series data, the underlying mechanisms leading to missingness can be restored, thereby improving the data completeness and reliability for subsequent analysis and model construction.

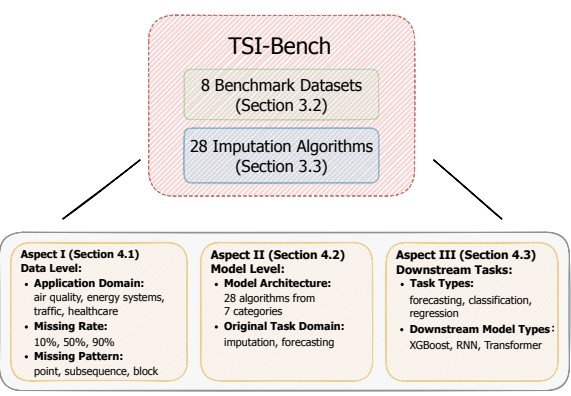

Figure 1: An overview of TSI-Bench.

The current research landscape in time series imputation is marked by inconsistencies, knowledge gaps, and wide performance disparities reported in literature reviews (Fang & Wang, 2020; Wang et al., 2024). These variations arise from differences in data characteristics and dimensionalities, domain-specific missing patterns, and task-specific requirements. They are further amplified by the heterogeneity of implementation,

Table 1: A comparison of TSI-Bench and other well-known time series imputation toolkits GluonTS (Alexandrov et al., 2020), Sktime (Löning et al., 2019), imputeTS (Moritz & Bartz-Beielstein), mice (Van Buuren & Groothuis-Oudshoorn, 2011), Impyute (Law, 2017), and ImputeBench (Khayati et al., 2020).

| | TSI-Bench (ours) | GluonTS | Sktime | imputeTS | mice | Impyute | ImputeBench |
|---|---|---|---|---|---|---|---|
| Naive Approaches? | ✓ | ✓ | ✓ | ✓ | ✓ | ✓ | ✗ |
| Machine Learning Approaches? | ✓ | ✗ | ✓ | ✗ | ✓ | ✗ | ✓ |
| Hyperparameter Optimization? | ✓ | ✗ | ✗ | ✗ | ✗ | ✗ | ✓ |
| Missing Pattern Simulations? | ✓ | ✗ | ✗ | ✗ | ✗ | ✗ | ✓ |
| Programming Language | Python | Python | Python | R | R | Python | C++ |
| Number of Integrated Datasets | **172** | 0 | 19 | 6 | 20 | 1 | 10 |
| Number of Integrated Algorithms | **39** | 6 | 10 | 10 | 1 | 2 | 19 |

experiment designs, and evaluation settings. This complicates model and study comparisons, significantly influencing algorithm performance, but remains deeply under-explored (Qian et al., 2024).

To bridge existing gaps, we designed the first (to the best of our knowledge) comprehensive time series imputation benchmark, named TSI-Bench. TSI-Bench is built on top of an *in-house developed imputation suite*, which includes 28 high-performing time series imputation and forecasting algorithms. It enables efficient and fair evaluation of newly proposed methods against existing baselines through a standardized *ecosystem*. This ecosystem provides a comprehensive set of data preprocessing tools, missingness simulation techniques, and performance evaluation methods. Researchers can use these resources to tailor the imputation task to suit the specific needs of their datasets, application domains, and downstream tasks. Notably, because forecasting is intertwined with imputation, TSI-Bench also incorporates a systematic paradigm to leverage time series forecasting models for imputation purposes. A comparison of TSI-Bench with prior works is presented in Table 1.

The primary contributions of TSI-Bench are as follows.

- **The first comprehensive benchmark for evaluating time series imputation**, which includes **28** ready-to-use algorithms spanning both imputation-based and forecasting-based approaches. The benchmark utilizes eight datasets across four diverse domains (air quality, traffic, electricity, and healthcare) with various dimensionalities, missing patterns, and data types.

- **TSI-Bench offers both research- and application-driven benchmarking perspectives**, enabling a standardized analysis of four critical dimensions of the imputation process, identified through the study of various domain requirements: (1) the capacity to simulate varying degrees of missingness and assess its impact on model performance; (2) the flexibility to incorporate different missing patterns (e.g., subsequence or block missingness) in time series, tailored to domain-specific needs, in addition to single-point missingness; (3) the selection of imputation or forecasting models; and (4) the alignment with expected downstream tasks. Researchers dealing with missing data can leverage TSI-Bench to evaluate the suitability of different imputation approaches for their specific problem, dataset, and downstream applications.

- **Identifying and addressing limitations inherent in existing TSI evaluation schemes to ensure rigorous and equitable comparisons**. We build an open-source ecosystem encompassing a standardized data preprocessing pipeline, flexible and user-driven missingness simulation, metric utilization, and hyperparameter tuning. This ecosystem not only facilitates the effortless reproduction of our results, but also provides users with the tools to assess and integrate their datasets and models into our ecosystem with ease.

Our collective experimental efforts with TSI-Bench assess the efficacy of the different algorithms in a total of **34,804 experiments**. Our extensive experiments reveal the following **key findings**: (1) Different missing patterns and rates significantly influence the performance of imputation methods, and none of the 28 algorithms significantly outperforms the others across all settings, emphasizing the importance of the model

tuning and data processing capabilities enabled by TSI-Bench; (2) Forecasting architectures demonstrate effectiveness when used as an imputation backbone; (3) Imputation enhances both flexibility and effectiveness across downstream tasks by enabling the broad application of high-quality imputed time series.

## 2 PRELIMINARIES AND RELATED WORK

**Notation**   A time series collection of length $T$ can be represented by a matrix $\mathbf{X} = (\boldsymbol{x}_1, \ldots, \boldsymbol{x}_t, \ldots, \boldsymbol{x}_T)^\top \in \mathbf{R}^{T \times D}$, with $D$ being the data dimension. An observation $\boldsymbol{x}_t \in \mathbb{R}^{1 \times D}$ is acquired at time step $t$. To encode the presence of missing values in $\mathbf{X}$, we introduce a mask matrix $\mathbf{M} \in \mathbb{R}^{T \times D}$, i.e., $M_{t,d} = 1$ if $X_{t,d}$ exists, otherwise $M_{t,d} = 0$. We denote the ground truth by $\tilde{\mathbf{X}} \in R^{T \times D}$. Given a reconstructed time series $\bar{\mathbf{X}}$ by an imputer, the objective of time series imputation is to obtain the imputed time series:

$$\hat{\mathbf{X}} = \mathbf{M} \odot \mathbf{X} + (1 - \mathbf{M}) \odot \bar{\mathbf{X}}, \tag{1}$$

so as to minimize the discrepancy between the imputed $\hat{\mathbf{X}}$ and the ground truth $\tilde{\mathbf{X}}$ as possible. $\odot$ denotes the element-wise multiplication.

**Related Work - Imputation Methods**   Deep imputation methods have recently gained popularity in handling missing values in time series data due to their ability to model the nonlinearities and temporal patterns inherent in time series. The literature contains a wide range of deep imputation methods, which can be broadly classified as either predictive or generative. Predictive imputation methods aim to consistently predict deterministic values for missing components within the time series (Yoon et al., 2019; Wu et al., 2023; Cini et al., 2022; Marisca et al., 2022; Ma et al., 2019; Bansal et al., 2021). Existing models employ a variety of neural network architectures, including Recurrent Neural Networks (RNNs), Convolutional Neural Networks (CNNs), Graph Neural Network (GNN), and attention mechanisms. Highly cited examples include GRU-D (Che et al., 2018a) and BRITS (Cao et al., 2018), which achieve improved imputation performance by integrating a time-decay mechanism into RNNs to capture irregularities caused by missing values. TimesNet (Wu et al., 2023) is a highly-performing CNN model that incorporates a Fast Fourier Transformation to restructure 1D time series into a 2D format. Finally, SAITS (Du et al., 2023) is an attention-based model comprising two diagonal-masked self-attention blocks and a weighted-combination block, which leverages attention weights and missingness indicators to enhance imputation precision.

Generative deep imputation models, built upon generative models such as VAEs, GANs, and diffusion models, are distinguished by their ability to generate varied outputs for missing observations, reflecting the inherent uncertainty of the imputation process (Mulyadi et al., 2021; Kim et al., 2023; Luo et al., 2018; 2019; Alcaraz & Strodthoff, 2023; Chen et al., 2023b; Yang et al., 2024). Examples include GP-VAE (Fortuin et al., 2020), which exploits a Gaussian process as a prior distribution to model temporal dependencies in the incomplete time series; US-GAN (Miao et al., 2021b) increases the complexity of discriminator training by compromising the masking matrix, subsequently leading to the improvement of generator performance through adversarial dynamics; and finally CSDI (Tashiro et al., 2021) adapts diffusion models for TSI through a conditioned training strategy by employing a subset of the observed data as conditional inputs to guide the generation process of the missing segments within the time series.

**Related Work - Forecasting Frameworks for Imputation**   In recent years, numerous deep learning architectures have exhibited exceptional performance in time series forecasting problems. Such as Transformer-based structures (Wu et al., 2021; Zhou et al., 2021a; Nie et al., 2023), GNN-based structures (Cao et al., 2020; Wu et al., 2020; Liu et al., 2022b; Jin et al., 2023), MLP-based structures (Ekambaram et al., 2023; Chen et al., 2023a; Wang et al., 2023a), and diffusion-based structures (Rasul et al., 2021; Tashiro et al., 2021; Shen & Kwok, 2023; Shen et al., 2024).

Figure 2: An overview of the designed TSI-Bench suite.

Transformers excel at time series forecasting due to their ability to model long-range dependencies and capture complex temporal patterns (Wen et al., 2023). Examples include Informer (Zhou et al., 2021b) and Crossformer (Zhang & Yan, 2022), which utilize different attention mechanisms to weigh the importance of different time steps dynamically. This is particularly beneficial for time series data, which often contains long-term dependencies and intricate patterns. At the same time, researchers showed that simple models can sometimes outperform Transformer architectures in long-term time series forecasting tasks (Zeng et al., 2023). MLP structures, known for their lower complexity and computational efficiency, have attracted growing interest from researchers. For instance, TSMixer (Chen et al., 2023a) extends the MLP mixer concept from computer vision (Tolstikhin et al., 2021) to time series forecasting by performing mixing operations along both time and feature dimensions (Ekambaram et al., 2023). Other innovative MLP-based designs, such as Koopa (Liu et al., 2023c) and FITS (Xu et al., 2023), further highlight the promising potential of MLP approaches in advancing time series forecasting.

The progress in forecasting models has the potential to be beneficial in imputation problems. Essentially, forecasting and imputation problems can be regarded as two similar tasks in time series analysis, as both require obtaining temporal representations of existing time series data to predict future values (Wen et al., 2019) or to impute missing values (Wang et al., 2024; 2023b). Given that time series forecasting is a highly active research area within time series analysis, a framework that enables researchers to directly apply forecasting methods to imputation challenges could potentially lead to the development of more efficient imputation techniques. This, in turn, might further advance research in time series analysis.

## 3 THE SETUP OF TSI-BENCH

In this section, we introduce the overview of the models evaluated in the experiments, along with the dataset descriptions and pertinent implementation details.

### 3.1 TSI-BENCH SUITE

TSI-Bench is a standardized suite that streamlines the imputation process and downstream tasks, built upon the `PyPOTS` ecosystem (Du, 2023). `PyPOTS` provides easy, standardized access to a variety of algorithms for imputation, classification, clustering, and forecasting. It supports the entire imputation workflow, from data loading and processing to model building and data imputation. The TSI-Bench workflow, depicted in Figure 2, begins with data being loaded from the time series database, `TSDB`, which serves as the data warehouse. The loaded data is then transformed into partially observed time series (POTS) using `PyGrinder`, a flexible,

user-driven missingness simulation tool. The POTS is subsequently processed by `BenchPOTS`, which provides standardized data preprocessing pipelines to benchmark machine learning algorithms on POTS. Following preprocessing, the time series is passed through an imputation model implemented in `PyPOTS` to estimate the missing values, resulting in the imputed time series. Finally, the imputed data is utilised by downstream models for further analysis, such as classification, regression, and forecasting.

## 3.2 DATASETS

In TSI-Bench, we conduct experiments using **eight** diverse and representative datasets from four domains: **air quality**, **traffic**, **electricity**, and **healthcare**. In the **air quality** domain, the **BeijingAir** dataset contains hourly data on six pollutants and meteorological variables from 12 sites in Beijing (Zhang et al., 2017). The **ItalyAir** dataset includes 9,358 hourly records of chemical sensor responses and pollutant concentrations collected from March 2004 to February 2005 (De Vito et al., 2008). In the **traffic** domain, the **PeMS** dataset provides hourly road occupancy rates from the San Francisco Bay Area freeways, while the **Pedestrian** dataset consists of automated pedestrian counts in Melbourne for the year 2017 (Tang et al., 2020). In the **electricity** domain, the **Electricity** dataset records hourly electricity consumption for 370 clients, and the **ETT** dataset includes power load and oil temperature data, with the **ETT_h1** subset used for our experiments (Zhou et al., 2021a). For **healthcare**, the **PhysioNet2012** dataset comprises 12,000 Intensive Care Unit (ICU) patient records, focusing on in-hospital mortality prediction in the presence of 80% missing values (Silva et al., 2012). Additionally, the **PhysioNet2019** dataset includes clinical data from 40,336 ICU patients, using subset A for analysis (Reyna et al., 2020).

## 3.3 MODELS

To comprehensively evaluate the performance of various models in the time series imputation task, we utilise **28** different models integrated into the `PyPOTS` Ecosystem. These models are based on distinct architectures, originally designed for different tasks. Among the evaluated models are **Transformer architectures**, including Transformer (Yıldız et al., 2022), Pyraformer (Liu et al., 2021), Autoformer (Wu et al., 2021), Informer (Zhou et al., 2021b), Crossformer (Zhang & Yan, 2022), PatchTST (Nie et al., 2023), ETSformer (Woo et al., 2022), Nonstationary Transformer (Liu et al., 2022c), SAITS (Du et al., 2023), and iTransformer (Liu et al., 2023b); **RNN architectures**, such as MRNN (Yoon et al., 2018), BRITS (Cao et al., 2018), and GRUD (Che et al., 2018b); **CNN architectures**, including MICN (Wang et al., 2022), SCINet (Liu et al., 2022a), and TimesNet (Wu et al., 2023); **GNN architectures**, such as StemGNN (Cao et al., 2020); **MLP architectures**, such as FiLM (Zhou et al., 2022), DLinear (Zeng et al., 2023), Koopa (Liu et al., 2024), and FreTS (Yi et al., 2024); and **Generative architectures**, including GP-VAE (Fortuin et al., 2020), US-GAN (Miao et al., 2021a), and CSDI (Tashiro et al., 2021), which are based on VAE, GAN, and diffusion models. Additionally, we apply several **traditional methods**, such as Mean, Median, LOCF (Last Observation Carried Forward), and Linear interpolation. For the original forecasting models (e.g., Transformer, Pyraformer, Autoformer), we adapt the embedding strategies and training methodologies from SAITS (Du et al., 2023) to reimplement them as imputation methods. Further details on the benchmarked models and the systematic approach used to adapt time series forecasting models for imputation purposes can be found in Appendix B and Appendix C.5.

## 3.4 IMPLEMENTATION DETAILS

**Dataset Preprocessing:** Except PhysioNet2012, PhysioNet2019, and Pedestrian, which already contain separated time series samples, all other datasets are split into training, validation, and test sets according to the time period. A sliding window function is then applied to generate individual data samples. Standardization is performed on all datasets. For further details on dataset preprocessing, please refer to Appendix A.2. **Missing Patterns:** To more effectively simulate missing scenarios in time series, we deviate from the traditional

Table 2: Imputation performance with 10% point missingness, shown as MAE (standard deviation). For each dataset, the best performance is indicated by the bold black text, the second-best performance by normal black text, and the third-best performance by bold dark grey.

| | BeijingAir | ItalyAir | PeMS | Pedestrian | ETT_h1 | Electricity | PhysioNet2012 | PhysioNet2019 |
|---|---|---|---|---|---|---|---|---|
| iTransformer | **0.123 (0.005)** | 0.223 (0.014) | **0.226 (0.001)** | 0.148 (0.005) | 0.263 (0.004) | 0.571 (0.178) | 0.379 (0.002) | 0.462 (0.006) |
| SAITS | 0.155 (0.004) | 0.185 (0.010) | 0.287 (0.001) | 0.131 (0.006) | **0.144 (0.006)** | **0.213 (0.014)** | 0.257 (0.019) | 0.352 (0.005) |
| Nonstationary | 0.209 (0.002) | 0.266 (0.007) | 0.331 (0.017) | 0.453 (0.024) | 0.359 (0.013) | **0.213 (0.014)** | 0.410 (0.002) | 0.458 (0.001) |
| ETSformer | 0.187 (0.002) | 0.259 (0.004) | 0.347 (0.006) | 0.207 (0.011) | 0.227 (0.007) | 0.412 (0.005) | 0.373 (0.003) | 0.451 (0.005) |
| PatchTST | 0.198 (0.011) | 0.274 (0.026) | 0.330 (0.013) | 0.126 (0.003) | 0.240 (0.013) | 0.550 (0.039) | 0.301 (0.011) | 0.420 (0.007) |
| Crossformer | 0.184 (0.004) | 0.246 (0.011) | 0.337 (0.007) | **0.119 (0.005)** | 0.232 (0.008) | 0.540 (0.034) | 0.525 (0.202) | 0.378 (0.007) |
| Informer | 0.148 (0.002) | 0.205 (0.008) | 0.302 (0.003) | 0.154 (0.010) | 0.167 (0.006) | 1.291 (0.031) | 0.297 (0.003) | 0.403 (0.002) |
| Autoformer | 0.257 (0.012) | 0.295 (0.008) | 0.598 (0.074) | 0.197 (0.008) | 0.267 (0.008) | 0.748 (0.027) | 0.417 (0.009) | 0.476 (0.002) |
| Pyraformer | 0.178 (0.004) | 0.217 (0.006) | 0.285 (0.003) | 0.153 (0.012) | 0.182 (0.008) | 1.096 (0.033) | 0.294 (0.002) | 0.387 (0.004) |
| Transformer | 0.142 (0.001) | **0.191 (0.010)** | 0.294 (0.002) | 0.136 (0.009) | 0.178 (0.015) | 1.316 (0.036) | **0.259 (0.006)** | **0.341 (0.002)** |
| BRITS | 0.127 (0.001) | 0.235 (0.007) | 0.271 (0.000) | 0.149 (0.005) | 0.145 (0.002) | 0.971 (0.016) | 0.297 (0.001) | **0.355 (0.001)** |
| MRNN | 0.568 (0.002) | 0.638 (0.003) | 0.624 (0.000) | 0.735 (0.006) | 0.789 (0.019) | 1.824 (0.005) | 0.708 (0.029) | 0.778 (0.015) |
| GRUD | 0.233 (0.002) | 0.368 (0.012) | 0.355 (0.002) | 0.204 (0.008) | 0.325 (0.004) | 0.976 (0.015) | 0.450 (0.004) | 0.471 (0.001) |
| TimesNet | 0.230 (0.010) | 0.280 (0.004) | 0.312 (0.001) | 0.157 (0.008) | 0.254 (0.008) | 1.011 (0.016) | 0.353 (0.003) | 0.394 (0.003) |
| MICN | 0.203 (0.001) | 0.283 (0.004) | 0.281 (0.003) | / | 0.267 (0.010) | 0.392 (0.006) | 0.378 (0.013) | 0.461 (0.007) |
| SCINet | 0.191 (0.011) | 0.288 (0.010) | 0.487 (0.101) | 0.149 (0.012) | 0.246 (0.019) | 0.581 (0.015) | 0.341 (0.005) | 0.427 (0.002) |
| StemGNN | 0.161 (0.002) | 0.260 (0.008) | 0.493 (0.079) | **0.127 (0.006)** | 0.248 (0.012) | 1.360 (0.078) | 0.331 (0.001) | 0.416 (0.002) |
| FreTS | 0.211 (0.008) | 0.273 (0.008) | 0.396 (0.027) | 0.138 (0.004) | 0.262 (0.029) | 0.718 (0.043) | 0.315 (0.008) | 0.406 (0.017) |
| Koopa | 0.363 (0.108) | 0.307 (0.041) | 0.532 (0.122) | 0.173 (0.020) | 0.435 (0.132) | 1.309 (0.531) | 0.413 (0.007) | 0.451 (0.019) |
| DLinear | 0.215 (0.016) | 0.242 (0.009) | 0.362 (0.009) | 0.179 (0.004) | 0.227 (0.006) | 0.519 (0.008) | 0.370 (0.000) | 0.432 (0.001) |
| FiLM | 0.318 (0.010) | 0.340 (0.011) | 0.784 (0.064) | 0.413 (0.010) | 0.583 (0.008) | 0.834 (0.043) | 0.458 (0.001) | 0.494 (0.003) |
| CSDI | **0.102 (0.010)** | 0.539 (0.418) | **0.238 (0.047)** | 0.231 (0.064) | 0.151 (0.064) | 1.483 (0.459) | **0.252 (0.002)** | 0.408 (0.019) |
| US-GAN | 0.137 (0.002) | 0.264 (0.012) | 0.296 (0.003) | 0.151 (0.016) | 0.458 (0.590) | 0.938 (0.009) | 0.310 (0.003) | 0.358 (0.002) |
| GP-VAE | 0.240 (0.006) | 0.369 (0.012) | 0.341 (0.007) | 0.319 (0.010) | 0.329 (0.017) | 1.152 (0.074) | 0.445 (0.006) | 0.562 (0.004) |
| Mean | 0.721 | 0.574 | 0.798 | 0.728 | 0.737 | 0.422 | 0.708 | 0.762 |
| Median | 0.681 | 0.518 | 0.778 | 0.667 | 0.71 | 0.408 | 0.69 | 0.747 |
| LOCF | 0.188 | 0.233 | 0.375 | 0.257 | 0.315 | 0.104 | 0.449 | 0.478 |
| Linear | 0.112 | **0.135** | **0.211** | 0.167 | 0.197 | **0.065** | 0.366 | 0.387 |

categories of missing mechanisms introduced by (Rubin, 1976). Instead, we implement point missing, subsequence missing, and block missing, which are particularly representative of common missing data situations in time series (Mitra et al., 2023; Cini et al., 2022; Khayati et al., 2020). These patterns capture both random and structured missingness, reflecting the dynamic nature of time series data and providing a robust framework for evaluating imputation strategies. For a detailed discussion and visual representation of these missing patterns, please refer to Appendix C.1. **Evaluation Metrics:** To assess imputation performance, we use mean absolute error (MAE), mean squared error (MSE), and mean relative error (MRE) (Cao et al., 2018). The mathematical formulations for these metrics can be found in Appendix C.2. Additionally, we collect inference time and the number of trainable parameters for further discussion. **Hyper-parameter Optimisation:** To ensure fair comparisons, hyper-parameter optimisation is applied to all deep-learning-based imputation algorithms. This process is implemented using PyPOTS (Du, 2023) and NNI (Microsoft), with details provided in Appendix C.3. **Downstream Task Design:** To evaluate the impact of imputation on downstream analysis, we perform various tasks using XGBoost, RNN, and Transformer models on the imputed datasets. Given that XGBoost has an inherent capability to manage incomplete data, we set XGBoost without imputation as the baseline for comparison. Further details on the downstream task design can be found in Appendix C.4.

## 4 EXPERIMENTAL ANALYSIS

We conduct a comprehensive analysis involving up to 34,804 experiments across 28 algorithms and 8 diverse domain datasets. The experiments are structured to examine three key perspectives: (1) **Data Perspective**: We evaluate different missing rates (10%, 50%, and 90%) and missing patterns (point, subsequence, block) across all datasets. (2) **Model Perspective**: We analyze each method based on its target task type (forecasting, imputation) and architecture (Transformer/attention, RNN, CNN, MLP, etc.), while also assessing the influence of model size and inference time on practical applications. (3) **Downstream Task Perspective**: We assess the quality of the imputed data using three architectures (XGBoost, RNN, Transformer) across three

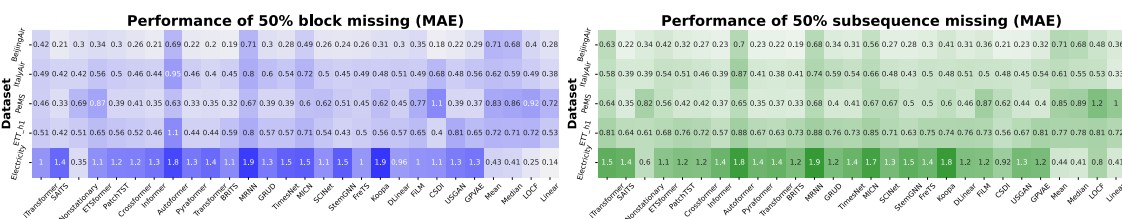

Figure 3: Performance of 50% block missing (left) and 50% subseq missing (right).

classical time series downstream tasks (classification, regression, forecasting) to highlight the importance and impact of imputation on downstream performance. Full results are available in Appendix D.

## 4.1 DATA PERSPECTIVE

We conduct experiments on the following aspects of data and missing patterns: datasets from different application domains, missing rates, and different missing patterns.

**Application Domains** Table 2 shows that different models show significant performance disparities across datasets from different application domains. For instance, Autoformer performs well on the Pedestrian dataset (with an MAE of 0.197) but performs poorly on the ItalyAir dataset (with an MAE of 0.295). This trend is also observed within the same application domain. BRITS excels on the BeijingAir dataset (with an MAE of 0.127) but shows worse performance on the ItalyAir dataset (with an MAE of 0.235). Overall, different models demonstrate varying performance across different application domains and datasets, with each model having its own strengths and weaknesses in specific datasets. Even in simpler application fields, such as electricity, we can get better results using traditional LOCF and Linear methods.

**Missing Rates** As the missingness data rate increases from 10% to 90%, there is a general trend of performance degradation across all models and datasets, as shown in Appendix D. For instance, Autoformer's MAE on the BeijingAir dataset rises from 0.257 at 10% to 0.898 at 50% missingness. Different models exhibit varying sensitivity to missing data, with BRITS maintaining relatively lower errors compared to others, whereas CSDI shows extreme sensitivity. CSDI's MAE on the ItalyAir dataset jumps from 0.539 at 10% missingness to 4.492 at 90%. CSDI is a conditional diffusion model that performs poorly and becomes less stable with high missing rates primarily due to noise accumulation and lack of contextual information. Computational time also varies, with CSDI showing substantially longer times at higher missing rates, while models like Autoformer and DLinear remain consistent. We want to highlight the importance of selecting appropriate models based on the expected missing data rate. For models with similar performance, it is better to consider computational efficiency for practical implementation.

**Missing Patterns** It is evident that different patterns significantly impact model performance, as shown in Figure 3. In general, block and subsequence missing patterns result in higher error metrics compared to point missingness, indicating that continuous missing data have a greater adverse effect on model performance. Specifically, point missingness has the least impact on most models, because missing data can be easily interpolated from adjacent points; for example, BRITS performs excellently under this pattern. In contrast, block missingness leads to more significant information loss over extended periods, making it challenging for models to infer missing data from distant points, as seen with the increased MAE and MSE for Autoformer and Crossformer. Subsequence missingness further degrades model performance since the missing subsequences may contain critical temporal patterns or trends that the models cannot easily reconstruct. Thus, different missing patterns affect model performance to varying degrees, with continuous missing patterns (block and subsequence missing) posing greater challenges.

We further provide visualization of imputed data from all imputation methods on ETT_h1 and Electricity in Appendix D.6. As shown in Figure 9 and Figure 10, different features within the same dataset exhibit

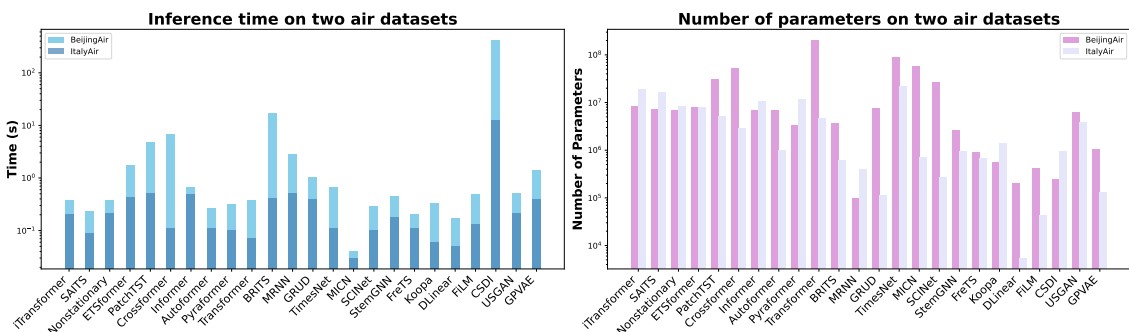

Figure 4: Inference time and the number of parameters on two air datasets with 10% point missing.

substantial variation. Some features like Feature #76 in Electricity exhibit limited temporal dependencies, whereas others like Feature #1 in ETT_h1 present clear temporal patterns, such as step changes. Moreover, when comparing the same feature across different missing patterns (e.g., point versus block missingness), the imputation results can differ significantly. While most algorithms perform well with point missingness, they often struggle with block missingness, which demands the preservation of longer-term dependencies. The results suggest that it is challenging for any single imputation algorithm to consistently perform well across different datasets and missingness scenarios effectively.

**Remark 1**: We suggest choosing imputation models tailored to data characteristics, such as application domains and missing patterns. Practitioners can leverage our comprehensive insights to make informed model choices to narrow their trials and experiments in selecting models, enhancing the development and open-source accessibility of time series imputation.

## 4.2 MODEL PERSPECTIVE

Given the space constraints, we will focus our discussion in this section on the model architectures and the primary task assuming a 10% missing rate.

**Architecture** The inference time, model size, and performance metrics of different architectures shown in Figure 4 and Table 2 indicate the following: Transformer/attention-based models, such as Autoformer and Informer, demonstrate excellent performance, but some models like Crossformer and SAITS have larger model sizes, potentially leading to lower inference efficiency. RNN-based models like BRITS show good performance (low MSE, MAE) but have longer inference times and moderate model sizes. CNN-based models like SCINet have faster inference times but relatively larger model sizes, with slightly lower performance compared to Transformer and RNN architectures. MLP-based models like DLinear have clear advantages in inference time and model size, with moderate performance, making them suitable for applications requiring quick inference and smaller models. CSDI, as a generative model, performs well on MAE, but has an extremely long inference time and moderate model size, suitable for scenarios demanding very high performance. Traditional methods, due to their limited scope, usually do not yield optimal results, especially in datasets with high missing rates and complexity. However, they can be more efficient and perform better on simpler datasets. In summary, different architectures have their own strengths and weaknesses in terms of inference time, model size, and performance metrics, necessitating the selection of an appropriate model based on specific application requirements.

**Forecasting Backbones for Imputation** With the adaptation paradigm in Appendix C.5, we transfer the backbone of the time series forecasting models to the imputation task and compare their performance against typical imputation methods. The results are categorized in Table 2, and detailed descriptions of the backbones are discussed in Appendix B. For forecasting backbones, DLinear, with the shortest inference time of 0.17

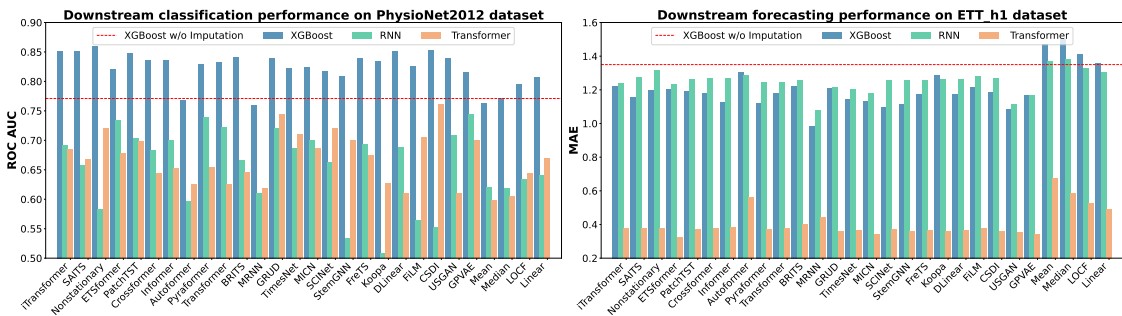

Figure 5: Comparison of downstream performance on the PhysioNet2012 dataset for classification with 10% point missing (left) and the ETT_h1 dataset for forecasting with 50% point missing (right).

seconds and a model size of 204K, shows efficient performance with an MAE of 0.215 on the Beijing dataset, making it ideal for real-time applications. In imputation, BRITS has an inference time of 17.04 seconds, a model size of 3.5M, and performs excellently on the Beijing dataset with an MAE of 0.127. CSDI, however, has a long inference time of 417.52 seconds, a model size of 244K, and higher MSE values, limiting its practical use despite advanced techniques. Generally, methods originally designed for imputation do not exhibit extraordinary performance compared to those designed for forecasting. In many scenarios, the adapted forecasting methods tend to perform better. A potential reason is that more research has been conducted on time series forecasting compared to time series imputation. Therefore, time series forecasting models are more updated and advanced in capturing temporal information and multivariate correlations. The experimental results demonstrate the success of our adaptation paradigm.

**Remark 2**: Existing forecasting backbones can be effectively applied to imputation tasks, enhancing the interaction between different time series tasks. This suggests that the imputation field can benefit from leveraging techniques and models originally designed for forecasting and other time series tasks. This cross-application improves the versatility and efficiency of time series models, opening new avenues for research and practical applications.

## 4.3 DOWNSTREAM TASKS PERSPECTIVE

**Forecasting Task** We evaluated the downstream forecasting performance of various models on the ETT_h1 dataset. As shown in Figure 5 (right), certain imputation methods significantly improve downstream forecasting performance. For instance, the Nonstationary imputation method consistently enhances accuracy across multiple models. Transformer-based models show notable gains with robust imputation strategies. XGBoost without imputation results in a higher MAE, but applying Transformer models with imputation consistently reduces the MAE. Overall, selecting the right imputation method is crucial for improving downstream forecasting task performance.

**Classification Task** In this test, we apply different models for imputation, followed by three types of classifiers. The goal is to assess the performance improvement brought by various imputation methods, as shown in Figure 5 (left). For instance, iTransformer achieves a ROC_AUC of 0.771 without imputation, which increases to 0.852 after imputation. The non-stationary method performs best with an ROC_AUC of 0.860, showcasing strong robustness in handling missing data. Overall, imputation significantly enhances classification performance, and the choice of imputation method is crucial for optimizing downstream tasks.

**Regression Task** As shown in Table 3, imputation methods significantly affect downstream regression performance under three missing patterns. For example, the RNN-based BRITS model achieves the lowest MAE of 0.826 in the block missing and subsequence scenarios, demonstrating its robustness in handling data

Table 3: Regression performance in ETT_h1 datasets with 50% point missing, 50% block missing and 50% subsequence missing. For each model, the best performance is indicated by the bold black text, the second-best performance by normal black text, and the third-best performance by bold dark grey.

| | ETT_h1 (point 50%) | | | ETT_h1 (subsequence 50%) | | | ETT_h1 (block 50%) | | |
|---|---|---|---|---|---|---|---|---|---|
| | XGB | RNN | Transformer | XGB | RNN | Transformer | XGB | RNN | Transformer |
| iTransformer | 1.224 | 1.377 | 1.406 | 1.170 | 1.434 | 1.470 | 1.208 | 1.422 | 1.399 |
| SAITS | 1.175 | 1.402 | 1.401 | 1.094 | 1.424 | 1.469 | 1.168 | 1.363 | 1.385 |
| Nonstationary | 1.227 | 1.446 | 1.384 | 1.284 | 1.448 | 1.469 | 1.189 | 1.438 | 1.368 |
| ETSformer | 1.222 | 1.329 | 1.351 | 1.187 | 1.407 | 1.433 | 1.004 | 1.285 | 1.327 |
| PatchTST | 1.234 | 1.398 | 1.370 | 1.253 | 1.476 | 1.502 | 1.183 | 1.362 | 1.340 |
| Crossformer | 1.228 | 1.391 | 1.380 | 1.145 | 1.449 | 1.474 | 1.149 | 1.335 | 1.285 |
| Informer | 1.214 | 1.391 | 1.398 | 1.196 | 1.423 | 1.442 | 1.158 | 1.333 | 1.351 |
| Autoformer | 1.348 | 1.450 | 1.442 | 1.364 | 1.471 | 1.538 | 1.327 | 1.267 | 1.363 |
| Pyraformer | 1.169 | 1.366 | 1.373 | 1.081 | 1.391 | 1.443 | 1.071 | 1.328 | 1.330 |
| Transformer | 1.170 | 1.378 | 1.357 | 1.074 | 1.391 | 1.350 | 1.128 | 1.296 | 1.277 |
| BRITS | 1.177 | 1.371 | 1.364 | 0.999 | 1.153 | 1.146 | 0.826 | 1.024 | 1.000 |
| MRNN | 1.070 | 1.250 | 1.379 | 1.111 | 1.362 | 1.483 | 0.870 | 1.274 | 1.387 |
| GRUD | 1.205 | 1.339 | 1.326 | 1.123 | 1.351 | 1.362 | 1.043 | 1.286 | 1.211 |
| TimesNet | 1.127 | 1.333 | 1.316 | 1.213 | 1.446 | 1.460 | 1.064 | 1.316 | 1.294 |
| MICN | 1.163 | 1.305 | 1.379 | 1.121 | 1.442 | 1.503 | 1.145 | 1.338 | 1.344 |
| SCINet | 1.174 | 1.411 | 1.375 | 1.181 | 1.450 | 1.446 | 1.182 | 1.361 | 1.323 |
| StemGNN | 1.139 | 1.392 | 1.410 | 1.211 | 1.391 | 1.439 | 1.136 | 1.373 | 1.323 |
| FreTS | 1.207 | 1.406 | 1.397 | 1.159 | 1.456 | 1.514 | 1.163 | 1.356 | 1.355 |
| Koopa | 1.227 | 1.384 | 1.419 | 1.233 | 1.446 | 1.509 | 1.274 | 1.347 | 1.302 |
| DLinear | 1.216 | 1.380 | 1.397 | 1.212 | 1.466 | 1.512 | 1.166 | 1.368 | 1.359 |
| FiLM | 1.247 | 1.379 | 1.442 | 1.422 | 1.389 | 1.455 | 1.301 | 1.368 | 1.368 |
| CSDI | 1.136 | 1.373 | 1.393 | 1.191 | 1.354 | 1.413 | 1.202 | 1.355 | 1.362 |
| USGAN | 1.155 | 1.299 | 1.283 | 1.004 | 1.244 | 1.219 | 0.951 | 1.174 | 1.098 |
| GPVAE | 1.178 | 1.336 | 1.340 | 1.138 | 1.310 | 1.354 | 0.996 | 1.177 | 1.184 |
| Mean | 1.431 | 1.512 | 1.627 | 1.430 | 1.859 | 1.779 | 1.413 | 1.669 | 1.750 |
| Median | 1.517 | 1.531 | 1.671 | 1.499 | 1.858 | 1.777 | 1.496 | 1.688 | 1.736 |
| LOCF | 1.446 | 1.477 | 1.571 | 1.446 | 1.699 | 1.651 | 1.433 | 1.601 | 1.611 |
| Linear | 1.393 | 1.453 | 1.525 | 1.423 | 1.610 | 1.588 | 1.395 | 1.558 | 1.556 |

loss. Transformer-based models also perform well, highlighting their ability to handle complex missing data patterns. In contrast, traditional methods such as Mean and Median imputation result in higher MAE values, underscoring their limitations. The improved performance of models like XGBoost applied to imputed data further emphasizes the importance of robust imputation for enhancing regression performance.

**Remark 3**: Imputation is not only critical on its own but also plays a key role in improving the performance of downstream tasks. Additionally, pre-imputation demonstrates high adaptability across different downstream tasks within the same datasets. Once high-quality imputed time series are generated, they can be effectively leveraged across a wide range of downstream applications.

## 5 CONCLUSION

In this paper, we introduce TSI-Bench, a comprehensive benchmark for time series imputation. Our evaluation encompasses 28 algorithms across 8 real-world datasets, illustrating that successful imputation hinges on factors like deep learning architectures, missing rates, missing patterns across space-time, and the datasets themselves. Additionally, incorporating uncertainty quantification in the evaluation process is essential, as it enables an assessment of confidence in the imputed data and helps to identify the limitations and reliability of the results. However, uncertainty quantification in time series imputation remains an emerging area, limited by the complexity of sequential dependencies, high-dimensional features, and the lack of standardized metrics.

Despite the complexities, TSI-Bench provides valuable insights and practical guidelines for time series imputation in real-world scenarios. We envision TSI-Bench as a continually evolving project. Our future plans include the integration of more advanced deep learning models, the deployment of more equitable evaluation metrics, and the expansion of our dataset collection. Additionally, we are committed to improving the user-friendliness of TSI-Bench, aiming to establish it as the standard tool for time series imputation.

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

CONTENTS

## A    DETAILED DESCRIPTION OF DATASETS AND PREPROCESSING

### A.1    DATASETS FOR EVALUATION

Detailed descriptions of the eight used datasets are listed below. An overview of the datasets' general information is presented in Table 4.

Table 4: General information of the eight datasets used in this work.

| Dataset | Air | | Traffic | | Electricity | | Healthcare | |
|---|---|---|---|---|---|---|---|---|
| | BeijingAir | ItalyAir | PeMS | Pedestrian | ETT_h1 | Electricity | PhysioNet2012 | PhysioNet2019 |
| # of Total Samples | 1458 | 774 | 727 | 3633 | 358 | 1457 | 3997 | 4927 |
| # of Variables | 132 | 13 | 862 | 1 | 7 | 370 | 35 | 33 |
| Sample Sequence Length | 24 | 12 | 24 | 24 | 48 | 96 | 48 | 48 |
| Time Interval | 1H | 1H | 1H | 1H | 1H | 15Min | 1H | 1H |
| Original Missing Rate (%) | 1.6 | 0 | 0 | 0 | 0 | 0 | 79.3 | 73.9 |
| Train Dataset Length | 851 | 466 | 455 | 955 | 212 | 851 | 2557 | 3152 |
| Validation Dataset Length | 304 | 154 | 122 | 239 | 75 | 304 | 640 | 789 |
| Test Dataset Length | 303 | 154 | 150 | 2439 | 71 | 302 | 800 | 986 |

### A.1.1    AIR QUALITY DATASETS

- **BeijingAir**[1] **(Zhang et al., 2017)**: BeijingAir dataset includes hourly air pollutants data from 12 nationally-controlled air-quality monitoring sites from March 2013 to February 2017. The dataset contains 11 continuous variables at multiple sites in Beijing. We aggregate these variables from all sites together so this dataset has 12*11=132 features.

- **ItalyAir**[2] **(De Vito et al., 2008)**: ItalyAir dataset contains 9358 instances of hourly averaged responses of 5 metal oxides from chemical sensors, along with hourly averaged concentrations of 7 pollutants from certified analyzers, from March 2004 to February 2005.

### A.1.2    TRAFFIC DATASETS

- **PeMS**[3]: The PeMS is a collection of hourly data from the California Department of Transportation, which describes the road occupancy rates measured by different sensors on San Francisco Bay area freeways.

- **Pedestrian**[4] (Tang et al., 2020): The City of Melbourne, Australia has developed an automated pedestrian counting system to better understand pedestrian activity within the municipality, The data of a specific region is from the whole year of 2017.

### A.1.3    ELECTRICITY DATASETS

- **Electricity**[5]: The Electricity Load Diagrams dataset contains hourly electricity consumption (kWh) for 370 clients over the period from January 2011 to December 2014.

- **ETT**[6] (Zhou et al., 2021a): The ETT dataset, collected from power transformers, includes preprocessed data on power load and oil temperature from July 2016 to July 2018. In the experiments, we use ETT_h1 included in ETT.

---

[1]https://archive.ics.uci.edu/dataset/501/beijing+multi+site+air+quality+data
[2]https://archive.ics.uci.edu/dataset/360/air+quality
[3]https://PeMS.dot.ca.gov
[4]https://www.timeseriesclassification.com/description.php?Dataset=MelbournePedestrian
[5]https://archive.ics.uci.edu/dataset/321/electricityloaddiagrams20112014
[6]https://github.com/zhouhaoyi/ETDataset

### A.1.4 HEALTHCARE DATASETS

- **PhysioNet2012**[7] (Silva et al., 2012): The PhysioNet/Computing in Cardiology Challenge 2012 dataset includes 12,000 ICU patient records from the MIMIC II Clinical database, version 2.6 (Saeed et al., 2011), focusing on patient-specific prediction of in-hospital mortality using data from the first 48 hours of ICU admission. Not all variables are available in all cases, hence about 80% values are missing in this dataset. The whole dataset has three subsets, and we only use the subset A in our experiments.

- **PhysioNet2019**[8] (Reyna et al., 2020): The PhysioNet Challenge 2019 dataset includes clinical data from ICU patients across three hospitals, with a total of 40,336 patient records and 40 clinical and physiological variables for each patient. Note that this dataset has two subsets, and we only use the subset A in our setting.

### A.2 DATASETS PREPROCESSING DETAILS

BeijingAir, ItalyAir, PeMS, Electricity, and ETT_h1 are all long-time series continuously collected from certain sources. Therefore, to generate them into the train, validation, and test sets and to avoid data leakage, we should first split them according to the time period. In BeijingAir, the first 28 months (2013/03 - 2015/06) of data are taken for training, the following 10 months (2015/07 - 2016/04) are for validation, and the left 10 months (2016/05 - 2017/02) are for test. In PeMS, the training set tasks the first 15 months (2016/07 - 2017/09) of data, and the validation set and test set take the following 4 months (2017/10 - 2018/01) and 6 months (2018/02 - 2018/07) respectively. Electricity uses the first 10 months (2011/01 - 2011/10) as the test set, the following 10 months (2011/11 - 2012/08) for validation, and the last 28 months (2012/09 - 2014/12) for training. The training, validation, and test sets of ETT_h1 separately take the first 14 months (2016/07 - 2017/08), the following 5 months (2017/09 - 2018/01), and the last 5 months (2018/02 - 2018/06). ItalyAir is split into 60%, 20%, and 20% for training, validation, and test. The sliding window function is applied to these five datasets to produce data samples. The window size of ItalyAir is 12, and 24 for BeijingAir and PeMS, 48 for ETT_h1, and 96 for Electricity. The sliding length is kept the same as the window size to guarantee there is no overlap between generated samples.

Dataset Pedestrian is offered with the split training set and test set, hence we separate 20% from the training set to form the validation set. Data samples in PhysioNet2012 share the same length, i.e. 48 steps. While samples in PhysioNet2019 have different lengths, hence we only keep samples with lengths larger than 48 and truncate the excess part to ensure samples all have 48 steps as well. For both PhysioNet2012 and PhysioNet2019, samples are firstly split into the training set and the test set according to 80% and 20%, then 20% of samples are taken from the training set as the validation set.

Note that standardization is applied in the preprocessing of all datasets.

## B DETAILED DESCRIPTION OF MODELS IN TSI-BENCH

### B.1 TRANSFORMER-BASED MODELS

- **iTransformer (Liu et al., 2023b)**: iTransformer repurposes the Transformer architecture by applying attention and feed-forward networks to inverted dimensions, embedding time points into variate tokens to better capture multivariate correlations and nonlinear representations, achieving state-of-the-art performance in time series forecasting.

---

[7]https://physionet.org/content/challenge-2012/1.0.0/
[8]https://physionet.org/content/challenge-2019/1.0.0/

Table 5: Summary of the 28 benchmarked algorithms, separated by **Architecture**.

| Model | Category | Architecture | Venue | Year |
|---|---|---|---|---|
| iTransformer (Liu et al., 2023b) | Forecasting | Transformer | ICLR | 2024 |
| SAITS (Du et al., 2023) | Imputation | Transformer | ESWA | 2023 |
| Nonstationary (Liu et al., 2022c) | Forecasting | Transformer | NeurIPS | 2022 |
| ETSformer (Woo et al., 2022) | Forecasting | Transformer | ArXiv | 2022 |
| PatchTST (Nie et al., 2023) | Forecasting | Transformer | ICLR | 2023 |
| Crossformer (Zhang & Yan, 2022) | Forecasting | Transformer | ICLR | 2022 |
| Informer (Zhou et al., 2021b) | Forecasting | Transformer | AAAI | 2021 |
| Autoformer (Wu et al., 2021) | Forecasting | Transformer | NeurIPS | 2021 |
| Pyraformer (Liu et al., 2021) | Forecasting | Transformer | ICLR | 2021 |
| Transformer (Yıldız et al., 2022) | Forecasting | Transformer | NeurIPS | 2017 |
| BRITS (Cao et al., 2018) | Imputation | RNN | NeurIPS | 2018 |
| MRNN (Yoon et al., 2018) | Imputation | RNN | TBME | 2018 |
| GRUD (Che et al., 2018b) | Imputation | RNN | Scientific reports | 2018 |
| TimesNet (Wu et al., 2023) | Generic | CNN | ICLR | 2023 |
| MICN (Wang et al., 2022) | Forecasting | CNN | ICLR | 2022 |
| SCINet (Liu et al., 2022a) | Forecasting | CNN | NeurIPS | 2022 |
| StemGNN (Cao et al., 2020) | Forecasting | GNN | NeurIPS | 2020 |
| FreTS (Yi et al., 2024) | Forecasting | MLP | NeurIPS | 2024 |
| Koopa (Liu et al., 2024) | Forecasting | MLP | NeurIPS | 2024 |
| DLinear (Zeng et al., 2023) | Forecasting | MLP | AAAI | 2023 |
| FiLM (Zhou et al., 2022) | Forecasting | MLP | NeurIPS | 2022 |
| CSDI (Tashiro et al., 2021) | Imputation | Diffusion | NeurIPS | 2021 |
| US-GAN (Miao et al., 2021a) | Imputation | GAN | AAAI | 2021 |
| GP-VAE (Fortuin et al., 2020) | Imputation | VAE | AISTATS | 2020 |
| Mean | Imputation | Traditional | - | - |
| Median | Imputation | Traditional | - | - |
| LOCF | Imputation | Traditional | - | - |
| Linear | Imputation | Traditional | - | - |

- **SAITS (Du et al., 2023)**: SAITS (Self-Attention-based Imputation for Time Series) is a self-attention imputation transformer with a weighted combination of two diagonally-masked self-attention (DMSA) blocks. It is designed to handle missing data in time series, ensuring robust and accurate data imputation.

- **Nonstationary (Liu et al., 2022c)**: Nonstationary (short for Nonstationary Transformer) addresses the challenge of non-stationarity in time series sequences by incorporating adaptive components that adjust to varying statistical properties over time.

- **ETSformer (Woo et al., 2022)**: ETSformer integrates exponential smoothing methods with Transformer models, aiming to provide accurate time series forecasting by combining statistical and deep learning approaches.

- **PatchTST (Nie et al., 2023)**: PatchTST uses a patching strategy combined with a Transformer architecture to enhance the time series forecasting task by capturing both local and global patterns effectively.

- **Crossformer (Zhang & Yan, 2022)**: Crossformer leverages cross-dimensional attention to model intricate dependencies within multivariate time series data, achieving good performance in complex forecasting application scenarios.

- **Informer (Zhou et al., 2021b)**: Informer enhances efficiency in long-time series forecasting by employing a self-attention distillation mechanism, which reduces redundant information while maintaining forecasting accuracy.

- **Autoformer (Wu et al., 2021)**: Autoformer introduces a novel decomposition architecture with an auto-correlation mechanism, effectively capturing both seasonal and trend patterns in time series forecasting tasks.

- **Pyraformer (Liu et al., 2021)**: Pyraformer is designed for long-term time series forecasting, utilizing a pyramid attention structure that efficiently captures temporal dependencies at multiple scales.

- **Transformer (Vaswani et al., 2017)**: Transformer introduces the self-attention mechanism, which enables the processing of sequential data by attending to different positions within the sequence simultaneously, leading to significant advancements in natural language processing and time series fields.

## B.2 RNN-BASED MODELS

- **BRITS (Cao et al., 2018)**: BRITS (Bidirectional Recurrent Imputation for Time Series) employs a bidirectional recurrent imputation strategy to handle missing values in time series, improving forecasting accuracy through iterative refinement.

- **MRNN (Yoon et al., 2018)**: The Multi-directional Recurrent Neural Network (MRNN) is designed for estimating missing values in spatiotemporal data and time series by leveraging temporal dependencies and multi-directional information sequence.

- **GRUD (Che et al., 2018b)**: GRUD (Gated Recurrent Unit for Decay) enhances the Gated Recurrent Unit (GRU) by incorporating decay mechanisms that model the impact of missing values over time, offering robust time series analysis.

## B.3 CNN-BASED MODELS

- **TimesNet (Wu et al., 2023)**: TimesNet is designed to efficiently model temporal patterns in time series data by incorporating multi-scale temporal convolutions and attention mechanisms. It can be used for short- and long-term forecasting, imputation, classification, and anomaly detection tasks.

- **MICN (Wang et al., 2022)**: MICN (Multi-scale Inception Convolutional Network) is a convolutional network architecture that captures multi-scale temporal features and combines local and global context for more accurate time series forecasting.

- **SCINet (Liu et al., 2022a)**: SCINet introduces a novel recursive downsample-convolve-interact architecture for time series forecasting, leveraging multiple convolutional filters to extract and aggregate valuable temporal features from downsampled sub-sequences, improving forecasting accuracy over existing models.

## B.4 GNN-BASED MODELS

- **StemGNN (Cao et al., 2020)**: StemGNN (Spectral Temporal Graph Neural Network) introduces an approach for time series forecasting by jointly capturing inter-series correlations and temporal dependencies in the spectral domain using Graph Fourier Transform (GFT) and Discrete Fourier Transform (DFT), eliminating the need for pre-defined priors.

## B.5 MLP-BASED MODELS

- **FreTS (Yi et al., 2024)**: FreTS uses MLPs in the frequency domain for time series forecasting, leveraging global view and energy compaction properties to enhance forecasting performance by transforming time-domain signals and learning frequency components.

- **Koopa (Liu et al., 2024)**: Koopa introduces a novel Koopman forecaster that disentangles time-variant and time-invariant components using Fourier Filter and employs Koopman operators for linear dynamics portrayal, achieving end-to-end forecasting with significant improvements in training time and memory efficiency.

- **DLinear (Zeng et al., 2023)**: DLinear introduces LTSF-Linear, a set of simple one-layer linear models, which outperform complex Transformer-based models in long-term time series forecasting by effectively preserving temporal information that Transformers inherently lose due to their permutation-invariant self-attention mechanism.

- **FiLM (Zhou et al., 2022)**: FiLM (Frequency improved Legendre Memory) introduces a novel approach by using Legendre Polynomials for historical information approximation, Fourier projection for noise reduction, and a low-rank approximation for computational efficiency, resulting in significant improvements in long-term forecasting accuracy.

## B.6 GENERATIVE MODELS

- **CSDI (Tashiro et al., 2021)**: CSDI (Conditional Score-based Diffusion Imputation) leverages conditional score-based diffusion models for accurate imputation and generation of missing values in time series applications.

- **US-GAN (Miao et al., 2021a)**: US-GAN integrates a classifier in a semi-supervised generative adversarial network to enhance its imputation of missing values in multivariate time series, leveraging both observed data and label information. Also, it introduces a temporal matrix to improve the discriminator's ability to differentiate between observed and imputed components.

- **GP-VAE (Fortuin et al., 2020)**: GP-VAE (Gaussian Process Variational Autoencoder) proposes a novel deep sequential latent variable model that combines VAE with a structured variational approximation to achieve non-linear dimensionality reduction and imputation, providing interpretable uncertainty estimates and improved imputation smoothness.

## B.7 TRADITIONAL METHODS

- **Mean**: The mean imputation method fills in missing values by calculating the mean (average) of the available values in the time series. This method is simple and quick to implement but may not be suitable for data with trends or seasonality, as it does not consider the temporal structure of the data.

- **Median**: Median imputation replaces missing values with the median value of the observed data points. The median is the middle value when the data points are ordered, making this method robust to outliers. It is particularly useful for skewed distributions or data with significant outliers.

- **LOCF**: LOCF (Last Observation Carried Forward) fills in missing values by carrying forward the last observed value. This method assumes that the value remains constant until the next observation is recorded. LOCF is straightforward and works well for short gaps in data, but it can introduce bias if the missing data spans a long period or if the data has a trend.

- **Linear**: Linear interpolation fills in missing values by connecting the last observed value before the missing data and the first observed value after the missing data with a straight line. This method assumes a linear trend between the two points and is useful for data with a relatively stable and linear pattern. However, it may not perform well with data that exhibit non-linear trends or seasonality.

## C  DETAILS IN TSI-BENCH

### C.1  MISSINGNESS

The concept of missingness can be understood through two fundamental aspects: missing mechanisms and missing patterns. Missing mechanisms describe the underlying processes that lead to missing values, while missing patterns define how these missing values manifest within the dataset.

**Missing Mechanisms**   The seminal taxonomy introduced by (Rubin, 1976) categorizes missing data mechanisms into three distinct types: Missing Completely At Random (MCAR), Missing At Random (MAR), and Missing Not At Random (MNAR). These classifications hinge on the conditional probability distribution $p(\mathbf{M}|\tilde{\mathbf{X}}, \phi)$, where $\phi$ parameterizes the stochastic mapping from the dataset $\tilde{\mathbf{X}}$ to the mask indicator $\mathbf{M}$. Specifically:

- **MCAR** implies that the probability of data being missing is independent of observations and missingness. This can be expressed as: $p(\mathbf{M}|\tilde{\mathbf{X}}, \phi) = p(\mathbf{M}|\phi)$,

- **MAR** suggests that the missingness depends solely on observations $p(\mathbf{M}|\tilde{\mathbf{X}}, \phi) = p(\mathbf{M}|\mathbf{X}, \phi)$,

- **MNAR** indicates that the missingness is related to the unobserved data and potentially the observed data as well:$p(\mathbf{M}|\tilde{\mathbf{X}}, \phi) = p(\mathbf{M}|\tilde{\mathbf{X}}, \phi)$.

**Missing Patterns**   Despite the clarity the above taxonomy provides, it often falls short in capturing the complexity of missing data scenarios in modern datasets (Mitra et al., 2023). This limitation has driven the development of new frameworks (Mitra et al., 2023; Cini et al., 2022; Khayati et al., 2020) that better align with diverse real-world data.

To effectively analyze missingness in time series data, we employ three distinct patterns of missing data: point, subsequence, and block. These patterns are specifically chosen to represent typical missing data scenarios in time series analysis, providing a robust framework for evaluating different imputation strategies. The missingness construction functions are directly available in the Python library `PyGrinder`[9]. The visualization of them is plotted in figures 6(a), 6(b), and 6(c) for vivid illustration.

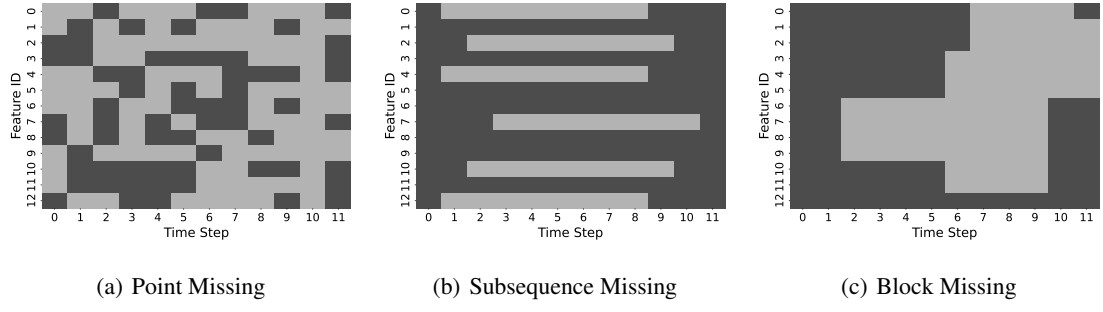

|          (a) Point Missing          |       (b) Subsequence Missing       |          (c) Block Missing          |

Figure 6: Heatmap visualization of three different missing patterns on the ItalyAir dataset. The observed values are presented in black, while the missing values are in grey.

---

[9]`https://github.com/WenjieDu/PyGrinder`

## C.2 Evaluation Metrics

Three metrics are applied to help evaluate the imputation performance in this work: MAE (Mean Absolute Error), MSE (Mean Square Error), and MRE (Mean Relative Error), whose math definitions are listed below. Note that errors are only calculated for values indicated by the mask in the input.

$$\text{MAE}(\hat{y}, y, m) = \frac{\sum_{d=1}^{D} \sum_{t=1}^{T} |(\hat{y}_t^d - y_t^d) \cdot m_t^d|}{\sum_{d=1}^{D} \sum_{t=1}^{T} m_t^d},$$

$$\text{MSE}(\hat{y}, y, m) = \frac{\sum_{d=1}^{D} \sum_{t=1}^{T} ((\hat{y}_t^d - y_t^d) \cdot m_t^d)^2}{\sum_{d=1}^{D} \sum_{t=1}^{T} m_t^d},$$

$$\text{MRE}(\hat{y}, y, m) = \frac{\sum_{d=1}^{D} \sum_{t=1}^{T} |(\hat{y}_t^d - y_t^d) \cdot m_t^d|}{\sum_{d=1}^{D} \sum_{t=1}^{T} |y_t^d \cdot m_t^d|},$$

where $\hat{y}$ is estimated value, $y$ indicates target value, and $m$ means mask with time index $t$ and dimension index $d$.

## C.3 Hyper-parameter Optimization

The performance of deep-learning models highly depends on the settings of hyper-parameters. To make fair comparisons across imputation methods and draw impartial conclusions, all neural network-based algorithms in TSI-Bench get their hyper-parameters optimized by `PyPOTS` and Microsoft NNI (Neural Network Intelligence). For each algorithm, we run at least 100 trials (i.e. tune it with 100 groups of hyper-parameters) and use the group of the best to produce the formal results. The HPO configuration files and the fixed model hyper-parameters are available in the supplementary material.

## C.4 Downstream Task Design

To further discuss the benefits that imputation can bring to downstream analysis, experiments are designed and conducted on three common downstream tasks (classification, regression, and forecasting) to evaluate the imputation quality across algorithms. For these three tasks, missing values in the selected datasets are imputed by different algorithms. Then XGBoost, LSTM, and Transformer perform downstream analysis tasks on the imputed datasets with fixed hyper-parameters. XGBoost, LSTM, and Transformer are selected because they are respectively representative models for classical machine learning approaches, traditional RNN methods, and emerging attention algorithms. Besides, thanks to XGBoost it can handle missing values by default. It allows us to compare XGBoost performance results with and without imputation to observe whether imputation can help.

**Classification**   PhysioNet2012 and Pedestrian are chosen for the classification task because every sample has a classification label. In PhysioNet2012, each sample has a label indicating if the patient is deceased in ICU, making it an unbalanced binary classification dataset with 13.9% positive samples. Sample classes in Pedestrian correspond to ten locations of sensor placement.

**Regression**   ETT_h1 and PeMS both have a target feature that makes them suitable for the regression and forecasting task. In the regression task, except for the target feature, the imputed data with other features are fed into the downstream algorithms to produce the regression results of the target feature.

**Forecasting**   For both ETT_h1 and PeMS datasets, we input the imputed data samples without their target features and the last five steps, then let the downstream models forecast the future five steps of the target

feature. Note that, different from the above regression task that inputs the full-length samples and outputs corresponding regression results, the forecasting task holds out the last five steps of the samples as ground truth to evaluate the forecasting performance, i.e. its input only contains (sample length - 5) steps.

As additional experiments, the forecasting downstream task is not mentioned in the main body of this paper, but its experimental results are presented in Appendix D.

### C.5 ADAPTATION PARADIGM

To tailor the time-series forecasting models for the imputation task, TSI-Bench adopts the imputation framework (including the embedding strategy and the training methodology) from SAITS (Du et al., 2023) and proposes such an adaptation paradigm that has been demonstrated to be feasible in this work's extensive experiments.

Fig. 7 presents an overview of how the forecasting models are transformed to impute missing values in time series. The forecasting model backbone is kept intact, and only the input and output processing are altered to align with the imputation task. In order to well train such a transformed model, the joint-optimization training approach designed for SAITS is employed here, as shown in Fig. 8.

Please refer to the SAITS paper (Du et al., 2023) for the details about how this imputation framework works.

### C.6 DEVELOPMENT ENVIRONMENT

All the experiments were conducted on the HPC platform, equipped with 512G RAM and 128 logical cores CPU (AMD EPYC 9554). One NVIDIA A100 80GB GPU is utilized for acceleration for each job, for a total of 32. The software environment is Ubuntu 12.3.0.

To help reproduce our Python environment easily, we freeze the development environment with Anaconda and save it into file for reference, which is available in the supplementary material.

Figure 7: An overview of the adaptation method for forecasting models.

## D FULL EXPERIMENTAL RESULTS

In the process of constructing the TSI-Bench, to analyse the imputation and downstream task performance of various imputation algorithms on different datasets and obtain valuable insights, we conduct extensive experiments. Specifically, we explore the imputation effects of 28 algorithms under 5 missing patterns and evaluate the performance of downstream tasks after imputation. It should be noted that, as PhysioNet2012 and PhysioNet2019 inherently contain a high proportion of missing data, these two datasets are not included in the experiments with 50% or 90% missing rates.

Note that MICN fails on Pedestrian because the official implementation of its backbone cannot accept univariate time series as input.

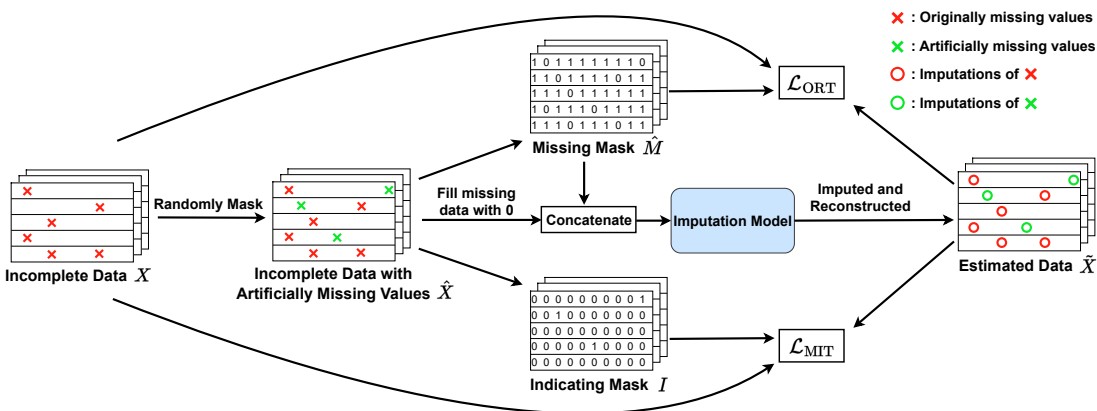

Figure 8: The training methodology from SAITS (Du et al., 2023) for the adapted forecasting models.

## D.1    10% POINT MISSING

Table 6 shows The details of the datasets involved in the experiments under the 10% missing setting. Table 11 shows the experimental results of the evaluated methods under the 10% missing setting, including size, the results of the three evaluation metrics previously mentioned, and inference time.

Table 6: Details of the preprocessed datasets with 10% point missingness.

| Dataset | Air | | Traffic | | Electricity | | Healthcare | |
|---|---|---|---|---|---|---|---|---|
| | BeijingAir | ItalyAir | PeMS | Pedestrian | ETT_h1 | Electricity | PhysioNet2012 | PhysioNet2019 |
| # of Total Samples | 1458 | 774 | 727 | 3633 | 358 | 1457 | 3997 | 4927 |
| # of Variables | 132 | 13 | 862 | 1 | 7 | 370 | 35 | 33 |
| Sample Sequence Length | 24 | 12 | 24 | 24 | 48 | 96 | 48 | 48 |
| Time Interval | 1H | 1H | 1H | 1H | 1H | 15Min | 1H | 1H |
| Original Missing Rate | 1.60% | 0% | 0% | 0% | 0% | 0% | 79.30% | 73.90% |
| Train Missing Rate | 11.69% | 10.14% | 10.01% | 9.79% | 9.96% | 9.99% | 80.50% | 78.45% |
| Validation Missing Rate | 10.85% | 9.77% | 9.99% | 10.13% | 10.24% | 10.00% | 82.66% | 80.30% |
| Test Missing Rate | 11.19% | 9.94% | 10.04% | 9.93% | 10.06% | 9.99% | 82.35% | 80.48% |
| Train Dataset Length | 851 | 466 | 455 | 955 | 212 | 851 | 2557 | 3152 |
| Validation Dataset Length | 304 | 154 | 122 | 239 | 75 | 304 | 640 | 789 |
| Test Dataset Length | 303 | 154 | 150 | 2439 | 71 | 302 | 800 | 986 |

## D.2    50% POINT MISSING

The details of the datasets involved are shown in Table 7, and Table 12 shows the experimental results of imputation with 50% point missing which is more challenging than with 10% point missing.

## D.3    90% POINT MISSING

The relative details of the datasets are shown in Table 8, and able 13 shows the imputation performance with 90% point missing. This is a relatively extreme scenario and brings difficulties in estimating the missing values.

Table 7: Details of the preprocessed datasets with 50% point missing.

| Dataset | Air | | Traffic | | Electricity | |
|---|---|---|---|---|---|---|
| | BeijingAir | ItalyAir | PeMS | Pedestrian | ETT_h1 | Electricity |
| # of Total Samples | 1458 | 774 | 727 | 3633 | 358 | 1457 |
| # of Variables | 132 | 13 | 862 | 1 | 7 | 370 |
| Sample Sequence Length | 24 | 12 | 24 | 24 | 48 | 96 |
| Time Interval | 1H | 1H | 1H | 1H | 1H | 15Min |
| Original Missing Rate | 1.60% | 0% | 0% | 0% | 0% | 0% |
| Train Missing Rate | 50.98% | 50.38% | 50.02% | 49.60% | 50.11% | 49.99% |
| Validation Missing Rate | 50.52% | 49.70% | 50.02% | 50.49% | 49.49% | 50.03% |
| Test Missing Rate | 50.70% | 49.96% | 50.03% | 49.78% | 49.97% | 49.97% |
| Train Dataset Length | 851 | 466 | 455 | 955 | 212 | 851 |
| Validation Dataset Length | 304 | 154 | 122 | 239 | 75 | 304 |
| Test Dataset Length | 303 | 154 | 150 | 2439 | 71 | 302 |

Table 8: Details of the preprocessed datasets with 90% point missing.

| Dataset | Air | | Traffic | | Electricity | |
|---|---|---|---|---|---|---|
| | BeijingAir | ItalyAir | PeMS | Pedestrian | ETT_h1 | Electricity |
| # of Total Samples | 1458 | 774 | 727 | 3633 | 358 | 1457 |
| # of Variables | 132 | 13 | 862 | 1 | 7 | 370 |
| Sample Sequence Length | 24 | 12 | 24 | 24 | 48 | 96 |
| Time Interval | 1H | 1H | 1H | 1H | 1H | 15Min |
| Original Missing Rate | 1.60% | 0% | 0% | 0% | 0% | 0% |
| Train Missing Rate | 90.21% | 90.07% | 90.00% | 90.37% | 90.11% | 90.00% |
| Validation Missing Rate | 90.12% | 89.81% | 90.01% | 90.17% | 89.77% | 90.02% |
| Test Missing Rate | 90.21% | 89.94% | 90.00% | 89.74% | 90.34% | 90.00% |
| Train Dataset Length | 851 | 466 | 455 | 955 | 212 | 851 |
| Validation Dataset Length | 304 | 154 | 122 | 239 | 75 | 304 |
| Test Dataset Length | 303 | 154 | 150 | 2439 | 71 | 302 |

Table 9: Details of the preprocessed datasets with 50% subsequence missing.

| Dataset | Air | | Traffic | | Electricity | |
|---|---|---|---|---|---|---|
| | BeijingAir | ItalyAir | PeMS | Pedestrian | ETT_h1 | Electricity |
| # of Total Samples | 1458 | 774 | 727 | 3633 | 358 | 1457 |
| # of Variables | 132 | 13 | 862 | 1 | 7 | 370 |
| Sample Sequence Length | 24 | 12 | 24 | 24 | 48 | 96 |
| Time Interval | 1H | 1H | 1H | 1H | 1H | 15Min |
| Original Missing Rate | 1.60% | 0% | 0% | 0% | 0% | 0% |
| Train Missing Rate | 50.92% | 50.01% | 50.00% | 50.03% | 50.03% | 50.00% |
| Validation Missing Rate | 50.45% | 50.02% | 50.00% | 50.21% | 50.00% | 50.00% |
| Test Missing Rate | 50.61% | 50.02% | 50.00% | 50.00% | 50.10% | 50.00% |
| Train Dataset Length | 851 | 466 | 455 | 955 | 212 | 851 |
| Validation Dataset Length | 304 | 154 | 122 | 239 | 75 | 304 |
| Test Dataset Length | 303 | 154 | 150 | 2439 | 71 | 302 |

Table 10: Details of the preprocessed datasets with 50% block missing. The values of Pedestrian are the same as those in Table 9 because this dataset has only 1 feature that makes subsequence missing and block missing identical.

| Dataset | Air | | Traffic | | Electricity | |
|---|---|---|---|---|---|---|
| | BeijingAir | ItalyAir | PeMS | Pedestrian | ETT_h1 | Electricity |
| # of Total Samples | 1458 | 774 | 727 | 3633 | 358 | 1457 |
| # of Variables | 132 | 13 | 862 | 1 | 7 | 370 |
| Sample Sequence Length | 24 | 12 | 24 | 24 | 48 | 96 |
| Time Interval | 1H | 1H | 1H | 1H | 1H | 15Min |
| Original Missing Rate | 1.60% | 0% | 0% | 0% | 0% | 0% |
| Train Missing Rate | 51.19% | 50.82% | 49.96% | 50.03% | 49.11% | 50.73% |
| Validation Missing Rate | 50.83% | 50.97% | 50.03% | 50.21% | 50.42% | 50.75% |
| Test Missing Rate | 50.92% | 50.28% | 50.03% | 50.00% | 49.58% | 50.73% |
| Train Dataset Length | 851 | 466 | 455 | 955 | 212 | 851 |
| Validation Dataset Length | 304 | 154 | 122 | 239 | 75 | 304 |
| Test Dataset Length | 303 | 154 | 150 | 2439 | 71 | 302 |

## D.4   50% SUBSEQUENCE MISSING

The details of the preprocessed datasets in this setting are shown as Table 9, and table 14 shows the imputation results with 50% subsequence missing. It could be observed that the imputation error is generally higher than that with 50% point missing. In this condition, the missing values in the subsequence cannot be easily estimated from their adjacent observed data.

## D.5   50% BLOCK MISSING

Table 10 shows the relative dataset details, and table 15 shows the imputation results under the scenario of 50% block missing. This missing pattern includes missing data across multiple dimensions at the same time points and consecutive missing data at multiple time points, thus presenting challenges for imputation.

## D.6   VISUALIZATION OF IMPUTATION PERFORMANCE

We provide the visualization of the imputation examples by different imputation methods on the ETT_h1 and Electricity datasets. Hereby, we display only the imputation results of SAITS in Figures 9 and 10. Results from other imputation algorithms are available under the folder "Imputation comparison" of supplementary material.

## D.7   VISUALIZATION OF TIME INTERVALS

We conducted a detailed statistical analysis and visualization of the time intervals in the PhysioNet2012 dataset, as shown in Figures 11. It is crucial to emphasize that the uneven time intervals in medical data represent a significant and common issue. As the figure illustrates, features such as heart rate (HR) and diastolic arterial blood pressure (DiasABP) tend to have shorter time intervals, while features like white blood cell count (WBC) and sodium levels (Na) have much longer intervals. This discrepancy arises from the nature of these measurements—vital signs like heart rate can be continuously monitored by instruments, whereas blood tests are inherently spaced out due to the invasive nature of the procedures.

Furthermore, different types of features, such as continuous variables (e.g., HR, DiasABP) versus categorical variables (e.g., Glasgow Coma Scale [GCS]), exhibit significant variability in their measurement intervals. Existing methods like BRITS and GRU-D attempt to address this by incorporating decay mechanisms,

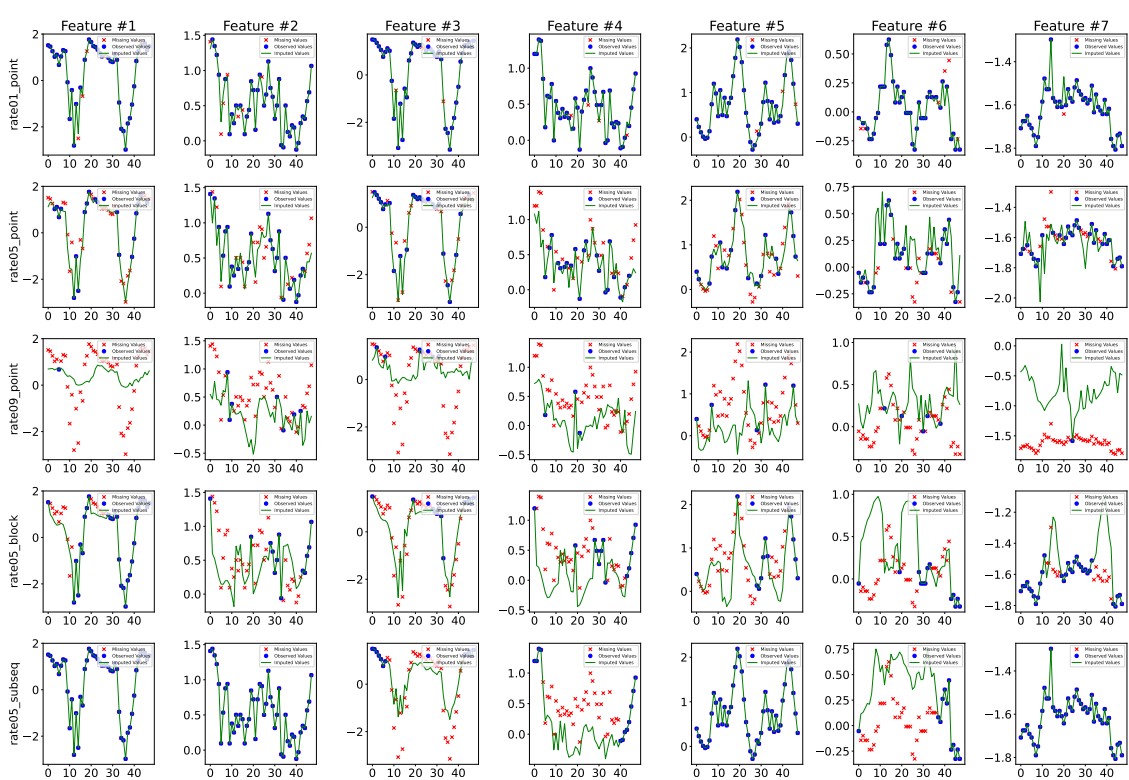

Figure 9: Visualization of imputation performance by SAITS on the ETT_h1 dataset.

where the influence of an observation decreases as the time interval increases. However, these approaches often overlook the inherent regularity of certain medical measurements. Handling unequal time intervals in healthcare time series is not just about filling in the gaps, but about the need to understand the context in data collection and ensuring that imputation methods respect the temporal dynamics of the data.

## D.8 EXPERIMENTS ON DOWNSTREAM TASKS

### D.8.1 CLASSIFICATION

Table 16 and 17 show the classification performance without and with imputation. In general, the classification performance can be improved after an imputation process for time series with missing values. This observation is particularly evident when using XGBoost as the classification method on the PhysioNet2012 dataset with a missing rate of 10%.

### D.8.2 REGRESSION

Table 18, 19 and 20 show the regression performance without and with imputation under 3 different missing patterns. It can be found that when XGBoost is used as the regression model, using advanced algorithms to impute missing values can improve the regression performance overall. However, for some simple methods

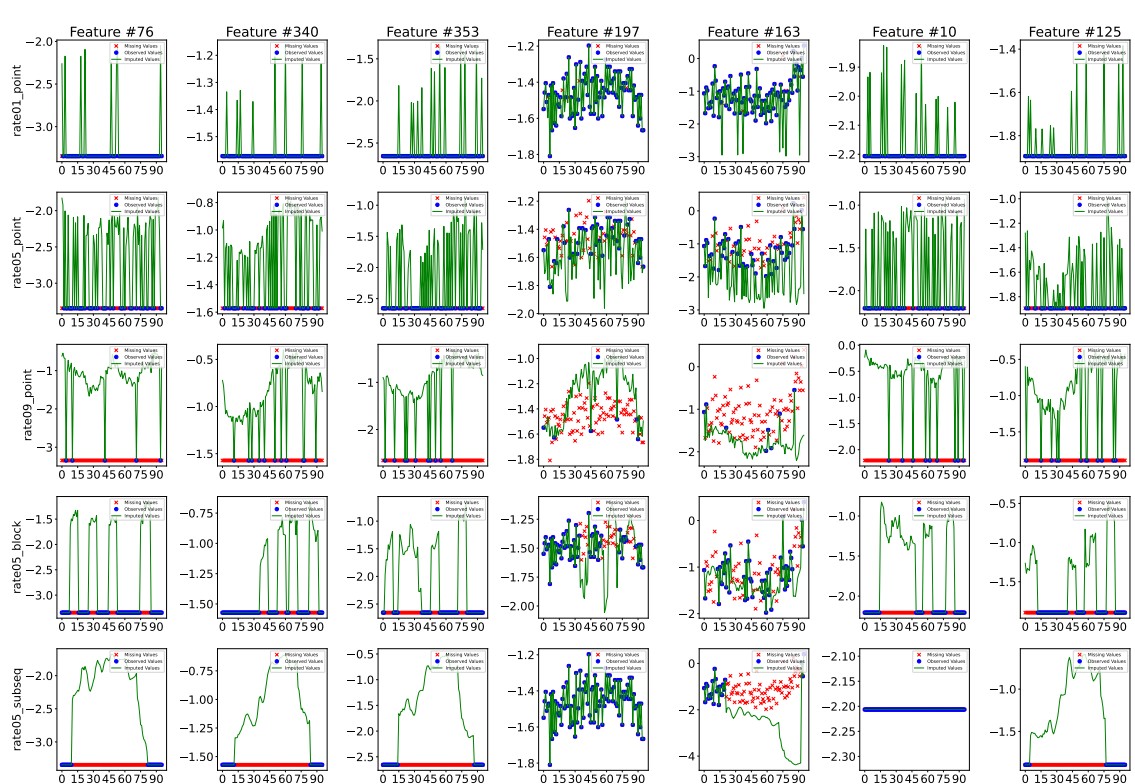

Figure 10: Visualization of imputation performance by SAITS on the Electricity dataset.

(such as Mean), the regression effect after imputation may not improve or may become worse, which also shows to some extent that it is meaningful and valuable to research the missing value imputation in time series.

### D.8.3 FORECASTING

Table 21, 22 and 23 show the forecasting performance without or with imputation under different missing patterns and show the same phenomenon when regression is the downstream task, that is, using advanced algorithms to impute missing values can improve the performance of downstream tasks, while using simple methods like Mean may not have a positive impact on forecasting performance.

### D.9 THE TOTAL NUMBER OF EXPERIMENTS

We conduct a total of 34,804 experiments across 28 algorithms and 8 datasets, focusing on 3 angles (i.e., on the data level, the model level, and the downstream tasks) for a comprehensive and fair evaluation and analysis through the experiments. Note that duplicated experiments for obtaining the final stable results are not included here.

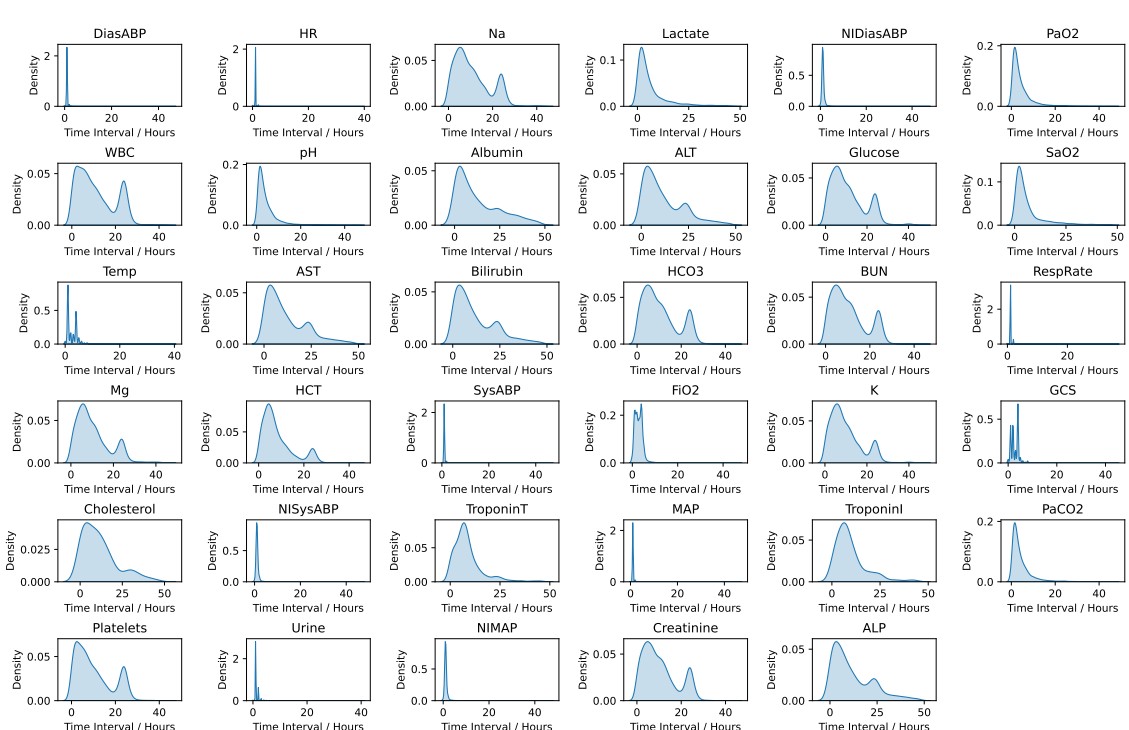

Figure 11: Density distribution of time intervals across features in PhysioNet2012 dataset.

**HPO experiments**. For 24 neural networks, we run 100 HPO trials for each of them on each dataset, with $24 * 100 * 8 = 19,200$ experiments.

**Imputation experiments**. For 24 neural network models, we run 5 rounds on 8 datasets with 10% point missing, 6 datasets with 50% point missing, 6 datasets with 90% point missing, 6 datasets with 50% subsequence missing, and 5 datasets with 50% block missing, with $24 * 5 * (8 + 6 + 6 + 6 + 5) = 3,720$ experiments. We also conduct $4 * (8 + 6 + 6 + 6 + 5) = 124$ experiments for 4 naive imputation methods.

**Downstream experiments**. Experiments on the downstream tasks are performed in 5 rounds with 4 algorithms on each dataset imputed by 28 methods. The classification task is on PhysioNet2012 with 10% point missing in the evaluation sets and Pedestrian with 10%, 50%, and 90% point missing, and 50% subsequence missing. The regression and forecasting tasks are on ETT_h1 and PeMS with 10% and 50% points missing, 50% subsequence missing and block missing. So the total number of downstream tasks is $5 * 4 * 28 * (1 + 4) + 5 * 4 * 28 * (4 + 4) + 5 * 4 * 28 * (4 + 4) = 11,760$ experiments.

Table 11: Performance comparison in 8 datasets with 10% point missing.

Table 12: Performance comparison in 6 datasets with 50% point missing.

| | BeijingAir | | | | | ItalyAir | | | | | PeMS | | | | |
|---|---|---|---|---|---|---|---|---|---|---|---|---|---|---|---|
| | Size | MAE | MSE | MRE | Time | Size | MAE | MSE | MRE | Time | Size | MAE | MSE | MRE | Time |
| iTransformer | 8,286,232 | 0.163 (0.003) | 0.233 (0.004) | 0.215 (0.004) | 0.34 | 18,932,236 | 0.321 (0.007) | 0.327 (0.011) | 0.419 (0.009) | 0.18 | 1,854,744 | 0.295 (0.007) | 0.539 (0.016) | 0.366 (0.009) | 0.46 |
| SATTS | 7,153,808 | 0.194 (0.003) | 0.193 (0.007) | 0.257 (0.004) | 0.24 | 16,628,642 | 0.285 (0.010) | 0.236 (0.014) | 0.372 (0.013) | 0.07 | 78,229,072 | 0.302 (0.001) | 0.595 (0.003) | 0.375 (0.001) | 0.17 |
| Nonstationary | 6,978,068 | 0.231 (0.001) | 0.271 (0.007) | 0.306 (0.002) | 0.39 | 8,441,077 | 0.314 (0.005) | 0.361 (0.010) | 0.410 (0.007) | 0.23 | 346,318 | 0.394 (0.013) | 0.688 (0.016) | 0.488 (0.016) | 0.24 |
| ETSformer | 7,928,510 | 0.249 (0.004) | 0.261 (0.009) | 0.331 (0.005) | 1.62 | 8,009,937 | 0.401 (0.007) | 0.421 (0.011) | 0.524 (0.009) | 0.22 | 5,962,188 | 0.386 (0.007) | 0.586 (0.009) | 0.479 (0.008) | 0.97 |
| PatchTST | 30,342,300 | 0.210 (0.009) | 0.206 (0.007) | 0.279 (0.012) | 4.51 | 5,077,145 | 0.345 (0.011) | 0.313 (0.010) | 0.451 (0.015) | 0.48 | 3,045,238 | 0.348 (0.006) | 0.609 (0.008) | 0.431 (0.007) | 0.38 |
| Crossformer | 52,933,788 | 0.215 (0.007) | 0.224 (0.004) | 0.285 (0.009) | 6.83 | 2,908,185 | 0.325 (0.009) | 0.293 (0.009) | 0.425 (0.012) | 0.13 | 12,645,238 | 0.357 (0.003) | 0.607 (0.008) | 0.443 (0.004) | 0.32 |
| Informer | 6,706,308 | 0.184 (0.005) | 0.213 (0.003) | 0.244 (0.006) | 0.66 | 10,540,045 | 0.304 (0.007) | 0.247 (0.015) | 0.397 (0.012) | 0.42 | 13,149,022 | 0.330 (0.005) | 0.600 (0.009) | 0.409 (0.006) | 0.81 |
| Autoformer | 6,700,164 | 0.898 (0.001) | 1.554 (0.003) | 1.190 (0.001) | 0.35 | 993,805 | 0.833 (0.017) | 1.880 (0.044) | 1.089 (0.023) | 0.23 | 608,926 | 0.602 (0.068) | 1.242 (0.173) | 0.747 (0.084) | 0.22 |
| Pyraformer | 3,230,212 | 0.198 (0.005) | 0.198 (0.005) | 0.263 (0.006) | 0.33 | 11,355,917 | 0.312 (0.012) | 0.254 (0.018) | 0.408 (0.015) | 0.11 | 4,048,606 | 0.305 (0.002) | 0.580 (0.004) | 0.379 (0.003) | 0.21 |
| Transformer | 203,038,852 | 0.185 (0.003) | 0.192 (0.005) | 0.245 (0.004) | 0.47 | 4,749,837 | 0.279 (0.011) | 0.230 (0.017) | 0.365 (0.014) | 0.07 | 23,135,326 | 0.316 (0.004) | 0.588 (0.005) | 0.392 (0.005) | 0.13 |
| BRITS | 3,598,496 | 0.169 (0.001) | 0.194 (0.003) | 0.224 (0.001) | 26.96 | 596,912 | 0.321 (0.005) | 0.283 (0.007) | 0.420 (0.007) | 0.62 | 32,012,048 | 0.287 (0.001) | 0.561 (0.002) | 0.357 (0.001) | 5.44 |
| MRNN | 96,585 | 0.603 (0.006) | 0.775 (0.009) | 0.799 (0.007) | 2.85 | 402,111 | 0.724 (0.003) | 1.391 (0.006) | 0.947 (0.003) | 0.62 | 3,076,301 | 0.645 (0.001) | 1.072 (0.003) | 0.800 (0.001) | 1.57 |
| GRID | 7,397,656 | 0.279 (0.001) | 0.303 (0.000) | 0.370 (0.001) | 1.1 | 112,707 | 0.476 (0.009) | 0.539 (0.011) | 0.622 (0.012) | 0.41 | 14,104,896 | 0.372 (0.002) | 0.619 (0.002) | 0.462 (0.002) | 0.82 |
| TimesNet | 87,063,940 | 0.265 (0.005) | 0.233 (0.002) | 0.351 (0.007) | 0.64 | 22,051,853 | 0.370 (0.010) | 0.323 (0.012) | 0.484 (0.014) | 0.08 | 91,622,238 | 0.348 (0.002) | 0.567 (0.001) | 0.432 (0.003) | 0.37 |
| MICN | 57,048,200 | 0.456 (0.006) | 0.553 (0.013) | 0.604 (0.008) | 0.09 | 695,569 | 0.548 (0.003) | 0.852 (0.012) | 0.717 (0.004) | 0.05 | 15,490,402 | 0.392 (0.006) | 0.608 (0.010) | 0.486 (0.007) | 0.17 |
| SCINet | 26,833,140 | 0.222 (0.012) | 0.230 (0.036) | 0.294 (0.016) | 0.31 | 263,517 | 0.337 (0.008) | 0.319 (0.006) | 0.441 (0.010) | 0.09 | 1,143,027,230 | 0.500 (0.093) | 0.849 (0.193) | 0.621 (0.115) | 0.41 |
| StemGNN | 2,645,628 | 0.186 (0.004) | 0.263 (0.005) | 0.246 (0.005) | 0.41 | 926,737 | 0.307 (0.014) | 0.280 (0.019) | 0.401 (0.018) | 0.18 | 2,386,294 | 0.446 (0.021) | 0.862 (0.064) | 0.553 (0.026) | 0.27 |
| FreTS | 909,852 | 0.235 (0.015) | 0.246 (0.010) | 0.311 (0.019) | 0.19 | 668,313 | 0.349 (0.015) | 0.345 (0.053) | 0.457 (0.020) | 0.07 | 1,715,958 | 0.422 (0.019) | 0.686 (0.027) | 0.524 (0.024) | 0.16 |
| Koopa | 563,692 | 0.373 (0.079) | 0.445 (0.105) | 0.494 (0.105) | 0.44 | 1,403,525 | 0.345 (0.032) | 0.359 (0.056) | 0.452 (0.042) | 0.07 | 13,306,214 | 0.506 (0.114) | 0.855 (0.184) | 0.628 (0.141) | 0.32 |
| DLinear | 204,728 | 0.245 (0.005) | 0.242 (0.006) | 0.324 (0.006) | 0.19 | 5,458 | 0.340 (0.004) | 0.337 (0.005) | 0.445 (0.005) | 0.05 | 5,301,100 | 0.389 (0.013) | 0.604 (0.020) | 0.483 (0.017) | 0.13 |
| FILM | 408,807 | 0.331 (0.009) | 0.409 (0.009) | 0.439 (0.012) | 0.5 | 43,072 | 0.402 (0.018) | 0.468 (0.052) | 0.525 (0.023) | 0.12 | 2,652,097 | 0.781 (0.059) | 1.499 (0.124) | 0.969 (0.073) | 0.26 |
| CSDI | 244,833 | 0.144 (0.007) | 0.472 (0.155) | 0.192 (0.009) | 390.62 | 933,161 | 0.958 (0.551) | 29.266 (31.183) | 1.253 (0.720) | 12.62 | 207,873 | 0.288 (0.040) | 0.651 (0.090) | 0.358 (0.049) | 265.37 |
| US-GAN | 6,123,812 | 0.192 (0.001) | 0.187 (0.005) | 0.255 (0.001) | 0.51 | 3,913,149 | 0.357 (0.009) | 0.278 (0.011) | 0.467 (0.012) | 0.18 | 50,674,286 | 0.330 (0.001) | 0.566 (0.001) | 0.409 (0.001) | 0.32 |
| GP-VAE | 1,013,913 | 0.258 (0.004) | 0.234 (0.008) | 0.343 (0.005) | 1.38 | 130,594 | 0.453 (0.014) | 0.495 (0.022) | 0.592 (0.018) | 0.47 | 2,396,536 | 0.346 (0.015) | 0.617 (0.015) | 0.430 (0.019) | 9.52 |
| Mean | / | 0.708 | 1.078 | 0.964 | / | / | 0.588 | 1.096 | 0.769 | / | / | 0.799 | 1.416 | 0.991 | / |
| Median | / | 0.677 | 1.143 | 0.922 | / | / | 0.533 | 1.116 | 0.697 | / | / | 0.777 | 1.476 | 0.965 | / |
| LOCF | / | 0.264 | 0.429 | 0.36 | / | / | 0.346 | 0.511 | 0.452 | / | / | 0.547 | 1.094 | 0.679 | / |
| Linear | / | 0.165 | 0.231 | 0.224 | / | / | 0.214 | 0.252 | 0.279 | / | / | 0.343 | 0.539 | 0.426 | / |

| | ETT_h1 | | | | | Electricity | | | | | Pedestrian | | | | |
|---|---|---|---|---|---|---|---|---|---|---|---|---|---|---|---|
| | Size | MAE | MSE | MRE | Time | Size | MAE | MSE | MRE | Time | Size | MAE | MSE | MRE | Time |
| iTransformer | 23,723,056 | 0.348 (0.002) | 0.233 (0.003) | 0.412 (0.002) | 0.07 | 12,989,024 | 0.893 (0.085) | 1.884 (0.160) | 0.478 (0.046) | 1.03 | 2,913,304 | 0.200 (0.006) | 0.343 (0.006) | 0.264 (0.008) | 0.61 |
| SATTS | 88,235,470 | 0.223 (0.007) | 0.107 (0.005) | 0.264 (0.008) | 0.18 | 63,624,720 | 1.399 (0.069) | 3.837 (0.316) | 0.749 (0.037) | 1.11 | 133,016 | 0.205 (0.011) | 0.392 (0.027) | 0.270 (0.015) | 1.79 |
| Nonstationary | 589,927 | 0.382 (0.004) | 0.292 (0.006) | 0.452 (0.005) | 0.06 | 24,811,090 | 0.217 (0.031) | 0.191 (0.048) | 0.116 (0.016) | 0.83 | 6,338,833 | 0.487 (0.033) | 0.859 (0.098) | 0.641 (0.043) | 1.4 |
| ETSformer | 809,057 | 0.364 (0.013) | 0.269 (0.022) | 0.431 (0.015) | 0.14 | 10,518,266 | 0.878 (0.008) | 1.687 (0.024) | 0.470 (0.004) | 1.84 | 106,905 | 0.320 (0.004) | 0.519 (0.010) | 0.421 (0.006) | 2.13 |
| PatchTST | 72,247 | 0.275 (0.023) | 0.149 (0.017) | 0.326 (0.027) | 0.04 | 4,419,410 | 0.856 (0.044) | 1.573 (0.141) | 0.459 (0.024) | 4.33 | 202,905 | 0.198 (0.003) | 0.351 (0.005) | 0.260 (0.003) | 0.65 |
| Crossformer | 223,479 | 0.270 (0.021) | 0.146 (0.017) | 0.319 (0.025) | 0.04 | 9,967,314 | 0.980 (0.344) | 2.255 (1.656) | 0.525 (0.184) | 1.7 | 446,785 | 0.191 (0.006) | 0.356 (0.014) | 0.251 (0.010) | 1.12 |
| Informer | 1,058,311 | 0.279 (0.008) | 0.162 (0.007) | 0.330 (0.009) | 0.11 | 15,311,986 | 1.277 (0.028) | 3.239 (0.080) | 0.684 (0.015) | 1 | 246,145 | 0.210 (0.006) | 0.378 (0.021) | 0.277 (0.007) | 1.66 |
| Autoformer | 166,919 | 0.984 (0.008) | 1.553 (0.025) | 1.164 (0.010) | 0.13 | 7,431,538 | 2.164 (0.001) | 8.092 (0.010) | 1.159 (0.001) | 0.54 | 957,057 | 1.033 (0.015) | 2.273 (0.082) | 1.359 (0.020) | 3.14 |
| Pyraformer | 15,262,215 | 0.291 (0.022) | 0.167 (0.021) | 0.345 (0.031) | 0.21 | 15,940,914 | 1.131 (0.036) | 2.711 (0.079) | 0.606 (0.019) | 1.36 | 13,787,649 | 0.202 (0.006) | 0.381 (0.007) | 0.266 (0.007) | 1.71 |
| Transformer | 5,800,199 | 0.274 (0.012) | 0.162 (0.017) | 0.325 (0.014) | 0.08 | 155,610,482 | 1.365 (0.034) | 3.554 (0.085) | 0.731 (0.018) | 1.91 | | 0.194 (0.014) | 0.342 (0.033) | 0.255 (0.018) | 1.53 |
| BRITS | 2,178,496 | 0.238 (0.006) | 0.127 (0.004) | 0.281 (0.007) | 0.71 | 17,082,800 | 1.124 (0.010) | 2.828 (0.023) | 0.602 (0.005) | 39.21 | 8,427,536 | 0.259 (0.017) | 0.433 (0.021) | 0.341 (0.022) | 10.86 |
| MRNN | 2,259 | 0.816 (0.006) | 1.219 (0.013) | 0.965 (0.007) | 0.49 | 949,749 | 1.810 (0.004) | 5.793 (0.011) | 0.969 (0.002) | 12.76 | 401,415 | 0.773 (0.001) | 1.258 (0.001) | 1.017 (0.001) | 11.24 |
| GRID | 409,407 | 0.417 (0.008) | 0.337 (0.014) | 0.493 (0.010) | 0.42 | 9,467,304 | 1.087 (0.011) | 2.458 (0.034) | 0.582 (0.006) | 3.43 | 100,227 | 0.307 (0.005) | 0.507 (0.007) | 0.404 (0.006) | 6.86 |
| TimesNet | 5,510,663 | 0.339 (0.004) | 0.210 (0.008) | 0.401 (0.005) | 0.2 | 45,569,394 | 1.131 (0.017) | 2.644 (0.077) | 0.606 (0.009) | 1.59 | 10,816,385 | 0.269 (0.016) | 0.392 (0.017) | 0.354 (0.021) | 3.02 |
| MICN | 3,153,163 | 0.606 (0.073) | 0.688 (0.152) | 0.717 (0.086) | 0.15 | 5,457,910 | 0.965 (0.008) | 2.018 (0.032) | 0.516 (0.004) | 0.47 | 43,783 | 0.251 (0.005) | 0.391 (0.015) | 0.331 (0.007) | 1.98 |
| SCINet | 79,493 | 0.326 (0.014) | 0.194 (0.013) | 0.386 (0.016) | 0.09 | 421,053,386 | 0.778 (0.023) | 1.162 (0.115) | 0.417 (0.012) | 1.48 | | | | | |
| StemGNN | 6,397,975 | 0.325 (0.019) | 0.200 (0.025) | 0.385 (0.023) | 0.26 | 16,863,634 | 1.362 (0.187) | 3.803 (0.920) | 0.729 (0.100) | 1.69 | 1,638,337 | 0.200 (0.009) | 0.343 (0.014) | 0.263 (0.011) | 2.99 |
| FreTS | 465,271 | 0.319 (0.025) | 0.195 (0.030) | 0.378 (0.029) | 0.05 | 3,706,194 | 0.871 (0.084) | 1.320 (0.275) | 0.466 (0.045) | 0.54 | 116,825 | 0.224 (0.004) | 0.314 (0.016) | 0.294 (0.005) | 0.84 |
| Koopa | 465,389 | 0.515 (0.159) | 0.577 (0.351) | 0.610 (0.188) | 0.02 | 2,680,114 | 1.755 (0.250) | 7.390 (1.677) | 0.940 (0.134) | 0.61 | 124,711 | 0.246 (0.017) | 0.330 (0.030) | 0.324 (0.023) | 0.85 |
| DLinear | 7,534 | 0.311 (0.003) | 0.186 (0.003) | 0.368 (0.004) | 0.03 | 2,294,692 | 0.734 (0.011) | 0.988 (0.038) | 0.393 (0.006) | 0.27 | 3,250 | 0.310 (0.002) | 0.455 (0.006) | 0.408 (0.002) | 0.82 |
| FILM | 12,490 | 0.589 (0.005) | 0.793 (0.003) | 0.697 (0.005) | 0.11 | 570,613 | 0.907 (0.024) | 1.434 (0.078) | 0.485 (0.013) | 0.62 | 6,244 | 0.453 (0.007) | 0.664 (0.008) | 0.596 (0.009) | 1.67 |
| CSDI | 1,194,993 | 0.318 (0.016) | 0.207 (0.011) | 0.376 (0.019) | 10.99 | 43,185 | 0.798 (0.455) | 21.850 (22.140) | 0.427 (0.244) | 986.19 | 325,473 | 0.351 (0.074) | 1.117 (0.220) | 0.462 (0.098) | 139.12 |
| US-GAN | 3,807,687 | 0.755 (0.973) | 2.119 (3.955) | 0.893 (1.151) | 0.27 | 11,224,866 | 1.119 (0.007) | 2.610 (0.018) | 0.599 (0.004) | 2.22 | 14,745,617 | 0.233 (0.005) | 0.328 (0.007) | 0.307 (0.007) | 3.75 |
| GP-VAE | 384,796 | 0.414 (0.013) | 0.301 (0.011) | 0.490 (0.015) | 0.23 | 1,825,022 | 1.099 (0.032) | 2.973 (0.040) | 0.588 (0.017) | 16.73 | 284,676 | 0.451 (0.022) | 0.677 (0.031) | 0.594 (0.029) | 2.01 |
| Mean | / | 0.738 | 0.971 | 0.873 | / | / | 0.423 | 0.581 | 0.227 | / | / | 0.763 | 1.258 | 1.004 | / |
| Median | / | 0.708 | 1.022 | 0.837 | / | / | 0.408 | 0.627 | 0.219 | / | / | 0.705 | 1.386 | 0.928 | / |
| LOCF | / | 0.425 | 0.491 | 0.502 | / | / | 0.14 | 0.181 | 0.075 | / | / | 0.365 | 0.636 | 0.481 | / |
| Linear | / | 0.267 | 0.178 | 0.316 | / | / | 0.078 | 0.035 | 0.042 | / | / | 0.247 | 0.279 | 0.326 | / |

Table 13: Performance comparison in 6 datasets with 90% point missing.

| | BeijingAir | | | | | ItalyAir | | | | | PeMS | | | | |
|---|---|---|---|---|---|---|---|---|---|---|---|---|---|---|---|
| | Size | MAE | MSE | MRE | Time | Size | MAE | MSE | MRE | Time | Size | MAE | MSE | MRE | Time |
| iTransformer | 8,286,232 | 0.352 (0.005) | 0.514 (0.008) | 0.468 (0.007) | 0.34 | 18,932,236 | 0.574 (0.002) | 0.836 (0.016) | 0.755 (0.003) | 0.17 | 1,854,744 | 0.450 (0.007) | 0.875 (0.015) | 0.558 (0.009) | 0.49 |
| SAITS | 7,153,808 | 0.331 (0.024) | 0.444 (0.034) | 0.439 (0.031) | 0.25 | 16,628,642 | 0.483 (0.012) | 0.590 (0.021) | 0.635 (0.016) | 0.07 | 78,229,072 | 0.353 (0.001) | 0.692 (0.003) | 0.438 (0.002) | 0.2 |
| Nonstationary | 6,978,068 | 0.357 (0.002) | 0.531 (0.011) | 0.473 (0.002) | 0.47 | 8,441,077 | 0.489 (0.008) | 0.755 (0.021) | 0.643 (0.011) | 0.2 | 346,318 | 0.736 (0.003) | 1.589 (0.013) | 0.913 (0.003) | 0.27 |
| ETSformer | 7,928,510 | 0.371 (0.011) | 0.495 (0.020) | 0.492 (0.015) | 1.82 | 8,009,937 | 0.682 (0.012) | 1.419 (0.028) | 0.896 (0.015) | 0.45 | 5,962,188 | 0.474 (0.009) | 0.778 (0.014) | 0.589 (0.012) | 0.99 |
| PatchTST | 30,342,300 | 0.299 (0.005) | 0.372 (0.003) | 0.396 (0.006) | 4.73 | 5,077,145 | 0.577 (0.093) | 0.996 (0.367) | 0.758 (0.122) | 0.55 | 3,045,238 | 0.440 (0.057) | 0.793 (0.107) | 0.546 (0.071) | 0.34 |
| Crossformer | 52,933,788 | 0.274 (0.003) | 0.355 (0.003) | 0.364 (0.004) | 6.89 | 2,908,185 | 0.458 (0.005) | 0.523 (0.010) | 0.602 (0.007) | 0.1 | 12,645,238 | 0.399 (0.007) | 0.733 (0.004) | 0.495 (0.008) | 0.32 |
| Informer | 6,706,308 | 0.258 (0.010) | 0.335 (0.011) | 0.342 (0.013) | 0.65 | 10,540,045 | 0.491 (0.005) | 0.613 (0.010) | 0.645 (0.007) | 0.45 | 13,149,022 | 0.376 (0.001) | 0.708 (0.008) | 0.467 (0.002) | 0.74 |
| Autoformer | 6,700,164 | 0.806 (0.002) | 1.375 (0.001) | 1.069 (0.002) | 0.26 | 993,805 | 0.806 (0.007) | 1.766 (0.015) | 1.059 (0.010) | 0.2 | 608,926 | 0.773 (0.099) | 1.488 (0.170) | 0.959 (0.123) | 0.25 |
| Pyraformer | 3,230,212 | 0.274 (0.005) | 0.372 (0.008) | 0.364 (0.007) | 0.32 | 11,355,917 | 0.517 (0.013) | 0.602 (0.027) | 0.679 (0.017) | 0.12 | 4,048,606 | 0.374 (0.010) | 0.707 (0.010) | 0.464 (0.001) | 0.22 |
| Transformer | 203,038,852 | 0.277 (0.005) | 0.333 (0.004) | 0.368 (0.007) | 0.4 | 4,749,837 | 0.459 (0.010) | 0.552 (0.033) | 0.604 (0.013) | 0.07 | 23,135,326 | 0.376 (0.002) | 0.711 (0.007) | 0.466 (0.003) | 0.14 |
| BRITS | 3,598,496 | 0.316 (0.003) | 0.419 (0.007) | 0.420 (0.005) | 22.45 | 596,912 | 0.522 (0.007) | 0.656 (0.013) | 0.687 (0.009) | 0.84 | 32,012,048 | 0.351 (0.002) | 0.705 (0.004) | 0.436 (0.003) | 5.48 |
| MRNN | 96,585 | 0.759 (0.005) | 1.121 (0.013) | 1.007 (0.006) | 2.99 | 402,111 | 0.758 (0.001) | 1.495 (0.002) | 0.996 (0.002) | 0.37 | 3,076,301 | 0.747 (0.002) | 1.317 (0.001) | 0.927 (0.002) | 4.06 |
| GRUD | 7,397,656 | 0.373 (0.003) | 0.461 (0.004) | 0.495 (0.004) | 1.1 | 112,707 | 0.677 (0.020) | 0.976 (0.055) | 0.890 (0.026) | 0.42 | 14,104,896 | 0.417 (0.002) | 0.710 (0.001) | 0.517 (0.003) | 0.81 |
| TimesNet | 87,063,940 | 0.337 (0.009) | 0.375 (0.008) | 0.447 (0.011) | 0.62 | 22,051,853 | 0.587 (0.021) | 0.670 (0.037) | 0.771 (0.027) | 0.11 | 91,622,238 | 0.412 (0.002) | 0.712 (0.002) | 0.511 (0.003) | 0.37 |
| MICN | 57,048,200 | 0.685 (0.005) | 1.090 (0.010) | 0.909 (0.006) | 0.04 | 695,569 | 0.685 (0.009) | 1.372 (0.019) | 0.900 (0.021) | 0.02 | 15,490,402 | 0.498 (0.005) | 0.934 (0.010) | 0.618 (0.006) | 0.16 |
| SCINet | 26,833,140 | 0.300 (0.008) | 0.371 (0.020) | 0.398 (0.010) | 0.31 | 263,517 | 0.532 (0.021) | 0.746 (0.081) | 0.699 (0.027) | 0.1 | 1,143,027,230 | 0.538 (0.066) | 0.947 (0.124) | 0.667 (0.082) | 0.37 |
| StemGNN | 2,645,628 | 0.267 (0.008) | 0.367 (0.006) | 0.355 (0.011) | 0.44 | 926,737 | 0.454 (0.012) | 0.513 (0.025) | 0.597 (0.016) | 0.18 | 2,386,294 | 0.430 (0.007) | 0.833 (0.015) | 0.534 (0.008) | 0.23 |
| FreTS | 909,852 | 0.271 (0.023) | 0.367 (0.014) | 0.359 (0.030) | 0.17 | 668,313 | 0.491 (0.014) | 0.641 (0.031) | 0.646 (0.079) | 0.08 | 1,715,958 | 0.462 (0.013) | 0.791 (0.021) | 0.574 (0.017) | 0.18 |
| Koopa | 563,692 | 0.299 (0.018) | 0.376 (0.056) | 0.397 (0.024) | 0.22 | 1,403,525 | 0.525 (0.040) | 0.758 (0.128) | 0.690 (0.079) | 0.04 | 13,306,214 | 0.485 (0.025) | 0.901 (0.064) | 0.602 (0.032) | 0.21 |
| DLinear | 204,728 | 0.319 (0.005) | 0.399 (0.005) | 0.424 (0.007) | 0.18 | 5,458 | 0.494 (0.006) | 0.682 (0.019) | 0.650 (0.007) | 0.07 | 5,301,100 | 0.437 (0.011) | 0.736 (0.015) | 0.542 (0.013) | 0.14 |
| FiLM | 408,807 | 0.367 (0.008) | 0.473 (0.008) | 0.487 (0.010) | 0.47 | 43,072 | 0.509 (0.022) | 0.724 (0.066) | 0.670 (0.029) | 0.15 | 2,652,097 | 0.753 (0.032) | 1.428 (0.038) | 0.934 (0.039) | 0.26 |
| CSDI | 244,833 | 0.423 (0.135) | 1.170 (0.634) | 0.561 (0.179) | 409.7 | 933,161 | 4.492 (1.464) | 221.583 (113.387) | 5.903 (1.924) | 13.44 | 207,873 | 0.507 (0.093) | 0.921 (0.141) | 0.629 (0.115) | 265.49 |
| US-GAN | 6,123,812 | 0.292 (0.001) | 0.352 (0.003) | 0.388 (0.001) | 0.52 | 3,913,149 | 0.603 (0.019) | 0.771 (0.040) | 0.792 (0.025) | 0.2 | 50,674,286 | 0.373 (0.003) | 0.667 (0.003) | 0.462 (0.004) | 0.82 |
| GP-VAE | 1,013,913 | 0.351 (0.007) | 0.432 (0.010) | 0.466 (0.009) | 1.4 | 130,594 | 0.571 (0.038) | 0.779 (0.134) | 0.750 (0.049) | 0.44 | 2,396,536 | 0.358 (0.013) | 0.651 (0.010) | 0.445 (0.016) | 8.21 |
| Mean | / | 0.716 | 1.099 | 0.964 | / | / | 0.598 | 1.125 | 0.785 | / | / | 0.8 | 1.425 | 0.992 | / |
| Median | / | 0.68 | 1.165 | 0.915 | / | / | 0.533 | 1.151 | 0.701 | / | / | 0.779 | 1.487 | 0.967 | / |
| LOCF | / | 0.5 | 0.835 | 0.673 | / | / | 0.614 | 1.11 | 0.807 | / | / | 0.899 | 1.939 | 1.115 | / |
| Linear | / | 0.366 | 0.608 | 0.493 | / | / | 0.481 | 0.75 | 0.632 | / | / | 0.834 | 1.944 | 1.035 | / |

| | ETT_h1 | | | | | Electricity | | | | | Pedestrian | | | | |
|---|---|---|---|---|---|---|---|---|---|---|---|---|---|---|---|
| | Size | MAE | MSE | MRE | Time | Size | MAE | MSE | MRE | Time | Size | MAE | MSE | MRE | Time |
| iTransformer | 23,723,056 | 0.636 (0.003) | 0.788 (0.006) | 0.748 (0.003) | 0.05 | 12,989,024 | 1.506 (0.161) | 4.123 (0.675) | 0.806 (0.086) | 1.11 | 2,913,304 | 0.511 (0.005) | 0.871 (0.023) | 0.675 (0.006) | 0.77 |
| SAITS | 88,235,470 | 0.507 (0.016) | 0.498 (0.038) | 0.597 (0.019) | 0.07 | 63,624,720 | 1.435 (0.025) | 4.196 (0.025) | 0.768 (0.006) | 1.15 | 133,406 | 0.492 (0.016) | 0.902 (0.065) | 0.650 (0.021) | 1.88 |
| Nonstationary | 589,927 | 0.526 (0.005) | 0.624 (0.013) | 0.619 (0.006) | 0.08 | 24,811,090 | 0.253 (0.016) | 0.259 (0.029) | 0.135 (0.008) | 0.86 | 6,338,833 | 0.563 (0.017) | 0.954 (0.067) | 0.743 (0.022) | 1.33 |
| ETSformer | 809,057 | 0.722 (0.008) | 1.001 (0.002) | 0.850 (0.009) | 0.08 | 10,518,266 | 1.404 (0.025) | 4.374 (0.044) | 0.752 (0.023) | 1.7 | 530,457 | 0.598 (0.023) | 1.095 (0.083) | 0.789 (0.031) | 3.21 |
| PatchTST | 72,247 | 0.517 (0.009) | 0.516 (0.017) | 0.609 (0.010) | 0.04 | 4,419,410 | 1.003 (0.032) | 2.472 (0.060) | 0.537 (0.017) | 4.45 | 106,905 | 0.468 (0.005) | 0.810 (0.016) | 0.617 (0.006) | 0.77 |
| Crossformer | 223,479 | 0.645 (0.117) | 0.782 (0.291) | 0.759 (0.137) | 0.08 | 9,967,314 | 1.025 (0.018) | 2.524 (0.069) | 0.549 (0.001) | 1.77 | 202,905 | 0.470 (0.007) | 0.790 (0.016) | 0.620 (0.009) | 1.52 |
| Informer | 1,058,311 | 0.621 (0.022) | 0.757 (0.067) | 0.731 (0.026) | 0.12 | 15,311,986 | 1.357 (0.007) | 4.048 (0.083) | 0.727 (0.004) | 1.38 | 446,785 | 0.492 (0.010) | 0.883 (0.038) | 0.650 (0.013) | 3.23 |
| Autoformer | 166,919 | 0.884 (0.001) | 1.398 (0.003) | 1.041 (0.001) | 0.12 | 7,431,538 | 1.905 (0.003) | 6.333 (0.010) | 1.020 (0.001) | 0.53 | 246,145 | 0.800 (0.003) | 1.548 (0.020) | 1.056 (0.004) | 2.61 |
| Pyraformer | 15,262,215 | 0.635 (0.018) | 0.797 (0.032) | 0.747 (0.021) | 0.18 | 155,940,914 | 1.305 (0.025) | 3.828 (0.088) | 0.699 (0.013) | 1.38 | 957,057 | 0.491 (0.013) | 0.880 (0.048) | 0.648 (0.017) | 2.01 |
| Transformer | 5,800,199 | 0.696 (0.034) | 0.971 (0.047) | 0.820 (0.040) | 0.04 | 155,610,482 | 1.383 (0.029) | 4.234 (0.069) | 0.741 (0.016) | 1.93 | 13,787,649 | 0.487 (0.011) | 0.887 (0.027) | 0.642 (0.014) | 1.57 |
| BRITS | 2,178,496 | 0.609 (0.003) | 0.667 (0.025) | 0.716 (0.012) | 0.73 | 17,082,800 | 1.513 (0.030) | 4.459 (0.136) | 0.810 (0.016) | 40.25 | 8,427,536 | 0.580 (0.013) | 1.015 (0.035) | 0.766 (0.017) | 17.36 |
| MRNN | 2,259 | 0.859 (0.003) | 1.321 (0.013) | 1.010 (0.004) | 0.42 | 949,749 | 1.904 (0.002) | 6.100 (0.012) | 1.019 (0.003) | 20.78 | 401,415 | 0.774 (0.001) | 1.216 (0.000) | 1.022 (0.001) | 9.88 |
| GRUD | 409,407 | 0.757 (0.056) | 0.985 (0.107) | 0.891 (0.066) | 0.37 | 9,467,304 | 1.150 (0.015) | 3.148 (0.069) | 0.615 (0.008) | 3.27 | 100,227 | 0.605 (0.006) | 0.968 (0.013) | 0.798 (0.008) | 6.22 |
| TimesNet | 5,510,663 | 0.622 (0.020) | 0.694 (0.049) | 0.732 (0.024) | 0.16 | 45,569,394 | 1.315 (0.005) | 3.876 (0.053) | 0.704 (0.003) | 1.61 | 10,816,385 | 0.533 (0.021) | 0.833 (0.013) | 0.704 (0.027) | 2.64 |
| MICN | 3,153,163 | 0.836 (0.004) | 1.261 (0.010) | 0.984 (0.004) | 0.1 | 5,457,910 | 1.517 (0.039) | 4.817 (0.039) | 0.812 (0.003) | 0.5 | / | / | / | / | / |
| SCINet | 79,493 | 0.650 (0.062) | 0.780 (0.156) | 0.765 (0.073) | 0.08 | 421,053,386 | 1.140 (0.086) | 2.824 (0.290) | 0.610 (0.046) | 1.24 | 43,783 | 0.593 (0.076) | 1.100 (0.200) | 0.783 (0.101) | 2.06 |
| StemGNN | 6,397,975 | 0.545 (0.014) | 0.553 (0.029) | 0.642 (0.017) | 0.18 | 16,863,634 | 1.294 (0.088) | 3.710 (0.368) | 0.693 (0.047) | 1.69 | 1,638,337 | 0.465 (0.009) | 0.813 (0.018) | 0.614 (0.012) | 2.88 |
| FreTS | 465,271 | 0.573 (0.003) | 0.607 (0.005) | 0.675 (0.004) | 0.06 | 3,706,194 | 1.477 (0.396) | 4.593 (2.641) | 0.791 (0.212) | 0.7 | 116,825 | 0.488 (0.010) | 0.827 (0.029) | 0.644 (0.013) | 0.8 |
| Koopa | 465,389 | 0.610 (0.072) | 0.720 (0.193) | 0.717 (0.084) | 0.02 | 2,680,114 | 1.805 (0.264) | 7.352 (1.769) | 0.966 (0.141) | 0.55 | 124,711 | 0.546 (0.082) | 0.949 (0.212) | 0.721 (0.108) | 1.12 |
| DLinear | 7,534 | 0.624 (0.010) | 0.754 (0.028) | 0.735 (0.012) | 0.03 | 2,294,692 | 0.891 (0.023) | 1.696 (0.062) | 0.477 (0.012) | 0.3 | 3,250 | 0.587 (0.020) | 0.986 (0.033) | 0.774 (0.026) | 0.83 |
| FiLM | 12,490 | 0.697 (0.009) | 0.967 (0.013) | 0.821 (0.010) | 0.12 | 570,613 | 1.049 (0.027) | 1.932 (0.078) | 0.561 (0.014) | 0.34 | 6,244 | 0.553 (0.006) | 0.897 (0.012) | 0.731 (0.008) | 1.9 |
| CSDI | 1,194,993 | 0.640 (0.053) | 0.886 (0.169) | 0.753 (0.062) | 9.86 | 43,185 | 1.832 (1.531) | 34.306 (34.776) | 0.981 (0.819) | 947.14 | 325,473 | 1.296 (0.659) | 16.656 (16.700) | 1.711 (0.871) | 108.39 |
| US-GAN | 3,807,687 | 0.623 (0.006) | 0.679 (0.015) | 0.733 (0.007) | 0.27 | 11,224,866 | 1.369 (0.024) | 3.922 (0.053) | 0.733 (0.009) | 2.23 | 14,745,617 | 0.587 (0.062) | 0.956 (0.117) | 0.775 (0.082) | 3.83 |
| GP-VAE | 384,796 | 0.762 (0.029) | 1.085 (0.087) | 0.896 (0.034) | 0.2 | 1,825,022 | 1.238 (0.027) | 3.527 (0.096) | 0.663 (0.014) | 12.46 | 284,676 | 0.757 (0.001) | 1.215 (0.000) | 1.000 (0.002) | 2.18 |
| Mean | / | 0.739 | 0.992 | 0.87 | / | / | 0.423 | 0.581 | 0.226 | / | / | 0.76 | 1.215 | 1.003 | / |
| Median | / | 0.715 | 1.052 | 0.841 | / | / | 0.408 | 0.626 | 0.219 | / | / | 0.702 | 1.341 | 0.927 | / |
| LOCF | / | 0.763 | 1.337 | 0.898 | / | / | 0.39 | 0.846 | 0.209 | / | / | 0.601 | 1.092 | 0.793 | / |
| Linear | / | 0.616 | 1.014 | 0.726 | / | / | 0.161 | 0.156 | 0.086 | / | / | 0.517 | 0.797 | 0.682 | / |

Table 14: Performance comparison in 6 datasets with 50% subsequence missing.

**BeijingAir**

| | Size | MAE | MSE | MRE | Time |
|---|---|---|---|---|---|
| iTransformer | 8,286,232 | 0.629 (0.004) | 0.902 (0.010) | 0.836 (0.006) | 0.4 |
| SAITS | 7,153,808 | 0.223 (0.005) | 0.262 (0.028) | 0.296 (0.007) | 0.23 |
| Nonstationary | 6,978,068 | 0.337 (0.002) | 0.452 (0.005) | 0.447 (0.002) | 0.4 |
| ETSformer | 7,928,510 | 0.417 (0.025) | 0.528 (0.030) | 0.555 (0.033) | 1.68 |
| PatchTST | 30,342,300 | 0.318 (0.003) | 0.404 (0.006) | 0.422 (0.004) | 4.85 |
| Crossformer | 52,933,788 | 0.274 (0.003) | 0.355 (0.005) | 0.364 (0.003) | 6.92 |
| Informer | 6,706,308 | 0.232 (0.013) | 0.294 (0.013) | 0.308 (0.018) | 0.89 |
| Autoformer | 6,700,164 | 0.704 (0.004) | 1.113 (0.006) | 0.936 (0.006) | 0.28 |
| Pyraformer | 3,230,212 | 0.227 (0.008) | 0.290 (0.009) | 0.301 (0.010) | 0.34 |
| Transformer | 203,038,852 | 0.221 (0.009) | 0.267 (0.011) | 0.293 (0.012) | 0.36 |
| BRITS | 3,598,496 | 0.193 (0.002) | 0.265 (0.004) | 0.256 (0.002) | 19.26 |
| MRNN | 96,585 | 0.682 (0.007) | 0.940 (0.018) | 0.907 (0.009) | 2.64 |
| GRID | 7,397,656 | 0.337 (0.001) | 0.434 (0.005) | 0.448 (0.001) | 1.08 |
| TimesNet | 87,063,940 | 0.307 (0.002) | 0.331 (0.004) | 0.408 (0.002) | 0.64 |
| MICN | 57,048,200 | 0.559 (0.006) | 0.758 (0.013) | 0.742 (0.008) | 0.26 |
| SCINet | 26,833,140 | 0.268 (0.010) | 0.344 (0.015) | 0.356 (0.013) | 0.3 |
| StemGNN | 2,645,628 | 0.276 (0.002) | 0.390 (0.004) | 0.366 (0.003) | 0.64 |
| FreTS | 909,852 | 0.297 (0.004) | 0.384 (0.013) | 0.395 (0.006) | 0.19 |
| Koopa | 563,692 | 0.415 (0.066) | 0.581 (0.103) | 0.551 (0.088) | 0.21 |
| DLinear | 204,728 | 0.310 (0.004) | 0.379 (0.007) | 0.412 (0.006) | 0.19 |
| FiLM | 408,807 | 0.362 (0.015) | 0.501 (0.014) | 0.481 (0.020) | 0.45 |
| CSDI | 244,833 | 0.211 (0.031) | 0.512 (0.323) | 0.280 (0.041) | 286.71 |
| US-GAN | 6,123,812 | 0.233 (0.002) | 0.269 (0.010) | 0.310 (0.003) | 0.51 |
| GP-VAE | 1,013,913 | 0.321 (0.005) | 0.343 (0.013) | 0.426 (0.007) | 1.15 |
| Mean | / | 0.711 | 1.102 | 0.971 | / |
| Median | / | 0.676 | 1.155 | 0.923 | / |
| LOCF | / | 0.482 | 0.832 | 0.658 | / |
| Linear | / | 0.358 | 0.588 | 0.488 | / |

**ETT_h1**

| | Size | MAE | MSE | MRE | Time |
|---|---|---|---|---|---|
| iTransformer | 23,723,056 | 0.810 (0.004) | 1.237 (0.011) | 0.912 (0.005) | 0.07 |
| SAITS | 88,235,470 | 0.635 (0.032) | 0.881 (0.062) | 0.715 (0.036) | 0.18 |
| Nonstationary | 589,927 | 0.612 (0.002) | 0.917 (0.003) | 0.690 (0.002) | 0.05 |
| ETSformer | 809,057 | 0.684 (0.013) | 1.002 (0.030) | 0.770 (0.015) | 0.19 |
| PatchTST | 72,247 | 0.755 (0.012) | 1.146 (0.034) | 0.850 (0.013) | 0.05 |
| Crossformer | 223,479 | 0.718 (0.011) | 1.052 (0.021) | 0.808 (0.012) | 0.08 |
| Informer | 1,058,311 | 0.571 (0.035) | 0.731 (0.096) | 0.643 (0.040) | 0.09 |
| Autoformer | 166,919 | 0.877 (0.016) | 1.421 (0.045) | 0.988 (0.018) | 0.1 |
| Pyraformer | 15,262,215 | 0.666 (0.063) | 0.924 (0.128) | 0.751 (0.071) | 0.23 |
| Transformer | 5,800,199 | 0.629 (0.039) | 0.850 (0.080) | 0.708 (0.043) | 0.11 |
| BRITS | 2,178,496 | 0.730 (0.027) | 1.065 (0.064) | 0.822 (0.030) | 1.18 |
| MRNN | 2,259 | 0.879 (0.008) | 1.472 (0.020) | 0.990 (0.009) | 0.45 |
| GRID | 409,407 | 0.760 (0.069) | 1.087 (0.150) | 0.857 (0.078) | 0.33 |
| TimesNet | 5,510,663 | 0.732 (0.014) | 1.043 (0.040) | 0.825 (0.016) | 0.17 |
| MICN | 3,153,163 | 0.851 (0.008) | 1.342 (0.021) | 0.959 (0.009) | 0.11 |
| SCINet | 79,493 | 0.711 (0.018) | 1.068 (0.041) | 0.801 (0.020) | 0.13 |
| StemGNN | 6,397,975 | 0.631 (0.029) | 0.859 (0.073) | 0.711 (0.032) | 0.26 |
| FreTS | 465,271 | 0.751 (0.014) | 1.161 (0.025) | 0.846 (0.016) | 0.06 |
| Koopa | 465,389 | 0.741 (0.098) | 1.162 (0.224) | 0.835 (0.111) | 0.03 |
| DLinear | 7,534 | 0.756 (0.010) | 1.158 (0.013) | 0.851 (0.011) | 0.04 |
| FiLM | 12,490 | 0.735 (0.007) | 1.323 (0.013) | 0.828 (0.007) | 0.09 |
| CSDI | 1,194,993 | 0.558 (0.048) | 0.783 (0.127) | 0.628 (0.054) | 9.83 |
| US-GAN | 3,807,687 | 0.670 (0.022) | 0.901 (0.059) | 0.754 (0.024) | 0.27 |
| GP-VAE | 384,796 | 0.806 (0.055) | 1.265 (0.147) | 0.908 (0.062) | 0.15 |
| Mean | / | 0.773 | 1.194 | 0.87 | / |
| Median | / | 0.784 | 1.337 | 0.883 | / |
| LOCF | / | 0.809 | 1.619 | 0.912 | / |
| Linear | / | 0.722 | 1.322 | 0.814 | / |

**Electricity**

| | Size | MAE | MSE | MRE | Time |
|---|---|---|---|---|---|
| iTransformer | 12,989,024 | 1.497 (0.066) | 4.043 (0.242) | 0.794 (0.035) | 0.91 |
| SAITS | 63,624,720 | 1.433 (0.033) | 3.842 (0.114) | 0.760 (0.018) | 1.07 |
| Nonstationary | 24,811,090 | 0.602 (0.098) | 1.252 (0.368) | 0.319 (0.052) | 0.85 |
| ETSformer | 10,518,266 | 1.081 (0.011) | 2.151 (0.015) | 0.574 (0.006) | 2.39 |
| PatchTST | 4,419,410 | 1.177 (0.123) | 3.142 (0.557) | 0.624 (0.065) | 4.63 |
| Crossformer | 9,967,314 | 1.152 (0.027) | 2.950 (0.107) | 0.611 (0.014) | 1.63 |
| Informer | 15,311,986 | 1.376 (0.011) | 3.966 (0.047) | 0.730 (0.006) | 1.23 |
| Autoformer | 7,431,538 | 1.821 (0.002) | 5.734 (0.006) | 0.966 (0.001) | 0.49 |
| Pyraformer | 15,940,914 | 1.366 (0.013) | 3.811 (0.039) | 0.724 (0.007) | 1.51 |
| Transformer | 155,610,482 | 1.431 (0.038) | 3.964 (0.034) | 0.759 (0.020) | 1.67 |
| BRITS | 17,082,800 | 1.230 (0.006) | 3.354 (0.031) | 0.653 (0.003) | 40.41 |
| MRNN | 949,749 | 1.871 (0.002) | 5.935 (0.006) | 0.993 (0.001) | 15.88 |
| GRID | 9,467,304 | 1.238 (0.021) | 3.214 (0.088) | 0.657 (0.011) | 3.73 |
| TimesNet | 45,569,394 | 1.385 (0.013) | 3.836 (0.055) | 0.734 (0.007) | 1.61 |
| MICN | 5,457,910 | 1.744 (0.010) | 5.395 (0.063) | 0.925 (0.006) | 0.44 |
| SCINet | 421,053,386 | 1.273 (0.081) | 3.422 (0.320) | 0.675 (0.043) | 1.51 |
| StemGNN | 16,863,634 | 1.504 (0.086) | 4.512 (0.420) | 0.798 (0.046) | 1.67 |
| FreTS | 3,706,194 | 1.375 (0.316) | 4.184 (1.872) | 0.729 (0.167) | 0.68 |
| Koopa | 2,680,114 | 1.848 (0.350) | 6.677 (1.972) | 0.980 (0.185) | 0.5 |
| DLinear | 2,294,692 | 1.205 (0.009) | 3.232 (0.048) | 0.639 (0.005) | 0.32 |
| FiLM | 570,613 | 1.169 (0.267) | 2.773 (1.369) | 0.620 (0.142) | 0.49 |
| CSDI | 43,185 | 0.922 (0.147) | 6.614 (4.812) | 0.489 (0.078) | 875.36 |
| US-GAN | 11,224,866 | 1.291 (0.011) | 3.491 (0.046) | 0.685 (0.006) | 2.23 |
| GP-VAE | 1,825,022 | 1.242 (0.030) | 3.624 (0.141) | 0.659 (0.016) | 19.88 |
| Mean | / | 0.44 | 0.607 | 0.234 | / |
| Median | / | 0.411 | 0.577 | 0.218 | / |
| LOCF | / | 0.797 | 1.982 | 0.423 | / |
| Linear | / | 0.409 | 0.651 | 0.217 | / |

**PeMS**

| | Size | MAE | MSE | MRE | Time |
|---|---|---|---|---|---|
| iTransformer | 1,854,744 | 0.643 (0.006) | 1.392 (0.007) | 0.760 (0.007) | 0.56 |
| SAITS | 78,229,072 | 0.346 (0.002) | 0.772 (0.001) | 0.409 (0.001) | 0.2 |
| Nonstationary | 346,318 | 0.816 (0.011) | 1.727 (0.022) | 0.965 (0.014) | 0.25 |
| ETSformer | 5,962,188 | 0.564 (0.096) | 1.034 (0.186) | 0.667 (0.114) | 0.76 |
| PatchTST | 3,045,238 | 0.417 (0.016) | 0.839 (0.035) | 0.493 (0.019) | 0.21 |
| Crossformer | 12,645,238 | 0.425 (0.005) | 0.853 (0.008) | 0.502 (0.006) | 0.29 |
| Informer | 13,149,022 | 0.369 (0.002) | 0.786 (0.003) | 0.436 (0.003) | 0.43 |
| Autoformer | 608,926 | 0.646 (0.057) | 1.377 (0.132) | 0.763 (0.068) | 0.18 |
| Pyraformer | 4,048,606 | 0.352 (0.002) | 0.764 (0.005) | 0.416 (0.002) | 0.21 |
| Transformer | 23,135,326 | 0.367 (0.002) | 0.776 (0.004) | 0.434 (0.002) | 0.12 |
| BRITS | 32,012,048 | 0.334 (0.001) | 0.747 (0.002) | 0.394 (0.001) | 5.45 |
| MRNN | 3,076,301 | 0.680 (0.002) | 1.272 (0.004) | 0.804 (0.002) | 1.7 |
| GRID | 14,104,896 | 0.403 (0.002) | 0.795 (0.002) | 0.476 (0.001) | 0.81 |
| TimesNet | 91,622,238 | 0.407 (0.003) | 0.800 (0.003) | 0.481 (0.003) | 0.39 |
| MICN | 15,490,402 | 0.671 (0.009) | 1.302 (0.027) | 0.793 (0.010) | 0.17 |
| SCINet | 1,143,027,230 | 0.668 (0.051) | 1.299 (0.110) | 0.789 (0.060) | 0.4 |
| StemGNN | 2,386,294 | 0.499 (0.073) | 1.076 (0.168) | 0.590 (0.086) | 0.26 |
| FreTS | 1,715,958 | 0.496 (0.026) | 0.953 (0.045) | 0.587 (0.030) | 0.2 |
| Koopa | 13,306,214 | 0.596 (0.184) | 1.195 (0.387) | 0.705 (0.218) | 0.22 |
| DLinear | 5,301,100 | 0.459 (0.012) | 0.876 (0.020) | 0.542 (0.014) | 0.11 |
| FiLM | 2,652,097 | 0.869 (0.068) | 1.970 (0.183) | 1.027 (0.080) | 0.28 |
| CSDI | 207,873 | 0.620 (0.316) | 1.413 (0.935) | 0.732 (0.374) | 277.65 |
| US-GAN | 50,674,286 | 0.439 (0.003) | 0.839 (0.007) | 0.518 (0.004) | 0.77 |
| GP-VAE | 2,396,536 | 0.404 (0.011) | 0.800 (0.009) | 0.477 (0.013) | 7.33 |
| Mean | / | 0.849 | 1.681 | 1.003 | / |
| Median | / | 0.886 | 1.877 | 1.047 | / |
| LOCF | / | 1.203 | 2.963 | 1.422 | / |
| Linear | / | 1 | 2.489 | 1.182 | / |

**ItalyAir**

| | Size | MAE | MSE | MRE | Time |
|---|---|---|---|---|---|
| iTransformer | 18,932,236 | 0.577 (0.007) | 0.767 (0.037) | 0.739 (0.009) | 0.19 |
| SAITS | 16,628,642 | 0.386 (0.010) | 0.386 (0.014) | 0.494 (0.012) | 0.1 |
| Nonstationary | 8,441,077 | 0.389 (0.008) | 0.464 (0.015) | 0.498 (0.010) | 0.21 |
| ETSformer | 8,009,937 | 0.536 (0.010) | 0.660 (0.015) | 0.656 (0.007) | 0.34 |
| PatchTST | 5,077,145 | 0.511 (0.005) | 0.524 (0.052) | 0.585 (0.029) | 0.51 |
| Crossformer | 2,908,185 | 0.456 (0.023) | 0.369 (0.021) | 0.498 (0.015) | 0.12 |
| Informer | 10,540,045 | 0.867 (0.008) | 2.052 (0.032) | 1.111 (0.010) | 0.19 |
| Autoformer | 993,805 | 0.412 (0.016) | 0.386 (0.029) | 0.529 (0.020) | 0.24 |
| Pyraformer | 11,355,917 | 0.388 (0.011) | 0.354 (0.018) | 0.483 (0.009) | 0.11 |
| Transformer | 4,749,837 | 0.377 (0.007) | 0.354 (0.018) | 0.483 (0.009) | 0.09 |
| BRITS | 596,912 | 0.409 (0.006) | 0.396 (0.012) | 0.524 (0.008) | 1.35 |
| MRNN | 402,111 | 0.743 (0.002) | 1.440 (0.006) | 0.953 (0.003) | 0.53 |
| GRID | 112,707 | 0.594 (0.009) | 0.769 (0.024) | 0.761 (0.012) | 0.37 |
| TimesNet | 22,051,853 | 0.536 (0.023) | 0.564 (0.040) | 0.687 (0.029) | 0.11 |
| MICN | 695,569 | 0.655 (0.023) | 1.085 (0.113) | 0.840 (0.030) | 0.09 |
| SCINet | 263,517 | 0.479 (0.011) | 0.586 (0.022) | 0.614 (0.014) | 0.1 |
| StemGNN | 926,737 | 0.433 (0.018) | 0.448 (0.038) | 0.555 (0.023) | 0.19 |
| FreTS | 668,313 | 0.499 (0.016) | 0.619 (0.060) | 0.640 (0.020) | 0.11 |
| Koopa | 1,403,525 | 0.479 (0.009) | 0.615 (0.019) | 0.614 (0.012) | 0.04 |
| DLinear | 5,458 | 0.509 (0.007) | 0.719 (0.012) | 0.652 (0.009) | 0.08 |
| FiLM | 43,072 | 0.498 (0.015) | 0.645 (0.026) | 0.638 (0.019) | 0.13 |
| CSDI | 933,161 | 0.477 (0.115) | 2.273 (1.532) | 0.612 (0.147) | 12.48 |
| US-GAN | 3,913,149 | 0.446 (0.025) | 0.406 (0.031) | 0.572 (0.032) | 0.13 |
| GP-VAE | 130,594 | 0.544 (0.018) | 0.661 (0.038) | 0.697 (0.023) | 0.45 |
| Mean | / | 0.612 | 1.13 | 0.784 | / |
| Median | / | 0.549 | 1.151 | 0.704 | / |
| LOCF | / | 0.528 | 0.9 | 0.678 | / |
| Linear | / | 0.329 | 0.407 | 0.422 | / |

**Pedestrian**

| | Size | MAE | MSE | MRE | Time |
|---|---|---|---|---|---|
| iTransformer | 2,913,304 | 0.638 (0.002) | 0.808 (0.011) | 0.833 (0.002) | 0.84 |
| SAITS | 133,406 | 0.514 (0.022) | 0.674 (0.045) | 0.672 (0.029) | 1.64 |
| Nonstationary | 6,338,833 | 0.615 (0.013) | 1.186 (0.112) | 0.803 (0.017) | 1.4 |
| ETSformer | 530,457 | 0.648 (0.015) | 0.910 (0.039) | 0.847 (0.020) | 2.67 |
| PatchTST | 106,905 | 0.622 (0.016) | 0.826 (0.025) | 0.812 (0.021) | 1.06 |
| Crossformer | 202,905 | 0.614 (0.011) | 0.797 (0.030) | 0.802 (0.015) | 1.55 |
| Informer | 446,785 | 0.547 (0.011) | 0.676 (0.019) | 0.714 (0.015) | 2.4 |
| Autoformer | 246,145 | 0.877 (0.028) | 1.788 (0.172) | 1.146 (0.037) | 2.76 |
| Pyraformer | 957,057 | 0.513 (0.008) | 0.640 (0.018) | 0.670 (0.010) | 1.44 |
| Transformer | 13,787,649 | 0.526 (0.018) | 0.694 (0.035) | 0.686 (0.024) | 1.61 |
| BRITS | 8,427,536 | 0.640 (0.006) | 0.925 (0.011) | 0.836 (0.007) | 16.47 |
| MRNN | 401,415 | 0.768 (0.000) | 0.992 (3.061e-05) | 1.003 (0.001) | 5.48 |
| GRID | 100,227 | 0.695 (0.085) | 0.856 (0.103) | 0.908 (0.111) | 6.04 |
| TimesNet | 10,816,385 | 0.713 (0.052) | 0.838 (0.053) | 0.931 (0.068) | 2 |
| MICN | / | / | / | / | / |
| SCINet | 43,783 | 0.647 (0.023) | 1.052 (0.215) | 0.846 (0.030) | 2.53 |
| StemGNN | 1,638,337 | 0.561 (0.022) | 0.707 (0.026) | 0.733 (0.029) | 3.54 |
| FreTS | 116,825 | 0.630 (0.008) | 0.853 (0.039) | 0.823 (0.010) | 0.85 |
| Koopa | 124,711 | 0.617 (0.021) | 0.855 (0.060) | 0.805 (0.028) | 0.8 |
| DLinear | 3,250 | 0.660 (0.006) | 0.953 (0.017) | 0.862 (0.008) | 0.82 |
| FiLM | 6,244 | 0.590 (0.005) | 0.813 (0.022) | 0.771 (0.006) | 1.69 |
| CSDI | 325,473 | 0.693 (0.117) | 4.141 (5.302) | 0.905 (0.153) | 105.95 |
| US-GAN | 14,745,617 | 0.664 (0.059) | 0.761 (0.051) | 0.867 (0.077) | 3.81 |
| GP-VAE | 284,676 | 0.717 (0.025) | 0.987 (0.018) | 0.936 (0.033) | 2.21 |
| Mean | / | 0.768 | 0.992 | 1.003 | / |
| Median | / | 0.714 | 1.108 | 0.933 | / |
| LOCF | / | 0.738 | 1.655 | 0.964 | / |
| Linear | / | 0.551 | 0.792 | 0.72 | / |

Table 15: Performance comparison in 6 datasets with 50% block missing.

| | BeijingAir | | | | | ItalyAir | | | | | PeMS | | | | |
|---|---|---|---|---|---|---|---|---|---|---|---|---|---|---|---|
| | Size | MAE | MSE | MRE | Time | Size | MAE | MSE | MRE | Time | Size | MAE | MSE | MRE | Time |
| iTransformer | 8,286,232 | 0.418 (0.081) | 0.576 (0.112) | 0.551 (0.106) | 0.35 | 18,932,236 | 0.493 (0.006) | 0.579 (0.017) | 0.603 (0.007) | 0.33 | 1,854,744 | 0.464 (0.007) | 0.989 (0.017) | 0.555 (0.008) | 0.91 |
| SAITS | 7,153,808 | 0.212 (0.003) | 0.226 (0.002) | 0.279 (0.004) | 0.2 | 16,628,642 | 0.416 (0.012) | 0.391 (0.029) | 0.508 (0.015) | 0.29 | 78,229,072 | 0.331 (0.001) | 0.726 (0.002) | 0.397 (0.001) | 0.44 |
| Nonstationary | 6,978,068 | 0.299 (0.001) | 0.374 (0.002) | 0.393 (0.002) | 0.36 | 8,441,077 | 0.422 (0.005) | 0.485 (0.014) | 0.516 (0.006) | 0.53 | 346,318 | 0.690 (0.004) | 1.385 (0.017) | 0.826 (0.005) | 0.75 |
| ETSformer | 7,928,510 | 0.345 (0.010) | 0.421 (0.014) | 0.454 (0.013) | 1.76 | 8,009,937 | 0.559 (0.012) | 0.718 (0.021) | 0.683 (0.014) | 0.87 | 5,962,188 | 0.871 (0.588) | 2.290 (2.491) | 1.043 (0.704) | 2.07 |
| PatchTST | 30,342,300 | 0.301 (0.002) | 0.367 (0.005) | 0.396 (0.002) | 4.96 | 5,077,145 | 0.497 (0.003) | 0.556 (0.008) | 0.607 (0.004) | 0.83 | 3,045,238 | 0.394 (0.020) | 0.805 (0.047) | 0.472 (0.024) | 0.63 |
| Crossformer | 52,933,788 | 0.256 (0.004) | 0.325 (0.004) | 0.337 (0.005) | 6.89 | 2,908,185 | 0.456 (0.019) | 0.471 (0.026) | 0.557 (0.023) | 0.39 | 12,645,238 | 0.406 (0.008) | 0.811 (0.006) | 0.486 (0.009) | 0.71 |
| Informer | 6,706,308 | 0.208 (0.003) | 0.257 (0.003) | 0.274 (0.004) | 0.59 | 10,540,045 | 0.438 (0.013) | 0.447 (0.026) | 0.535 (0.016) | 1.11 | 13,149,022 | 0.351 (0.002) | 0.738 (0.006) | 0.420 (0.003) | 2.05 |
| Autoformer | 6,700,164 | 0.694 (0.001) | 1.101 (0.002) | 0.914 (0.002) | 0.25 | 993,805 | 0.953 (0.009) | 2.222 (0.034) | 1.165 (0.011) | 0.93 | 608,926 | 0.627 (0.027) | 1.320 (0.067) | 0.751 (0.033) | 0.77 |
| Pyraformer | 3,230,212 | 0.224 (0.008) | 0.275 (0.019) | 0.294 (0.011) | 0.31 | 11,355,917 | 0.464 (0.007) | 0.452 (0.007) | 0.567 (0.008) | 0.5 | 4,048,606 | 0.334 (0.002) | 0.713 (0.002) | 0.400 (0.002) | 0.57 |
| Transformer | 203,038,852 | 0.204 (0.002) | 0.230 (0.004) | 0.269 (0.003) | 0.33 | 4,749,837 | 0.404 (0.014) | 0.396 (0.041) | 0.493 (0.018) | 0.33 | 23,135,326 | 0.353 (0.003) | 0.733 (0.003) | 0.422 (0.004) | 0.37 |
| BRITS | 3,598,496 | 0.189 (0.003) | 0.244 (0.007) | 0.249 (0.003) | 24.07 | 596,912 | 0.452 (0.005) | 0.451 (0.009) | 0.552 (0.006) | 1.07 | 32,012,048 | 0.320 (0.001) | 0.697 (0.002) | 0.383 (0.001) | 6.47 |
| MRNN | 96,585 | 0.713 (0.002) | 1.014 (0.006) | 0.939 (0.003) | 3.11 | 402,111 | 0.798 (0.002) | 1.525 (0.003) | 0.975 (0.002) | 1.79 | 3,076,301 | 0.672 (0.005) | 1.239 (0.012) | 0.804 (0.005) | 6.35 |
| GRUD | 7,397,656 | 0.303 (0.004) | 0.362 (0.002) | 0.398 (0.005) | 1.07 | 112,707 | 0.604 (0.010) | 0.778 (0.031) | 0.738 (0.012) | 1.17 | 14,104,896 | 0.392 (0.002) | 0.751 (0.002) | 0.470 (0.003) | 2.5 |
| TimesNet | 87,063,940 | 0.281 (0.005) | 0.288 (0.007) | 0.370 (0.007) | 0.66 | 22,051,853 | 0.536 (0.024) | 0.582 (0.045) | 0.655 (0.029) | 0.58 | 91,622,238 | 0.391 (0.002) | 0.742 (0.004) | 0.468 (0.002) | 1.15 |
| MICN | 57,048,200 | 0.492 (0.007) | 0.632 (0.014) | 0.647 (0.009) | 0.18 | 695,569 | 0.725 (0.006) | 1.284 (0.019) | 0.885 (0.007) | 0.46 | 15,490,402 | 0.598 (0.011) | 1.120 (0.029) | 0.716 (0.014) | 0.5 |
| SCINet | 26,833,140 | 0.258 (0.004) | 0.326 (0.003) | 0.339 (0.006) | 0.32 | 263,517 | 0.503 (0.010) | 0.584 (0.021) | 0.615 (0.012) | 0.42 | 1,143,027,230 | 0.619 (0.034) | 1.218 (0.078) | 0.741 (0.040) | 0.27 |
| StemGNN | 2,645,628 | 0.242 (0.003) | 0.357 (0.003) | 0.319 (0.003) | 0.69 | 926,737 | 0.447 (0.011) | 0.457 (0.017) | 0.546 (0.013) | 0.6 | 2,386,294 | 0.513 (0.069) | 1.085 (0.179) | 0.614 (0.082) | 0.91 |
| FreTS | 909,852 | 0.264 (0.014) | 0.323 (0.015) | 0.347 (0.019) | 0.18 | 668,313 | 0.491 (0.014) | 0.541 (0.025) | 0.600 (0.017) | 0.35 | 1,715,958 | 0.452 (0.013) | 0.841 (0.017) | 0.541 (0.016) | 0.39 |
| Koopa | 563,692 | 0.309 (0.028) | 0.432 (0.054) | 0.407 (0.037) | 0.22 | 1,403,525 | 0.482 (0.020) | 0.558 (0.017) | 0.589 (0.025) | 0.11 | 13,306,214 | 0.625 (0.149) | 1.158 (0.253) | 0.748 (0.179) | 0.47 |
| DLinear | 204,728 | 0.301 (0.004) | 0.349 (0.005) | 0.396 (0.006) | 0.19 | 5,458 | 0.510 (0.009) | 0.614 (0.020) | 0.623 (0.011) | 0.37 | 5,301,100 | 0.454 (0.056) | 0.852 (0.116) | 0.544 (0.067) | 0.33 |
| FiLM | 408,807 | 0.355 (0.008) | 0.479 (0.004) | 0.468 (0.010) | 0.51 | 43,072 | 0.493 (0.008) | 0.554 (0.011) | 0.602 (0.010) | 0.67 | 2,652,097 | 0.772 (0.035) | 1.643 (0.090) | 0.924 (0.042) | 0.92 |
| CSDI | 244,833 | 0.181 (0.025) | 0.399 (0.172) | 0.238 (0.033) | 283.28 | 933,161 | 0.675 (0.358) | 7.791 (11.133) | 0.825 (0.437) | 24.11 | 207,873 | 1.132 (1.318) | 14.906 (27.389) | 1.356 (1.578) | 309.66 |
| US-GAN | 6,123,812 | 0.216 (0.001) | 0.233 (0.006) | 0.285 (0.001) | 0.52 | 3,913,149 | 0.484 (0.007) | 0.451 (0.008) | 0.592 (0.009) | 1.38 | 50,674,286 | 0.387 (0.003) | 0.739 (0.004) | 0.464 (0.004) | 8.5 |
| GP-VAE | 1,013,913 | 0.294 (0.009) | 0.307 (0.021) | 0.387 (0.011) | 1.34 | 130,594 | 0.556 (0.012) | 0.649 (0.044) | 0.680 (0.014) | 0.98 | 2,396,536 | 0.372 (0.013) | 0.738 (0.013) | 0.445 (0.016) | 21.8 |
| Mean | / | 0.714 | 1.114 | 0.966 | / | / | 0.625 | 1.163 | 0.764 | / | / | 0.829 | 1.618 | 0.993 | / |
| Median | / | 0.682 | 1.175 | 0.921 | / | / | 0.589 | 1.192 | 0.72 | / | / | 0.856 | 1.79 | 1.025 | / |
| LOCF | / | 0.4 | 0.715 | 0.541 | / | / | 0.493 | 0.736 | 0.602 | / | / | 0.92 | 2.266 | 1.101 | / |
| Linear | / | 0.285 | 0.418 | 0.386 | / | / | 0.377 | 0.478 | 0.461 | / | / | 0.716 | 1.585 | 0.857 | / |

| | ETT_h1 | | | | | Electricity | | | | | Pedestrian | | | | |
|---|---|---|---|---|---|---|---|---|---|---|---|---|---|---|---|
| | Size | MAE | MSE | MRE | Time | Size | MAE | MSE | MRE | Time | Size | MAE | MSE | MRE | Time |
| iTransformer | 23,723,056 | 0.509 (0.007) | 0.525 (0.015) | 0.633 (0.009) | 0.16 | 12,989,024 | 1.014 (0.076) | 2.383 (0.247) | 0.545 (0.041) | 0.98 | 2,913,304 | 0.638 (0.002) | 0.808 (0.011) | 0.833 (0.002) | 0.84 |
| SAITS | 88,235,470 | 0.424 (0.034) | 0.387 (0.049) | 0.527 (0.042) | 0.3 | 63,624,720 | 1.441 (0.032) | 3.900 (0.119) | 0.773 (0.017) | 1.11 | 133,406 | 0.514 (0.022) | 0.674 (0.045) | 0.672 (0.029) | 1.64 |
| Nonstationary | 589,927 | 0.510 (0.009) | 0.541 (0.019) | 0.633 (0.011) | 0.22 | 24,811,090 | 0.350 (0.103) | 0.478 (0.309) | 0.188 (0.055) | 0.87 | 6,338,833 | 0.615 (0.013) | 1.186 (0.112) | 0.803 (0.017) | 1.4 |
| ETSformer | 809,057 | 0.646 (0.012) | 0.852 (0.019) | 0.802 (0.015) | 1 | 10,518,266 | 1.065 (0.015) | 2.414 (0.050) | 0.572 (0.008) | 2.48 | 530,457 | 0.648 (0.015) | 0.910 (0.039) | 0.847 (0.020) | 2.67 |
| PatchTST | 72,247 | 0.558 (0.020) | 0.633 (0.039) | 0.694 (0.024) | 0.28 | 4,419,410 | 1.201 (0.021) | 2.996 (0.058) | 0.645 (0.011) | 4.42 | 106,905 | 0.622 (0.016) | 0.826 (0.025) | 0.812 (0.021) | 1.06 |
| Crossformer | 223,479 | 0.515 (0.018) | 0.559 (0.032) | 0.640 (0.022) | 0.29 | 9,967,314 | 1.211 (0.037) | 2.975 (0.125) | 0.650 (0.020) | 1.56 | 202,905 | 0.614 (0.011) | 0.797 (0.030) | 0.802 (0.015) | 1.55 |
| Informer | 1,058,311 | 0.460 (0.011) | 0.493 (0.023) | 0.572 (0.014) | 0.55 | 15,311,986 | 1.331 (0.017) | 3.586 (0.044) | 0.714 (0.009) | 1.02 | 446,785 | 0.547 (0.011) | 0.676 (0.019) | 0.714 (0.015) | 2.4 |
| Autoformer | 166,919 | 1.111 (0.065) | 2.026 (0.224) | 1.380 (0.081) | 0.44 | 7,431,538 | 1.757 (0.017) | 5.433 (0.041) | 0.943 (0.007) | 0.49 | 246,145 | 0.877 (0.028) | 1.788 (0.172) | 1.146 (0.037) | 2.76 |
| Pyraformer | 15,262,215 | 0.440 (0.026) | 0.427 (0.047) | 0.546 (0.033) | 0.41 | 15,940,914 | 1.290 (0.017) | 3.356 (0.059) | 0.692 (0.009) | 1.47 | 957,057 | 0.513 (0.008) | 0.640 (0.018) | 0.670 (0.010) | 1.44 |
| Transformer | 5,800,199 | 0.440 (0.019) | 0.408 (0.023) | 0.547 (0.024) | 0.18 | 155,610,482 | 1.424 (0.015) | 3.808 (0.080) | 0.764 (0.008) | 1.55 | 13,787,649 | 0.526 (0.018) | 0.694 (0.035) | 0.686 (0.024) | 1.61 |
| BRITS | 2,178,496 | 0.593 (0.020) | 0.672 (0.037) | 0.736 (0.025) | 3.3 | 17,082,800 | 1.128 (0.024) | 3.124 (0.066) | 0.606 (0.013) | 39.46 | 8,427,536 | 0.640 (0.006) | 0.925 (0.011) | 0.836 (0.007) | 16.47 |
| MRNN | 2,259 | 0.801 (0.003) | 1.214 (0.003) | 0.994 (0.003) | 1.35 | 949,749 | 1.858 (0.001) | 5.877 (0.003) | 0.998 (0.001) | 22.17 | 401,415 | 0.768 (0.000) | 0.992 (3.061e-05) | 1.003 (0.001) | 5.48 |
| GRUD | 409,407 | 0.570 (0.015) | 0.696 (0.040) | 0.708 (0.019) | 0.98 | 9,467,304 | 1.276 (0.006) | 3.402 (0.037) | 0.685 (0.003) | 3.66 | 100,227 | 0.695 (0.085) | 0.856 (0.103) | 0.908 (0.111) | 6.04 |
| TimesNet | 5,510,663 | 0.570 (0.012) | 0.647 (0.018) | 0.708 (0.015) | 0.45 | 45,569,394 | 1.496 (0.038) | 4.271 (0.106) | 0.803 (0.021) | 1.61 | 10,816,385 | 0.713 (0.052) | 0.838 (0.053) | 0.931 (0.068) | 2 |
| MICN | 3,153,163 | 0.801 (0.020) | 0.950 (0.048) | 0.877 (0.025) | 0.3 | 5,457,910 | 1.542 (0.020) | 4.476 (0.097) | 0.828 (0.011) | 0.49 | / | / | / | / | / |
| SCINet | 79,493 | 0.544 (0.017) | 0.605 (0.037) | 0.676 (0.021) | 0.27 | 421,053,386 | 1.054 (0.097) | 2.459 (0.054) | 0.566 (0.010) | 1.49 | 43,783 | 0.647 (0.023) | 1.052 (0.215) | 0.846 (0.030) | 2.53 |
| StemGNN | 6,397,975 | 0.433 (0.011) | 0.384 (0.017) | 0.538 (0.014) | 0.48 | 16,863,634 | 1.545 (0.181) | 4.636 (1.007) | 0.829 (0.097) | 1.66 | 1,638,337 | 0.561 (0.022) | 0.707 (0.026) | 0.733 (0.029) | 3.54 |
| FreTS | 465,271 | 0.496 (0.020) | 0.510 (0.037) | 0.616 (0.025) | 0.23 | 3,706,194 | 1.000 (0.012) | 2.091 (0.055) | 0.537 (0.007) | 0.54 | 116,825 | 0.630 (0.008) | 0.853 (0.039) | 0.823 (0.010) | 0.85 |
| Koopa | 465,389 | 0.556 (0.111) | 0.643 (0.280) | 0.691 (0.137) | 0.13 | 2,680,114 | 1.919 (0.520) | 8.306 (3.409) | 1.030 (0.279) | 0.64 | 124,711 | 0.617 (0.021) | 0.855 (0.060) | 0.805 (0.028) | 0.8 |
| DLinear | 7,534 | 0.567 (0.002) | 0.661 (0.010) | 0.704 (0.003) | 0.15 | 2,294,692 | 0.964 (0.011) | 2.199 (0.011) | 0.517 (0.001) | 0.31 | 3,250 | 0.660 (0.006) | 0.953 (0.017) | 0.862 (0.008) | 0.82 |
| FiLM | 12,490 | 0.646 (0.002) | 0.965 (0.010) | 0.802 (0.003) | 0.4 | 570,613 | 1.032 (0.157) | 2.067 (0.662) | 0.554 (0.084) | 0.52 | 6,244 | 0.590 (0.005) | 0.813 (0.022) | 0.771 (0.006) | 1.69 |
| CSDI | 1,194,993 | 0.401 (0.067) | 0.340 (0.100) | 0.497 (0.083) | 20.91 | 43,185 | 1.056 (0.452) | 49.497 (39.024) | 0.567 (0.243) | 948.55 | 325,473 | 0.693 (0.117) | 4.141 (5.302) | 0.905 (0.153) | 105.95 |
| US-GAN | 3,807,687 | 0.815 (0.459) | 1.549 (1.788) | 1.012 (0.570) | 1.51 | 11,224,866 | 1.251 (0.015) | 3.323 (0.043) | 0.672 (0.008) | 2.25 | 14,745,617 | 0.664 (0.059) | 0.761 (0.051) | 0.867 (0.077) | 3.81 |
| GP-VAE | 384,796 | 0.651 (0.077) | 0.812 (0.169) | 0.808 (0.096) | 0.68 | 1,825,022 | 1.308 (0.033) | 3.637 (0.071) | 0.702 (0.017) | 19.6 | 284,676 | 0.717 (0.025) | 0.987 (0.018) | 0.936 (0.033) | 2.21 |
| Mean | / | 0.72 | 0.973 | 0.894 | / | / | 0.432 | 0.588 | 0.232 | / | / | 0.768 | 0.992 | 1.003 | / |
| Median | / | 0.712 | 1.079 | 0.884 | / | / | 0.406 | 0.598 | 0.218 | / | / | 0.714 | 1.108 | 0.933 | / |
| LOCF | / | 0.721 | 1.317 | 0.896 | / | / | 0.25 | 0.381 | 0.134 | / | / | 0.738 | 1.655 | 0.964 | / |
| Linear | / | 0.527 | 0.699 | 0.654 | / | / | 0.14 | 0.105 | 0.075 | / | / | 0.551 | 0.792 | 0.72 | / |

Table 16: Performance comparison for the classification task on PhysioNet2012 and Pedestrian datasets with 10% point missing.

**PhysioNet2012 (10% missing rate)**

| | PR_AUC wt XGB | PR_AUC w XGB | PR_AUC w RNN | PR_AUC w Transformer | ROC_AUC wt XGB | ROC_AUC w XGB | ROC_AUC w RNN | ROC_AUC w Transformer |
|---|---|---|---|---|---|---|---|---|
| iTransformer | | 0.521 (0.000) | 0.359 (0.049) | 0.286 (0.040) | | 0.852 (0.000) | 0.692 (0.073) | 0.685 (0.032) |
| SAITS | | 0.490 (0.000) | 0.274 (0.054) | 0.277 (0.039) | | 0.851 (0.000) | 0.658 (0.076) | 0.668 (0.037) |
| Nonstationary | | 0.542 (0.000) | 0.294 (0.029) | 0.374 (0.061) | | 0.860 (0.000) | 0.583 (0.078) | 0.721 (0.049) |
| ETSformer | | 0.433 (0.000) | 0.392 (0.035) | 0.278 (0.013) | | 0.820 (0.000) | 0.734 (0.030) | 0.678 (0.013) |
| PatchTST | | 0.512 (0.000) | 0.347 (0.038) | 0.303 (0.067) | | 0.848 (0.000) | 0.703 (0.064) | 0.698 (0.061) |
| Crossformer | | 0.489 (0.000) | 0.317 (0.029) | 0.253 (0.032) | | 0.836 (0.000) | 0.683 (0.051) | 0.644 (0.026) |
| Informer | | 0.468 (0.000) | 0.306 (0.046) | 0.252 (0.032) | | 0.836 (0.000) | 0.701 (0.021) | 0.653 (0.029) |
| Autoformer | | 0.368 (0.000) | 0.203 (0.008) | 0.223 (0.009) | | 0.768 (0.000) | 0.597 (0.009) | 0.625 (0.008) |
| Pyraformer | | 0.461 (0.000) | 0.381 (0.017) | 0.250 (0.047) | | 0.829 (0.000) | 0.739 (0.026) | 0.654 (0.047) |
| Transformer | | 0.458 (0.000) | 0.360 (0.034) | 0.223 (0.024) | | 0.833 (0.000) | 0.723 (0.036) | 0.626 (0.036) |
| BRITS | | 0.455 (0.000) | 0.315 (0.069) | 0.270 (0.068) | | 0.841 (0.000) | 0.667 (0.066) | 0.646 (0.074) |
| MRNN | | 0.346 (0.000) | 0.232 (0.013) | 0.219 (0.012) | | 0.760 (0.000) | 0.611 (0.008) | 0.619 (0.015) |
| GRUD | | 0.423 (0.000) | 0.373 (0.064) | 0.392 (0.058) | | 0.840 (0.000) | 0.721 (0.075) | 0.745 (0.035) |
| TimesNet | 0.388 (0.000) | 0.432 (0.000) | 0.332 (0.055) | 0.344 (0.062) | 0.771 (0.000) | 0.823 (0.000) | 0.687 (0.087) | 0.710 (0.043) |
| MICN | | 0.410 (0.000) | 0.316 (0.069) | 0.295 (0.058) | | 0.824 (0.000) | 0.700 (0.071) | 0.686 (0.050) |
| SCINet | | 0.443 (0.000) | 0.319 (0.071) | 0.340 (0.052) | | 0.818 (0.000) | 0.663 (0.055) | 0.720 (0.034) |
| StemGNN | | 0.423 (0.000) | 0.277 (0.074) | 0.312 (0.024) | | 0.809 (0.000) | 0.534 (0.124) | 0.701 (0.025) |
| FreTS | | 0.471 (0.000) | 0.333 (0.025) | 0.275 (0.012) | | 0.840 (0.000) | 0.693 (0.035) | 0.674 (0.018) |
| Koopa | | 0.470 (0.000) | 0.237 (0.056) | 0.238 (0.039) | | 0.835 (0.000) | 0.509 (0.113) | 0.628 (0.036) |
| DLinear | | 0.486 (0.000) | 0.323 (0.057) | 0.230 (0.034) | | 0.851 (0.000) | 0.689 (0.038) | 0.611 (0.053) |
| FILM | | 0.422 (0.000) | 0.292 (0.058) | 0.361 (0.066) | | 0.825 (0.000) | 0.565 (0.095) | 0.706 (0.046) |
| CSDI | | 0.459 (0.000) | 0.291 (0.056) | 0.438 (0.016) | | 0.853 (0.000) | 0.553 (0.108) | 0.762 (0.019) |
| US-GAN | | 0.492 (0.000) | 0.333 (0.044) | 0.223 (0.046) | | 0.839 (0.000) | 0.708 (0.019) | 0.610 (0.043) |
| GP-VAE | | 0.456 (0.000) | 0.396 (0.036) | 0.334 (0.072) | | 0.816 (0.000) | 0.745 (0.056) | 0.701 (0.041) |
| Mean | | 0.384 (0.000) | 0.251 (0.022) | 0.200 (0.016) | | 0.763 (0.000) | 0.620 (0.017) | 0.598 (0.025) |
| Median | | 0.361 (0.023) | 0.236 (0.024) | 0.206 (0.015) | | 0.772 (0.010) | 0.619 (0.013) | 0.605 (0.025) |
| LOCF | | 0.391 (0.045) | 0.267 (0.073) | 0.277 (0.102) | | 0.795 (0.033) | 0.634 (0.074) | 0.644 (0.061) |
| Linear | | 0.417 (0.060) | 0.289 (0.078) | 0.309 (0.111) | | 0.807 (0.036) | 0.640 (0.076) | 0.669 (0.069) |

**Pedestrian (10% missing rate)**

| | PR_AUC wt XGB | PR_AUC w XGB | PR_AUC w RNN | PR_AUC w Transformer | ROC_AUC wt XGB | ROC_AUC w XGB | ROC_AUC w RNN | ROC_AUC w Transformer |
|---|---|---|---|---|---|---|---|---|
| iTransformer | | 0.977 (0.000) | 0.478 (0.079) | 0.903 (0.011) | | 0.996 (0.000) | 0.896 (0.026) | 0.985 (0.001) |
| SAITS | | 0.982 (0.000) | 0.458 (0.050) | 0.926 (0.024) | | 0.997 (0.000) | 0.890 (0.020) | 0.989 (0.003) |
| Nonstationary | | 0.973 (0.000) | 0.485 (0.060) | 0.956 (0.000) | | 0.996 (0.000) | 0.902 (0.020) | 0.993 (0.000) |
| ETSformer | | 0.977 (0.000) | 0.487 (0.057) | 0.910 (0.035) | | 0.996 (0.000) | 0.901 (0.020) | 0.986 (0.005) |
| PatchTST | | 0.981 (0.000) | 0.463 (0.058) | 0.915 (0.029) | | 0.997 (0.000) | 0.892 (0.019) | 0.988 (0.004) |
| Crossformer | | 0.980 (0.000) | 0.509 (0.054) | 0.923 (0.036) | | 0.997 (0.000) | 0.907 (0.016) | 0.989 (0.005) |
| Informer | | 0.980 (0.000) | 0.479 (0.049) | 0.908 (0.026) | | 0.997 (0.000) | 0.898 (0.015) | 0.986 (0.004) |
| Autoformer | | 0.978 (0.000) | 0.480 (0.086) | 0.932 (0.027) | | 0.996 (0.000) | 0.898 (0.025) | 0.989 (0.004) |
| Pyraformer | | 0.978 (0.000) | 0.506 (0.064) | 0.939 (0.022) | | 0.997 (0.000) | 0.908 (0.018) | 0.991 (0.003) |
| Transformer | | 0.979 (0.000) | 0.496 (0.051) | 0.897 (0.019) | | 0.997 (0.000) | 0.902 (0.015) | 0.985 (0.003) |
| BRITS | | 0.981 (0.000) | 0.462 (0.051) | 0.938 (0.027) | | 0.997 (0.000) | 0.898 (0.017) | 0.991 (0.004) |
| MRNN | | 0.970 (0.000) | 0.474 (0.051) | 0.916 (0.011) | | 0.995 (0.000) | 0.900 (0.017) | 0.987 (0.002) |
| GRUD | | 0.977 (0.000) | 0.493 (0.058) | 0.916 (0.025) | | 0.997 (0.000) | 0.899 (0.019) | 0.988 (0.003) |
| TimesNet | 0.972 (0.000) | 0.978 (0.000) | 0.503 (0.064) | 0.910 (0.034) | 0.996 (0.000) | 0.997 (0.000) | 0.906 (0.021) | 0.987 (0.005) |
| MICN | | | | | | | | |
| SCINet | | 0.978 (0.000) | 0.502 (0.051) | 0.902 (0.022) | | 0.997 (0.000) | 0.904 (0.017) | 0.986 (0.003) |
| StemGNN | | 0.981 (0.000) | 0.483 (0.034) | 0.928 (0.038) | | 0.997 (0.000) | 0.899 (0.009) | 0.989 (0.005) |
| FreTS | | 0.982 (0.000) | 0.474 (0.063) | 0.902 (0.028) | | 0.997 (0.000) | 0.895 (0.023) | 0.985 (0.004) |
| Koopa | | 0.978 (0.000) | 0.498 (0.047) | 0.907 (0.050) | | 0.997 (0.000) | 0.906 (0.014) | 0.986 (0.006) |
| DLinear | | 0.981 (0.000) | 0.520 (0.054) | 0.904 (0.029) | | 0.997 (0.000) | 0.912 (0.016) | 0.986 (0.004) |
| FILM | | 0.973 (0.000) | 0.482 (0.020) | 0.955 (0.007) | | 0.996 (0.000) | 0.903 (0.007) | 0.992 (0.001) |
| CSDI | | 0.978 (0.000) | 0.511 (0.048) | 0.924 (0.029) | | 0.996 (0.000) | 0.907 (0.019) | 0.989 (0.004) |
| US-GAN | | 0.979 (0.000) | 0.498 (0.045) | 0.941 (0.011) | | 0.997 (0.000) | 0.905 (0.014) | 0.991 (0.002) |
| GP-VAE | | 0.974 (0.000) | 0.472 (0.065) | 0.898 (0.003) | | 0.996 (0.000) | 0.893 (0.020) | 0.985 (0.001) |
| Mean | | 0.970 (0.000) | 0.486 (0.050) | 0.912 (0.018) | | 0.995 (0.000) | 0.904 (0.019) | 0.986 (0.003) |
| Median | | 0.968 (0.002) | 0.498 (0.068) | 0.900 (0.019) | | 0.995 (0.000) | 0.906 (0.023) | 0.984 (0.003) |
| LOCF | | 0.971 (0.004) | 0.489 (0.063) | 0.909 (0.023) | | 0.996 (0.001) | 0.903 (0.021) | 0.986 (0.004) |
| Linear | | 0.973 (0.005) | 0.498 (0.059) | 0.916 (0.025) | | 0.996 (0.001) | 0.906 (0.019) | 0.987 (0.004) |

Table 17: Performance comparison for the classification task on Pedestrian datasets with 50% point missing, 90% point missing, and 50% subsequence missing.

**Pedestrian (50% point missing rate)**

| | PR_AUC wt XGB | PR_AUC w XGB | PR_AUC w RNN | PR_AUC w Transformer | ROC_AUC wt XGB | ROC_AUC w XGB | ROC_AUC w RNN | ROC_AUC w Transformer |
|---|---|---|---|---|---|---|---|---|
| iTransformer | | 0.936 (0.000) | 0.495 (0.070) | 0.848 (0.011) | | 0.990 (0.000) | 0.901 (0.022) | 0.972 (0.003) |
| SAITS | | 0.944 (0.000) | 0.486 (0.051) | 0.823 (0.005) | | 0.991 (0.000) | 0.900 (0.017) | 0.970 (0.001) |
| Nonstationary | | 0.900 (0.000) | 0.286 (0.097) | 0.803 (0.016) | | 0.984 (0.000) | 0.790 (0.074) | 0.968 (0.002) |
| ETSformer | | 0.900 (0.000) | 0.563 (0.038) | 0.782 (0.034) | | 0.984 (0.000) | 0.925 (0.010) | 0.964 (0.006) |
| PatchTST | | 0.948 (0.000) | 0.504 (0.077) | 0.839 (0.010) | | 0.991 (0.000) | 0.904 (0.023) | 0.974 (0.001) |
| Crossformer | | 0.942 (0.000) | 0.518 (0.080) | 0.850 (0.021) | | 0.991 (0.000) | 0.912 (0.025) | 0.975 (0.003) |
| Informer | | 0.941 (0.000) | 0.496 (0.034) | 0.824 (0.005) | | 0.991 (0.000) | 0.905 (0.010) | 0.969 (0.003) |
| Autoformer | | 0.855 (0.000) | 0.528 (0.021) | 0.725 (0.030) | | 0.973 (0.000) | 0.917 (0.005) | 0.946 (0.002) |
| Pyraformer | | 0.941 (0.000) | 0.498 (0.073) | 0.850 (0.021) | | 0.991 (0.000) | 0.902 (0.020) | 0.973 (0.004) |
| Transformer | | 0.951 (0.000) | 0.483 (0.059) | 0.836 (0.009) | | 0.992 (0.000) | 0.897 (0.017) | 0.971 (0.003) |
| BRITS | | 0.929 (0.000) | 0.470 (0.073) | 0.846 (0.015) | | 0.989 (0.000) | 0.895 (0.027) | 0.974 (0.003) |
| MRNN | | 0.884 (0.000) | 0.479 (0.130) | 0.694 (0.018) | | 0.981 (0.000) | 0.885 (0.070) | 0.945 (0.003) |
| GRUD | | 0.932 (0.000) | 0.579 (0.030) | 0.817 (0.017) | | 0.989 (0.000) | 0.928 (0.005) | 0.969 (0.003) |
| TimesNet | 0.912 (0.000) | | | | 0.986 (0.000) | | | |
| MICN | | 0.897 (0.000) | 0.460 (0.038) | 0.835 (0.011) | | 0.983 (0.000) | 0.894 (0.016) | 0.970 (0.003) |
| SCINet | | 0.924 (0.000) | 0.497 (0.064) | 0.856 (0.008) | | 0.988 (0.000) | 0.902 (0.021) | 0.975 (0.002) |
| StemGNN | | 0.943 (0.000) | 0.495 (0.086) | 0.854 (0.025) | | 0.991 (0.000) | 0.897 (0.024) | 0.976 (0.004) |
| FreTS | | 0.935 (0.000) | 0.474 (0.079) | 0.829 (0.020) | | 0.990 (0.000) | 0.896 (0.026) | 0.972 (0.004) |
| Koopa | | 0.918 (0.000) | 0.491 (0.069) | 0.859 (0.016) | | 0.987 (0.000) | 0.900 (0.021) | 0.975 (0.002) |
| DLinear | | 0.882 (0.000) | 0.550 (0.025) | 0.798 (0.016) | | 0.981 (0.000) | 0.918 (0.008) | 0.967 (0.003) |
| FILM | | 0.894 (0.000) | 0.545 (0.048) | 0.772 (0.019) | | 0.983 (0.000) | 0.918 (0.011) | 0.961 (0.004) |
| CSDI | | 0.921 (0.000) | 0.409 (0.029) | 0.857 (0.007) | | 0.987 (0.000) | 0.869 (0.019) | 0.975 (0.002) |
| US-GAN | | 0.924 (0.000) | 0.466 (0.047) | 0.826 (0.018) | | 0.988 (0.000) | 0.896 (0.014) | 0.970 (0.003) |
| GP-VAE | | 0.877 (0.000) | 0.565 (0.020) | 0.726 (0.021) | | 0.980 (0.000) | 0.928 (0.005) | 0.955 (0.003) |
| Mean | | 0.884 (0.000) | 0.500 (0.035) | 0.692 (0.005) | | 0.981 (0.000) | 0.910 (0.008) | 0.944 (0.002) |
| Median | | 0.881 (0.004) | 0.481 (0.089) | 0.717 (0.029) | | 0.981 (0.001) | 0.896 (0.048) | 0.948 (0.005) |
| LOCF | | 0.895 (0.020) | 0.511 (0.086) | 0.750 (0.052) | | 0.983 (0.003) | 0.906 (0.042) | 0.955 (0.011) |
| Linear | | 0.904 (0.024) | 0.525 (0.081) | 0.768 (0.056) | | 0.984 (0.004) | 0.911 (0.038) | 0.959 (0.012) |

**Pedestrian (90% point missing rate)**

| | PR_AUC wt XGB | PR_AUC w XGB | PR_AUC w RNN | PR_AUC w Transformer | ROC_AUC wt XGB | ROC_AUC w XGB | ROC_AUC w RNN | ROC_AUC w Transformer |
|---|---|---|---|---|---|---|---|---|
| iTransformer | | 0.392 (0.000) | 0.254 (0.009) | 0.332 (0.005) | | 0.845 (0.000) | 0.784 (0.020) | 0.835 (0.002) |
| SAITS | | 0.386 (0.000) | 0.242 (0.027) | 0.348 (0.011) | | 0.843 (0.000) | 0.750 (0.045) | 0.826 (0.003) |
| Nonstationary | | 0.366 (0.000) | 0.179 (0.016) | 0.310 (0.019) | | 0.828 (0.000) | 0.659 (0.046) | 0.800 (0.005) |
| ETSformer | | 0.329 (0.000) | 0.232 (0.013) | 0.340 (0.017) | | 0.809 (0.000) | 0.756 (0.024) | 0.834 (0.004) |
| PatchTST | | 0.391 (0.000) | 0.197 (0.029) | 0.383 (0.020) | | 0.846 (0.000) | 0.680 (0.065) | 0.846 (0.004) |
| Crossformer | | 0.389 (0.000) | 0.263 (0.013) | 0.387 (0.006) | | 0.848 (0.000) | 0.797 (0.003) | 0.855 (0.004) |
| Informer | | 0.361 (0.000) | 0.228 (0.010) | 0.319 (0.011) | | 0.833 (0.000) | 0.739 (0.032) | 0.809 (0.002) |
| Autoformer | | 0.327 (0.000) | 0.234 (0.007) | 0.265 (0.005) | | 0.801 (0.000) | 0.760 (0.007) | 0.796 (0.008) |
| Pyraformer | | 0.405 (0.000) | 0.266 (0.009) | 0.349 (0.021) | | 0.847 (0.000) | 0.784 (0.022) | 0.837 (0.003) |
| Transformer | | 0.400 (0.000) | 0.237 (0.046) | 0.347 (0.019) | | 0.842 (0.000) | 0.710 (0.092) | 0.819 (0.006) |
| BRITS | | 0.408 (0.000) | 0.259 (0.011) | 0.371 (0.019) | | 0.848 (0.000) | 0.786 (0.009) | 0.846 (0.007) |
| MRNN | | 0.434 (0.000) | 0.259 (0.007) | 0.339 (0.015) | | 0.841 (0.000) | 0.778 (0.006) | 0.840 (0.004) |
| GRUD | | 0.433 (0.000) | 0.312 (0.005) | 0.351 (0.009) | | 0.859 (0.000) | 0.828 (0.002) | 0.839 (0.002) |
| TimesNet | 0.452 (0.000) | | | | 0.851 (0.000) | | | |
| MICN | | 0.330 (0.000) | 0.244 (0.006) | 0.324 (0.018) | | 0.811 (0.000) | 0.776 (0.004) | 0.820 (0.004) |
| SCINet | | 0.380 (0.000) | 0.210 (0.052) | 0.326 (0.011) | | 0.846 (0.000) | 0.687 (0.111) | 0.818 (0.005) |
| StemGNN | | 0.394 (0.000) | 0.193 (0.018) | 0.357 (0.020) | | 0.846 (0.000) | 0.693 (0.052) | 0.836 (0.004) |
| FreTS | | 0.396 (0.000) | 0.226 (0.032) | 0.367 (0.030) | | 0.848 (0.000) | 0.717 (0.076) | 0.846 (0.001) |
| Koopa | | 0.376 (0.000) | 0.238 (0.024) | 0.383 (0.013) | | 0.836 (0.000) | 0.753 (0.026) | 0.855 (0.004) |
| DLinear | | 0.346 (0.000) | 0.213 (0.069) | 0.333 (0.021) | | 0.829 (0.000) | 0.695 (0.168) | 0.834 (0.011) |
| FILM | | 0.323 (0.000) | 0.257 (0.006) | 0.316 (0.018) | | 0.818 (0.000) | 0.792 (0.005) | 0.806 (0.015) |
| CSDI | | 0.260 (0.000) | 0.239 (0.002) | 0.265 (0.019) | | 0.770 (0.000) | 0.772 (0.003) | 0.780 (0.013) |
| US-GAN | | 0.392 (0.000) | 0.240 (0.039) | 0.369 (0.017) | | 0.845 (0.000) | 0.768 (0.056) | 0.842 (0.004) |
| GP-VAE | | 0.420 (0.000) | 0.237 (0.018) | 0.300 (0.033) | | 0.847 (0.000) | 0.749 (0.023) | 0.820 (0.021) |
| Mean | | 0.413 (0.000) | 0.217 (0.009) | 0.291 (0.033) | | 0.830 (0.000) | 0.728 (0.012) | 0.814 (0.025) |
| Median | | 0.414 (0.002) | 0.233 (0.007) | 0.288 (0.027) | | 0.830 (0.000) | 0.728 (0.009) | 0.795 (0.027) |
| LOCF | | 0.432 (0.025) | 0.246 (0.041) | 0.303 (0.033) | | 0.841 (0.016) | 0.758 (0.042) | 0.808 (0.029) |
| Linear | | 0.425 (0.025) | 0.255 (0.039) | 0.310 (0.032) | | 0.841 (0.014) | 0.769 (0.041) | 0.809 (0.025) |

**Pedestrian (50% subsequence missing rate)**

| | PR_AUC wt XGB | PR_AUC w XGB | PR_AUC w RNN | PR_AUC w Transformer | ROC_AUC wt XGB | ROC_AUC w XGB | ROC_AUC w RNN | ROC_AUC w Transformer |
|---|---|---|---|---|---|---|---|---|
| iTransformer | | 0.750 (0.000) | 0.201 (0.034) | 0.553 (0.007) | | 0.952 (0.000) | 0.676 (0.053) | 0.920 (0.002) |
| SAITS | | 0.766 (0.000) | 0.251 (0.036) | 0.584 (0.011) | | 0.955 (0.000) | 0.715 (0.131) | 0.927 (0.004) |
| Nonstationary | | 0.729 (0.000) | 0.217 (0.037) | 0.499 (0.011) | | 0.946 (0.000) | 0.733 (0.054) | 0.911 (0.004) |
| ETSformer | | 0.743 (0.000) | 0.223 (0.043) | 0.544 (0.022) | | 0.950 (0.000) | 0.715 (0.060) | 0.920 (0.003) |
| PatchTST | | 0.747 (0.000) | 0.213 (0.036) | 0.532 (0.022) | | 0.952 (0.000) | 0.652 (0.099) | 0.917 (0.004) |
| Crossformer | | 0.745 (0.000) | 0.187 (0.029) | 0.538 (0.011) | | 0.951 (0.000) | 0.613 (0.082) | 0.920 (0.001) |
| Informer | | 0.755 (0.000) | 0.236 (0.039) | 0.557 (0.030) | | 0.953 (0.000) | 0.737 (0.056) | 0.926 (0.006) |
| Autoformer | | 0.755 (0.000) | 0.203 (0.094) | 0.562 (0.013) | | 0.953 (0.000) | 0.627 (0.198) | 0.924 (0.002) |
| Pyraformer | | 0.741 (0.000) | 0.238 (0.067) | 0.582 (0.028) | | 0.950 (0.000) | 0.721 (0.079) | 0.927 (0.005) |
| Transformer | | 0.762 (0.000) | 0.255 (0.033) | 0.519 (0.037) | | 0.954 (0.000) | 0.758 (0.048) | 0.915 (0.009) |
| BRITS | | 0.758 (0.000) | 0.183 (0.035) | 0.574 (0.018) | | 0.952 (0.000) | 0.627 (0.091) | 0.925 (0.005) |
| MRNN | | 0.728 (0.000) | 0.232 (0.075) | 0.485 (0.033) | | 0.947 (0.000) | 0.694 (0.115) | 0.902 (0.005) |
| GRUD | | 0.733 (0.000) | 0.271 (0.082) | 0.538 (0.021) | | 0.949 (0.000) | 0.756 (0.089) | 0.913 (0.004) |
| TimesNet | 0.741 (0.000) | | | | 0.949 (0.000) | | | |
| MICN | | 0.760 (0.000) | 0.305 (0.072) | 0.555 (0.025) | | 0.954 (0.000) | 0.808 (0.059) | 0.921 (0.006) |
| SCINet | | | | | | | | |
| StemGNN | | 0.754 (0.000) | 0.211 (0.040) | 0.565 (0.019) | | 0.952 (0.000) | 0.658 (0.098) | 0.927 (0.004) |
| FreTS | | 0.752 (0.000) | 0.180 (0.027) | 0.539 (0.024) | | 0.953 (0.000) | 0.582 (0.053) | 0.919 (0.004) |
| Koopa | | 0.743 (0.000) | 0.172 (0.052) | 0.572 (0.030) | | 0.951 (0.000) | 0.586 (0.107) | 0.923 (0.003) |
| DLinear | | 0.755 (0.000) | 0.202 (0.035) | 0.559 (0.022) | | 0.953 (0.000) | 0.644 (0.081) | 0.921 (0.005) |
| FILM | | 0.730 (0.000) | 0.296 (0.039) | 0.530 (0.027) | | 0.948 (0.000) | 0.815 (0.037) | 0.917 (0.006) |
| CSDI | | 0.732 (0.000) | 0.273 (0.083) | 0.573 (0.025) | | 0.948 (0.000) | 0.756 (0.091) | 0.927 (0.006) |
| US-GAN | | 0.743 (0.000) | 0.295 (0.091) | 0.530 (0.025) | | 0.950 (0.000) | 0.783 (0.103) | 0.913 (0.005) |
| GP-VAE | | 0.734 (0.000) | 0.258 (0.034) | 0.534 (0.019) | | 0.948 (0.000) | 0.779 (0.040) | 0.915 (0.003) |
| Mean | | 0.724 (0.000) | 0.215 (0.019) | 0.475 (0.019) | | 0.945 (0.000) | 0.642 (0.167) | 0.900 (0.005) |
| Median | | 0.714 (0.010) | 0.198 (0.076) | 0.472 (0.023) | | 0.944 (0.002) | 0.622 (0.142) | 0.897 (0.007) |
| LOCF | | 0.717 (0.009) | 0.209 (0.069) | 0.474 (0.026) | | 0.944 (0.002) | 0.664 (0.137) | 0.900 (0.007) |
| Linear | | 0.727 (0.019) | 0.225 (0.070) | 0.487 (0.034) | | 0.946 (0.004) | 0.696 (0.133) | 0.903 (0.009) |

Table 18: Performance comparison for the regression task on ETT_h1 datasets with 50% block missing and 50% subsequence missing.

**ETT_h1 (block_50%)**

| | MAE wt XGB | MRE wt XGB | MSE wt XGB | MAE w XGB | MRE w XGB | MSE w XGB | MAE w RNN | MRE w RNN | MSE w RNN | MAE w Transformer | MRE w Transformer | MSE w Transformer |
|---|---|---|---|---|---|---|---|---|---|---|---|---|
| iTransformer | | | | 1.208 (0.000) | 1.067 (0.000) | 1.766 (0.000) | 1.422 (0.075) | 1.256 (0.067) | 2.387 (0.199) | 1.399 (0.080) | 1.235 (0.070) | 2.254 (0.234) |
| SAITS | | | | 1.168 (0.000) | 1.032 (0.000) | 1.625 (0.000) | 1.363 (0.072) | 1.203 (0.063) | 2.223 (0.184) | 1.385 (0.055) | 1.223 (0.050) | 2.200 (0.155) |
| Nonstationary | | | | 1.189 (0.000) | 1.050 (0.000) | 1.712 (0.000) | 1.438 (0.059) | 1.270 (0.052) | 2.459 (0.152) | 1.368 (0.048) | 1.208 (0.048) | 2.195 (0.142) |
| ETSformer | | | | 1.004 (0.000) | 0.887 (0.000) | 1.357 (0.000) | 1.285 (0.064) | 1.135 (0.056) | 1.997 (0.139) | 1.327 (0.096) | 1.172 (0.084) | 2.083 (0.247) |
| PatchTST | | | | 1.183 (0.000) | 1.044 (0.000) | 1.744 (0.000) | 1.362 (0.078) | 1.203 (0.069) | 2.224 (0.191) | 1.340 (0.074) | 1.183 (0.066) | 2.086 (0.185) |
| Crossformer | | | | 1.149 (0.000) | 1.015 (0.000) | 1.654 (0.000) | 1.335 (0.070) | 1.179 (0.062) | 2.143 (0.163) | 1.285 (0.081) | 1.135 (0.072) | 1.942 (0.192) |
| Informer | | | | 1.158 (0.000) | 1.022 (0.000) | 1.702 (0.000) | 1.333 (0.080) | 1.177 (0.070) | 2.141 (0.202) | 1.351 (0.055) | 1.193 (0.049) | 2.111 (0.155) |
| Autoformer | | | | 1.327 (0.000) | 1.172 (0.000) | 2.180 (0.000) | 1.267 (0.047) | 1.119 (0.042) | 1.819 (0.136) | 1.363 (0.204) | 1.203 (0.180) | 2.218 (0.582) |
| Pyraformer | | | | 1.071 (0.000) | 0.945 (0.000) | 1.357 (0.000) | 1.328 (0.069) | 1.173 (0.061) | 2.115 (0.169) | 1.330 (0.053) | 1.174 (0.047) | 2.046 (0.138) |
| Transformer | | | | 1.128 (0.000) | 0.996 (0.000) | 1.544 (0.000) | 1.296 (0.064) | 1.144 (0.057) | 2.031 (0.140) | 1.277 (0.077) | 1.128 (0.068) | 1.917 (0.172) |
| BRITS | | | | 0.826 (0.000) | 0.729 (0.000) | 0.917 (0.000) | 1.024 (0.010) | 0.905 (0.009) | 1.337 (0.032) | 1.000 (0.057) | 0.883 (0.050) | 1.240 (0.137) |
| MRNN | | | | 0.870 (0.000) | 0.768 (0.000) | 0.999 (0.000) | 1.274 (0.078) | 1.125 (0.069) | 1.975 (0.207) | 1.387 (0.053) | 1.225 (0.047) | 2.242 (0.140) |
| GRUD | 1.379 (0.000) | 1.218 (0.000) | 2.303 (0.000) | 1.043 (0.000) | 0.921 (0.000) | 1.343 (0.000) | 1.286 (0.046) | 1.135 (0.041) | 1.996 (0.081) | 1.211 (0.080) | 1.069 (0.070) | 1.744 (0.190) |
| TimesNet | | | | 1.064 (0.000) | 0.939 (0.000) | 1.451 (0.000) | 1.316 (0.068) | 1.162 (0.060) | 2.082 (0.152) | 1.294 (0.101) | 1.143 (0.089) | 1.989 (0.244) |
| MICN | | | | 1.145 (0.000) | 1.011 (0.000) | 1.585 (0.000) | 1.338 (0.066) | 1.181 (0.058) | 2.138 (0.153) | 1.344 (0.073) | 1.187 (0.064) | 2.122 (0.187) |
| SCINet | | | | 1.182 (0.000) | 1.044 (0.000) | 1.762 (0.000) | 1.361 (0.075) | 1.202 (0.066) | 2.218 (0.180) | 1.323 (0.065) | 1.168 (0.057) | 2.038 (0.162) |
| StemGNN | | | | 1.136 (0.000) | 1.003 (0.000) | 1.599 (0.000) | 1.373 (0.064) | 1.212 (0.057) | 2.248 (0.159) | 1.323 (0.052) | 1.168 (0.046) | 2.034 (0.129) |
| FreTS | | | | 1.163 (0.000) | 1.027 (0.000) | 1.703 (0.000) | 1.356 (0.080) | 1.197 (0.070) | 2.214 (0.205) | 1.355 (0.068) | 1.196 (0.060) | 2.124 (0.194) |
| Koopa | | | | 1.274 (0.000) | 1.125 (0.000) | 2.034 (0.000) | 1.347 (0.063) | 1.189 (0.056) | 2.180 (0.154) | 1.302 (0.052) | 1.150 (0.046) | 1.983 (0.128) |
| DLinear | | | | 1.166 (0.000) | 1.029 (0.000) | 1.642 (0.000) | 1.368 (0.064) | 1.208 (0.056) | 2.222 (0.143) | 1.359 (0.114) | 1.200 (0.101) | 2.170 (0.292) |
| FILM | | | | 1.301 (0.000) | 1.149 (0.000) | 2.046 (0.000) | 1.391 (0.057) | 1.228 (0.053) | 2.307 (0.144) | 1.368 (0.060) | 1.208 (0.053) | 2.177 (0.165) |
| CSDI | | | | 1.202 (0.000) | 1.062 (0.000) | 1.728 (0.000) | 1.355 (0.048) | 1.197 (0.043) | 2.197 (0.114) | 1.362 (0.054) | 1.203 (0.048) | 2.162 (0.136) |
| US-GAN | | | | 0.951 (0.000) | 0.840 (0.000) | 1.151 (0.000) | 1.174 (0.069) | 1.036 (0.061) | 1.731 (0.147) | 1.098 (0.145) | 0.969 (0.128) | 1.492 (0.340) |
| GP-VAE | | | | 0.996 (0.000) | 0.880 (0.000) | 1.260 (0.000) | 1.177 (0.061) | 1.039 (0.054) | 1.737 (0.121) | 1.184 (0.077) | 1.046 (0.068) | 1.663 (0.192) |
| Mean | | | | 1.413 (0.000) | 1.248 (0.000) | 2.346 (0.000) | 1.669 (0.083) | 1.473 (0.074) | 3.141 (0.266) | 1.750 (0.043) | 1.545 (0.038) | 3.453 (0.173) |
| Median | | | | 1.496 (0.083) | 1.321 (0.073) | 2.648 (0.302) | 1.688 (0.100) | 1.490 (0.088) | 3.228 (0.332) | 1.736 (0.061) | 1.533 (0.054) | 3.385 (0.248) |
| LOCF | | | | 1.433 (0.112) | 1.266 (0.099) | 2.484 (0.339) | 1.601 (0.150) | 1.414 (0.133) | 2.952 (0.479) | 1.611 (0.188) | 1.423 (0.166) | 2.972 (0.626) |
| Linear | | | | 1.395 (0.118) | 1.232 (0.104) | 2.365 (0.358) | 1.558 (0.151) | 1.376 (0.134) | 2.821 (0.477) | 1.556 (0.191) | 1.374 (0.169) | 2.791 (0.633) |

**ETT_h1 (subseq_50%)**

| | MAE wt XGB | MRE wt XGB | MSE wt XGB | MAE w XGB | MRE w XGB | MSE w XGB | MAE w RNN | MRE w RNN | MSE w RNN | MAE w Transformer | MRE w Transformer | MSE w Transformer |
|---|---|---|---|---|---|---|---|---|---|---|---|---|
| iTransformer | | | | 1.170 (0.000) | 1.033 (0.000) | 1.768 (0.000) | 1.434 (0.047) | 1.266 (0.042) | 2.429 (0.159) | 1.470 (0.053) | 1.298 (0.046) | 2.499 (0.153) |
| SAITS | | | | 1.094 (0.000) | 0.966 (0.000) | 1.593 (0.000) | 1.424 (0.077) | 1.257 (0.068) | 2.433 (0.203) | 1.469 (0.054) | 1.297 (0.048) | 2.477 (0.162) |
| Nonstationary | | | | 1.284 (0.000) | 1.134 (0.000) | 2.090 (0.000) | 1.448 (0.053) | 1.278 (0.046) | 2.471 (0.130) | 1.469 (0.036) | 1.297 (0.032) | 2.471 (0.104) |
| ETSformer | | | | 1.187 (0.000) | 1.048 (0.000) | 1.804 (0.000) | 1.407 (0.059) | 1.243 (0.052) | 2.358 (0.139) | 1.433 (0.058) | 1.265 (0.051) | 2.367 (0.155) |
| PatchTST | | | | 1.253 (0.000) | 1.106 (0.000) | 1.993 (0.000) | 1.476 (0.079) | 1.303 (0.069) | 2.595 (0.217) | 1.502 (0.042) | 1.326 (0.037) | 2.569 (0.120) |
| Crossformer | | | | 1.145 (0.000) | 1.011 (0.000) | 1.730 (0.000) | 1.449 (0.058) | 1.280 (0.051) | 2.506 (0.139) | 1.474 (0.042) | 1.301 (0.037) | 2.495 (0.107) |
| Informer | | | | 1.196 (0.000) | 1.056 (0.000) | 1.764 (0.000) | 1.423 (0.076) | 1.256 (0.067) | 2.426 (0.200) | 1.442 (0.042) | 1.273 (0.037) | 2.395 (0.120) |
| Autoformer | | | | 1.364 (0.000) | 1.205 (0.000) | 2.281 (0.000) | 1.471 (0.061) | 1.299 (0.054) | 2.545 (0.228) | 1.538 (0.072) | 1.358 (0.063) | 2.728 (0.237) |
| Pyraformer | | | | 1.081 (0.000) | 0.955 (0.000) | 1.594 (0.000) | 1.391 (0.077) | 1.228 (0.068) | 2.328 (0.200) | 1.443 (0.042) | 1.274 (0.037) | 2.393 (0.126) |
| Transformer | | | | 1.074 (0.000) | 0.948 (0.000) | 1.399 (0.000) | 1.391 (0.060) | 1.228 (0.053) | 2.320 (0.140) | 1.350 (0.090) | 1.192 (0.079) | 2.138 (0.232) |
| BRITS | | | | 0.999 (0.000) | 0.882 (0.000) | 1.336 (0.000) | 1.153 (0.042) | 1.018 (0.037) | 1.734 (0.056) | 1.146 (0.036) | 1.012 (0.031) | 1.645 (0.093) |
| MRNN | | | | 1.111 (0.000) | 0.982 (0.000) | 1.619 (0.000) | 1.362 (0.061) | 1.202 (0.054) | 2.211 (0.179) | 1.483 (0.055) | 1.310 (0.049) | 2.521 (0.141) |
| GRUD | 1.489 (0.000) | 1.315 (0.000) | 2.781 (0.000) | 1.123 (0.000) | 0.991 (0.000) | 1.585 (0.000) | 1.351 (0.058) | 1.193 (0.051) | 2.194 (0.146) | 1.362 (0.044) | 1.202 (0.039) | 2.179 (0.117) |
| TimesNet | | | | 1.213 (0.000) | 1.071 (0.000) | 1.891 (0.000) | 1.446 (0.058) | 1.277 (0.051) | 2.514 (0.139) | 1.460 (0.041) | 1.290 (0.036) | 2.461 (0.108) |
| MICN | | | | 1.121 (0.000) | 0.990 (0.000) | 1.609 (0.000) | 1.442 (0.052) | 1.273 (0.046) | 2.457 (0.185) | 1.503 (0.060) | 1.327 (0.053) | 2.604 (0.186) |
| SCINet | | | | 1.181 (0.000) | 1.043 (0.000) | 1.767 (0.000) | 1.450 (0.061) | 1.280 (0.054) | 2.514 (0.153) | 1.446 (0.038) | 1.277 (0.034) | 2.414 (0.105) |
| StemGNN | | | | 1.211 (0.000) | 1.069 (0.000) | 1.810 (0.000) | 1.391 (0.067) | 1.228 (0.059) | 2.313 (0.171) | 1.439 (0.045) | 1.270 (0.040) | 2.378 (0.137) |
| FreTS | | | | 1.159 (0.000) | 1.023 (0.000) | 1.667 (0.000) | 1.456 (0.079) | 1.286 (0.070) | 2.516 (0.217) | 1.514 (0.045) | 1.337 (0.040) | 2.608 (0.132) |
| Koopa | | | | 1.233 (0.000) | 1.089 (0.000) | 1.961 (0.000) | 1.446 (0.083) | 1.277 (0.073) | 2.486 (0.228) | 1.509 (0.046) | 1.333 (0.040) | 2.594 (0.133) |
| DLinear | | | | 1.212 (0.000) | 1.071 (0.000) | 1.873 (0.000) | 1.466 (0.058) | 1.295 (0.051) | 2.557 (0.144) | 1.512 (0.044) | 1.335 (0.038) | 2.609 (0.144) |
| FILM | | | | 1.422 (0.000) | 1.255 (0.000) | 2.469 (0.000) | 1.389 (0.057) | 1.226 (0.051) | 2.278 (0.146) | 1.455 (0.036) | 1.285 (0.032) | 2.417 (0.115) |
| CSDI | | | | 1.191 (0.000) | 1.052 (0.000) | 1.758 (0.000) | 1.354 (0.049) | 1.196 (0.043) | 2.194 (0.115) | 1.413 (0.041) | 1.248 (0.036) | 2.301 (0.115) |
| US-GAN | | | | 1.004 (0.000) | 0.886 (0.000) | 1.346 (0.000) | 1.244 (0.052) | 1.098 (0.046) | 1.937 (0.091) | 1.219 (0.079) | 1.077 (0.069) | 1.809 (0.205) |
| GP-VAE | | | | 1.138 (0.000) | 1.005 (0.000) | 1.692 (0.000) | 1.310 (0.046) | 1.157 (0.041) | 2.129 (0.097) | 1.354 (0.077) | 1.195 (0.068) | 2.160 (0.195) |
| Mean | | | | 1.430 (0.000) | 1.263 (0.000) | 2.526 (0.000) | 1.859 (0.095) | 1.642 (0.084) | 3.821 (0.349) | 1.779 (0.035) | 1.571 (0.031) | 3.478 (0.133) |
| Median | | | | 1.499 (0.069) | 1.324 (0.061) | 2.766 (0.240) | 1.858 (0.089) | 1.641 (0.079) | 3.825 (0.328) | 1.777 (0.034) | 1.569 (0.030) | 3.472 (0.133) |
| LOCF | | | | 1.446 (0.094) | 1.277 (0.083) | 2.619 (0.285) | 1.699 (0.237) | 1.500 (0.210) | 3.293 (0.800) | 1.651 (0.182) | 1.458 (0.161) | 3.055 (0.603) |
| Linear | | | | 1.423 (0.091) | 1.256 (0.080) | 2.540 (0.283) | 1.610 (0.258) | 1.422 (0.228) | 3.010 (0.850) | 1.588 (0.193) | 1.402 (0.171) | 2.858 (0.627) |

Table 19: Performance comparison for the regression task on ETT_h1 datasets with 10% point missing and 50% point missing.

**ETT_h1 (point_10%)**

Overall XGB values (wt XGB): MAE wt XGB = 1.181 (0.000), MRE wt XGB = 1.043 (0.000), MSE wt XGB = 1.661 (0.000)

| Model | MAE w XGB | MRE w XGB | MSE w XGB | MAE w RNN | MRE w RNN | MSE w RNN | MAE w Transformer | MRE w Transformer | MSE w Transformer |
|---|---|---|---|---|---|---|---|---|---|
| iTransformer | 1.194 (0.000) | .054 (0.000) | 1.773 (0.000) | 1.403 (0.057) | .239 (0.051) | 2.352 (0.139) | 1.401 (0.071) | .237 (0.063) | 2.260 (0.187) |
| SAITS | 1.186 (0.000) | .047 (0.000) | 1.708 (0.000) | 1.403 (0.058) | .239 (0.051) | 2.352 (0.141) | 1.397 (0.073) | .233 (0.065) | 2.248 (0.193) |
| Nonstationary | 1.161 (0.000) | .025 (0.000) | 1.664 (0.000) | 1.424 (0.056) | .258 (0.050) | 2.417 (0.138) | 1.399 (0.075) | .236 (0.066) | 2.260 (0.198) |
| ETSformer | 1.151 (0.000) | .016 (0.000) | 1.617 (0.000) | 1.400 (0.057) | .236 (0.050) | 2.344 (0.138) | 1.393 (0.070) | .230 (0.062) | 2.238 (0.182) |
| PatchTST | 1.166 (0.000) | .029 (0.000) | 1.672 (0.000) | 1.406 (0.056) | .242 (0.049) | 2.360 (0.136) | 1.390 (0.073) | .228 (0.065) | 2.229 (0.191) |
| Crossformer | 1.152 (0.000) | .018 (0.000) | 1.637 (0.000) | 1.403 (0.056) | .239 (0.050) | 2.351 (0.137) | 1.393 (0.072) | .230 (0.064) | 2.239 (0.187) |
| Informer | 1.155 (0.000) | .020 (0.000) | 1.648 (0.000) | 1.401 (0.057) | .238 (0.051) | 2.347 (0.139) | 1.399 (0.073) | .235 (0.065) | 2.253 (0.192) |
| Autoformer | 1.159 (0.000) | .023 (0.000) | 1.636 (0.000) | 1.401 (0.057) | .237 (0.050) | 2.344 (0.137) | 1.390 (0.071) | .228 (0.062) | 2.228 (0.184) |
| Pyraformer | 1.166 (0.000) | .030 (0.000) | 1.672 (0.000) | 1.404 (0.056) | .239 (0.050) | 2.353 (0.137) | 1.397 (0.072) | .233 (0.064) | 2.247 (0.189) |
| Transformer | 1.144 (0.000) | .010 (0.000) | 1.589 (0.000) | 1.401 (0.057) | .237 (0.050) | 2.346 (0.138) | 1.395 (0.073) | .232 (0.065) | 2.241 (0.192) |
| BRITS | 1.147 (0.000) | .012 (0.000) | 1.574 (0.000) | 1.401 (0.057) | .237 (0.051) | 2.346 (0.139) | 1.394 (0.073) | .231 (0.065) | 2.240 (0.192) |
| MRNN | 1.157 (0.000) | .022 (0.000) | 1.639 (0.000) | 1.374 (0.062) | .213 (0.055) | 2.268 (0.151) | 1.387 (0.078) | .225 (0.069) | 2.221 (0.208) |
| GRID | 1.172 (0.000) | .035 (0.000) | 1.664 (0.000) | 1.394 (0.057) | .231 (0.051) | 2.324 (0.139) | 1.388 (0.071) | .226 (0.063) | 2.221 (0.185) |
| TimesNet | 1.143 (0.000) | .009 (0.000) | 1.590 (0.000) | 1.378 (0.054) | .217 (0.048) | 2.271 (0.128) | 1.378 (0.068) | .217 (0.060) | 2.193 (0.174) |
| MICN | 1.170 (0.000) | .033 (0.000) | 1.688 (0.000) | 1.400 (0.058) | .236 (0.051) | 2.345 (0.142) | 1.395 (0.071) | .232 (0.063) | 2.243 (0.188) |
| SCINet | 1.166 (0.000) | .030 (0.000) | 1.660 (0.000) | 1.404 (0.056) | .240 (0.050) | 2.355 (0.136) | 1.392 (0.074) | .230 (0.065) | 2.237 (0.192) |
| StemGNN | 1.166 (0.000) | .030 (0.000) | 1.671 (0.000) | 1.401 (0.057) | .237 (0.050) | 2.345 (0.139) | 1.394 (0.073) | .231 (0.064) | 2.239 (0.190) |
| FreTS | 1.168 (0.000) | .031 (0.000) | 1.663 (0.000) | 1.407 (0.056) | .242 (0.049) | 2.362 (0.136) | 1.396 (0.073) | .233 (0.065) | 2.247 (0.191) |
| Koopa | 1.193 (0.000) | .053 (0.000) | 1.716 (0.000) | 1.394 (0.057) | .231 (0.051) | 2.325 (0.139) | 1.395 (0.073) | .232 (0.065) | 2.244 (0.191) |
| DLinear | 1.159 (0.000) | .024 (0.000) | 1.651 (0.000) | 1.405 (0.056) | .240 (0.049) | 2.357 (0.136) | 1.394 (0.072) | .231 (0.063) | 2.240 (0.186) |
| FILM | 1.208 (0.000) | .067 (0.000) | 1.784 (0.000) | 1.398 (0.058) | .235 (0.051) | 2.340 (0.140) | 1.399 (0.076) | .236 (0.067) | 2.260 (0.200) |
| CSDI | 1.178 (0.000) | .040 (0.000) | 1.696 (0.000) | 1.401 (0.057) | .237 (0.050) | 2.345 (0.138) | 1.394 (0.074) | .231 (0.066) | 2.241 (0.196) |
| US-GAN | 1.163 (0.000) | .027 (0.000) | 1.637 (0.000) | 1.399 (0.058) | .235 (0.051) | 2.341 (0.141) | 1.389 (0.080) | .227 (0.071) | 2.228 (0.213) |
| GP-VAE | 1.159 (0.000) | .024 (0.000) | 1.652 (0.000) | 1.403 (0.059) | .239 (0.052) | 2.355 (0.145) | 1.397 (0.073) | .234 (0.064) | 2.249 (0.192) |
| Mean | 1.209 (0.000) | .067 (0.000) | 1.716 (0.000) | 1.424 (0.069) | .258 (0.061) | 2.416 (0.178) | 1.445 (0.075) | .276 (0.067) | 2.390 (0.213) |
| Median | 1.208 (0.001) | .066 (0.001) | 1.717 (0.001) | 1.423 (0.069) | .257 (0.061) | 2.415 (0.176) | 1.448 (0.072) | .278 (0.063) | 2.398 (0.202) |
| LOCF | 1.194 (0.020) | .054 (0.017) | 1.682 (0.050) | 1.417 (0.066) | .251 (0.058) | 2.395 (0.168) | 1.430 (0.076) | .263 (0.067) | 2.346 (0.212) |
| Linear | 1.188 (0.020) | .049 (0.018) | 1.677 (0.044) | 1.414 (0.064) | .248 (0.057) | 2.385 (0.162) | 1.422 (0.077) | .256 (0.068) | 2.321 (0.212) |

**ETT_h1 (point_50%)**

Overall XGB values (wt XGB): MAE wt XGB = 1.287 (0.000), MRE wt XGB = 1.137 (0.000), MSE wt XGB = 1.950 (0.000)

| Model | MAE w XGB | MRE w XGB | MSE w XGB | MAE w RNN | MRE w RNN | MSE w RNN | MAE w Transformer | MRE w Transformer | MSE w Transformer |
|---|---|---|---|---|---|---|---|---|---|
| iTransformer | 1.224 (0.000) | .081 (0.000) | 1.778 (0.000) | 1.377 (0.062) | .216 (0.055) | 2.280 (0.151) | 1.406 (0.061) | .242 (0.061) | 2.285 (0.185) |
| SAITS | 1.175 (0.000) | .038 (0.000) | 1.654 (0.000) | 1.402 (0.056) | .238 (0.050) | 2.349 (0.136) | 1.401 (0.083) | .237 (0.074) | 2.262 (0.224) |
| Nonstationary | 1.227 (0.000) | .083 (0.000) | 1.802 (0.000) | 1.446 (0.061) | .276 (0.054) | 2.484 (0.154) | 1.384 (0.087) | .222 (0.076) | 2.236 (0.202) |
| ETSformer | 1.222 (0.000) | .079 (0.000) | 1.885 (0.000) | 1.329 (0.060) | .173 (0.053) | 2.125 (0.138) | 1.351 (0.061) | .193 (0.054) | 2.113 (0.153) |
| PatchTST | 1.234 (0.000) | .090 (0.000) | 1.840 (0.000) | 1.398 (0.059) | .235 (0.052) | 2.344 (0.143) | 1.370 (0.074) | .210 (0.066) | 2.176 (0.192) |
| Crossformer | 1.228 (0.000) | .084 (0.000) | 1.838 (0.000) | 1.391 (0.059) | .228 (0.051) | 2.323 (0.141) | 1.380 (0.070) | .218 (0.062) | 2.204 (0.181) |
| Informer | 1.214 (0.000) | .072 (0.000) | 1.782 (0.000) | 1.391 (0.059) | .229 (0.052) | 2.315 (0.142) | 1.398 (0.076) | .234 (0.068) | 2.252 (0.205) |
| Autoformer | 1.348 (0.000) | .190 (0.000) | 2.060 (0.000) | 1.450 (0.108) | .280 (0.095) | 2.444 (0.368) | 1.442 (0.155) | .273 (0.137) | 2.438 (0.520) |
| Pyraformer | 1.169 (0.000) | .032 (0.000) | 1.640 (0.000) | 1.366 (0.057) | .206 (0.050) | 2.235 (0.134) | 1.373 (0.071) | .212 (0.063) | 2.176 (0.186) |
| Transformer | 1.170 (0.000) | .034 (0.000) | 1.631 (0.000) | 1.378 (0.056) | .217 (0.049) | 2.275 (0.133) | 1.357 (0.081) | .198 (0.072) | 2.130 (0.213) |
| BRITS | 1.177 (0.000) | .039 (0.000) | 1.680 (0.000) | 1.371 (0.061) | .210 (0.054) | 2.259 (0.147) | 1.364 (0.076) | .205 (0.068) | 2.155 (0.198) |
| MRNN | 1.070 (0.000) | .945 (0.000) | 1.417 (0.000) | 1.250 (0.076) | .104 (0.067) | 1.900 (0.190) | 1.379 (0.058) | .218 (0.051) | 2.216 (0.142) |
| GRID | 1.205 (0.000) | .064 (0.000) | 1.870 (0.000) | 1.339 (0.061) | .182 (0.054) | 2.154 (0.145) | 1.326 (0.074) | .171 (0.065) | 2.043 (0.192) |
| TimesNet | 1.127 (0.000) | .995 (0.000) | 1.513 (0.000) | 1.333 (0.054) | .177 (0.048) | 2.149 (0.125) | 1.316 (0.066) | .162 (0.058) | 2.027 (0.163) |
| MICN | 1.163 (0.000) | 1.027 (0.000) | 1.611 (0.000) | 1.305 (0.078) | .152 (0.065) | 2.049 (0.189) | 1.379 (0.070) | .218 (0.062) | 2.185 (0.195) |
| SCINet | 1.174 (0.000) | .037 (0.000) | 1.654 (0.000) | 1.411 (0.055) | .246 (0.049) | 2.390 (0.134) | 1.375 (0.080) | .214 (0.070) | 2.203 (0.209) |
| StemGNN | 1.139 (0.000) | 1.006 (0.000) | 1.561 (0.000) | 1.392 (0.057) | .229 (0.051) | 2.311 (0.139) | 1.410 (0.078) | .245 (0.069) | 2.285 (0.212) |
| FreTS | 1.207 (0.000) | .066 (0.000) | 1.764 (0.000) | 1.406 (0.056) | .242 (0.050) | 2.361 (0.136) | 1.397 (0.082) | .233 (0.073) | 2.252 (0.219) |
| Koopa | 1.227 (0.000) | .084 (0.000) | 1.804 (0.000) | 1.384 (0.064) | .222 (0.056) | 2.304 (0.153) | 1.419 (0.062) | .253 (0.055) | 2.315 (0.169) |
| DLinear | 1.216 (0.000) | .074 (0.000) | 1.808 (0.000) | 1.380 (0.056) | .219 (0.049) | 2.283 (0.132) | 1.397 (0.076) | .234 (0.067) | 2.254 (0.200) |
| FILM | 1.247 (0.000) | .101 (0.000) | 1.880 (0.000) | 1.379 (0.057) | .218 (0.051) | 2.281 (0.137) | 1.442 (0.048) | .273 (0.042) | 2.391 (0.134) |
| CSDI | 1.136 (0.000) | .003 (0.000) | 1.550 (0.000) | 1.373 (0.050) | .212 (0.044) | 2.256 (0.114) | 1.393 (0.070) | .230 (0.062) | 2.245 (0.184) |
| US-GAN | 1.155 (0.000) | .019 (0.000) | 1.691 (0.000) | 1.299 (0.040) | .147 (0.036) | 2.022 (0.103) | 1.283 (0.055) | .133 (0.048) | 1.944 (0.144) |
| GP-VAE | 1.178 (0.000) | .040 (0.000) | 1.721 (0.000) | 1.336 (0.066) | .180 (0.058) | 2.179 (0.156) | 1.340 (0.076) | .183 (0.068) | 2.098 (0.203) |
| Mean | 1.431 (0.000) | .264 (0.000) | 2.304 (0.000) | 1.512 (0.120) | .335 (0.106) | 2.625 (0.351) | 1.627 (0.072) | .437 (0.064) | 3.010 (0.233) |
| Median | 1.517 (0.086) | 1.340 (0.076) | 2.572 (0.269) | 1.531 (0.128) | .352 (0.113) | 2.705 (0.382) | 1.671 (0.069) | .476 (0.061) | 3.142 (0.213) |
| LOCF | 1.446 (0.123) | 1.277 (0.109) | 2.416 (0.312) | 1.477 (0.134) | .304 (0.118) | 2.552 (0.388) | 1.571 (0.158) | .388 (0.139) | 2.821 (0.501) |
| Linear | 1.393 (0.140) | 1.230 (0.124) | 2.281 (0.357) | 1.453 (0.127) | .283 (0.112) | 2.486 (0.362) | 1.525 (0.163) | .347 (0.144) | 2.670 (0.516) |

Table 20: Performance comparison for the regression task on PeMS datasets with 50% point missing, 50% block missing and 50% subsequence missing.

**PeMS (point 50%)** — overall XGB (wt): MAE wt XGB 0.591 (0.000), MRE wt XGB 0.475 (0.000), MSE wt XGB 0.692 (0.000)

| Model | MAE w XGB | MRE w XGB | MSE w XGB | MAE w RNN | MRE w RNN | MSE w RNN | MAE w Transformer | MRE w Transformer | MSE w Transformer |
|---|---|---|---|---|---|---|---|---|---|
| Transformer | 0.546 (0.000) | 0.439 (0.000) | 0.639 (0.000) | 0.440 (0.020) | 0.354 (0.016) | 0.423 (0.033) | 0.432 (0.012) | 0.347 (0.010) | 0.410 (0.020) |
| SAITS | 0.538 (0.000) | 0.432 (0.000) | 0.621 (0.000) | 0.428 (0.015) | 0.344 (0.012) | 0.400 (0.020) | 0.414 (0.006) | 0.333 (0.005) | 0.380 (0.007) |
| Nonstationary | 0.538 (0.000) | 0.433 (0.000) | 0.581 (0.000) | 0.447 (0.027) | 0.359 (0.022) | 0.444 (0.040) | 0.422 (0.024) | 0.340 (0.019) | 0.400 (0.036) |
| ETSformer | 0.545 (0.000) | 0.438 (0.000) | 0.608 (0.000) | 0.433 (0.018) | 0.348 (0.015) | 0.444 (0.025) | 0.435 (0.017) | 0.335 (0.013) | 0.388 (0.031) |
| PatchTST | 0.553 (0.000) | 0.445 (0.000) | 0.640 (0.000) | 0.431 (0.013) | 0.347 (0.013) | 0.405 (0.025) | 0.442 (0.013) | 0.355 (0.011) | 0.423 (0.012) |
| Crossformer | 0.578 (0.000) | 0.465 (0.000) | 0.693 (0.000) | 0.427 (0.013) | 0.344 (0.011) | 0.401 (0.018) | 0.423 (0.007) | 0.340 (0.005) | 0.395 (0.016) |
| Informer | 0.546 (0.000) | 0.439 (0.000) | 0.625 (0.000) | 0.437 (0.015) | 0.352 (0.012) | 0.411 (0.021) | 0.420 (0.018) | 0.338 (0.014) | 0.396 (0.025) |
| Autoformer | 0.630 (0.000) | 0.507 (0.000) | 0.796 (0.000) | 0.495 (0.007) | 0.399 (0.006) | 0.488 (0.000) | 0.485 (0.009) | 0.390 (0.007) | 0.470 (0.018) |
| Pyraformer | 0.536 (0.000) | 0.431 (0.000) | 0.582 (0.000) | 0.430 (0.017) | 0.347 (0.013) | 0.403 (0.024) | 0.432 (0.000) | 0.341 (0.011) | 0.412 (0.017) |
| Transformer | 0.542 (0.000) | 0.436 (0.000) | 0.620 (0.000) | 0.427 (0.017) | 0.344 (0.014) | 0.399 (0.021) | 0.418 (0.013) | 0.336 (0.010) | 0.392 (0.014) |
| BRITS | 0.565 (0.000) | 0.455 (0.000) | 0.685 (0.000) | 0.432 (0.015) | 0.347 (0.012) | 0.402 (0.018) | 0.434 (0.014) | 0.349 (0.011) | 0.411 (0.023) |
| MRNN | 0.580 (0.000) | 0.467 (0.000) | 0.692 (0.000) | 0.456 (0.020) | 0.367 (0.016) | 0.436 (0.030) | 0.442 (0.017) | 0.356 (0.013) | 0.415 (0.023) |
| GRUD | 0.534 (0.000) | 0.429 (0.000) | 0.620 (0.000) | 0.451 (0.013) | 0.363 (0.011) | 0.429 (0.021) | 0.434 (0.008) | 0.349 (0.007) | 0.409 (0.013) |
| TimesNet | 0.541 (0.000) | 0.435 (0.000) | 0.614 (0.000) | 0.441 (0.016) | 0.355 (0.013) | 0.415 (0.020) | 0.440 (0.021) | 0.354 (0.017) | 0.418 (0.033) |
| MICN | 0.591 (0.000) | 0.475 (0.000) | 0.685 (0.000) | 0.505 (0.017) | 0.406 (0.013) | 0.521 (0.031) | 0.477 (0.011) | 0.384 (0.009) | 0.469 (0.021) |
| SCINet | 0.547 (0.000) | 0.440 (0.000) | 0.624 (0.000) | 0.474 (0.014) | 0.382 (0.011) | 0.445 (0.019) | 0.482 (0.028) | 0.387 (0.022) | 0.478 (0.036) |
| StemGNN | 0.548 (0.000) | 0.441 (0.000) | 0.642 (0.000) | 0.462 (0.015) | 0.372 (0.012) | 0.453 (0.024) | 0.448 (0.011) | 0.360 (0.008) | 0.433 (0.020) |
| FreTS | 0.558 (0.000) | 0.449 (0.000) | 0.623 (0.000) | 0.447 (0.012) | 0.360 (0.010) | 0.421 (0.020) | 0.437 (0.010) | 0.352 (0.008) | 0.407 (0.010) |
| Koopa | 0.574 (0.000) | 0.461 (0.000) | 0.648 (0.000) | 0.485 (0.012) | 0.390 (0.009) | 0.459 (0.018) | 0.472 (0.021) | 0.379 (0.017) | 0.449 (0.032) |
| DLinear | 0.551 (0.000) | 0.443 (0.000) | 0.610 (0.000) | 0.427 (0.015) | 0.344 (0.012) | 0.401 (0.021) | 0.430 (0.014) | 0.346 (0.011) | 0.414 (0.024) |
| FILM | 0.597 (0.000) | 0.480 (0.000) | 0.688 (0.000) | 0.443 (0.018) | 0.356 (0.015) | 0.408 (0.033) | 0.441 (0.015) | 0.355 (0.012) | 0.394 (0.016) |
| CSDI | 0.531 (0.000) | 0.427 (0.000) | 0.604 (0.000) | 0.428 (0.023) | 0.345 (0.018) | 0.417 (0.033) | 0.428 (0.008) | 0.344 (0.008) | 0.416 (0.016) |
| US-GAN | 0.531 (0.000) | 0.427 (0.000) | 0.599 (0.000) | 0.427 (0.020) | 0.344 (0.016) | 0.399 (0.028) | 0.397 (0.011) | 0.320 (0.009) | 0.380 (0.018) |
| GP-VAE | 0.547 (0.000) | 0.440 (0.000) | 0.636 (0.000) | 0.433 (0.023) | 0.348 (0.019) | 0.408 (0.033) | 0.439 (0.014) | 0.353 (0.011) | 0.424 (0.018) |
| Mean | 0.607 (0.000) | 0.488 (0.001) | 0.723 (0.000) | 0.428 (0.027) | 0.344 (0.022) | 0.409 (0.037) | 0.400 (0.012) | 0.322 (0.009) | 0.389 (0.018) |
| Median | 0.606 (0.001) | 0.488 (0.001) | 0.720 (0.003) | 0.422 (0.023) | 0.340 (0.019) | 0.401 (0.032) | 0.397 (0.011) | 0.320 (0.009) | 0.380 (0.018) |
| LOCF | 0.604 (0.003) | 0.486 (0.002) | 0.709 (0.017) | 0.436 (0.028) | 0.351 (0.022) | 0.422 (0.040) | 0.413 (0.026) | 0.332 (0.021) | 0.391 (0.028) |
| Linear | 0.592 (0.022) | 0.476 (0.017) | 0.688 (0.039) | 0.444 (0.030) | 0.357 (0.023) | 0.431 (0.043) | 0.419 (0.028) | 0.337 (0.023) | 0.397 (0.031) |

**PeMS (block 50%)** — overall XGB (wt): MAE wt XGB 0.624 (0.000), MRE wt XGB 0.502 (0.000), MSE wt XGB 0.780 (0.000)

| Model | MAE w XGB | MRE w XGB | MSE w XGB | MAE w RNN | MRE w RNN | MSE w RNN | MAE w Transformer | MRE w Transformer | MSE w Transformer |
|---|---|---|---|---|---|---|---|---|---|
| Transformer | 0.606 (0.000) | 0.487 (0.000) | 0.800 (0.000) | 0.434 (0.018) | 0.349 (0.014) | 0.432 (0.024) | 0.419 (0.022) | 0.317 (0.018) | 0.415 (0.032) |
| SAITS | 0.544 (0.000) | 0.437 (0.000) | 0.638 (0.000) | 0.407 (0.021) | 0.327 (0.017) | 0.383 (0.026) | 0.408 (0.011) | 0.328 (0.008) | 0.386 (0.019) |
| Nonstationary | 0.604 (0.000) | 0.486 (0.000) | 0.711 (0.000) | 0.444 (0.011) | 0.357 (0.009) | 0.434 (0.020) | 0.405 (0.024) | 0.325 (0.019) | 0.375 (0.033) |
| ETSformer | 0.555 (0.000) | 0.447 (0.000) | 0.639 (0.000) | 0.447 (0.011) | 0.359 (0.009) | 0.432 (0.015) | 0.422 (0.012) | 0.339 (0.010) | 0.392 (0.019) |
| PatchTST | 0.548 (0.000) | 0.441 (0.000) | 0.625 (0.000) | 0.416 (0.014) | 0.335 (0.012) | 0.397 (0.019) | 0.431 (0.013) | 0.347 (0.011) | 0.421 (0.013) |
| Crossformer | 0.547 (0.000) | 0.440 (0.000) | 0.622 (0.000) | 0.405 (0.019) | 0.326 (0.016) | 0.379 (0.027) | 0.409 (0.009) | 0.329 (0.007) | 0.379 (0.016) |
| Informer | 0.547 (0.000) | 0.440 (0.000) | 0.683 (0.000) | 0.416 (0.019) | 0.335 (0.011) | 0.399 (0.016) | 0.416 (0.013) | 0.334 (0.011) | 0.394 (0.021) |
| Autoformer | 0.584 (0.000) | 0.470 (0.000) | 0.616 (0.000) | 0.487 (0.016) | 0.392 (0.013) | 0.472 (0.026) | 0.477 (0.024) | 0.384 (0.019) | 0.454 (0.037) |
| Pyraformer | 0.540 (0.000) | 0.434 (0.000) | 0.616 (0.000) | 0.399 (0.019) | 0.321 (0.015) | 0.377 (0.022) | 0.411 (0.015) | 0.331 (0.012) | 0.394 (0.022) |
| Transformer | 0.560 (0.000) | 0.451 (0.000) | 0.655 (0.000) | 0.413 (0.013) | 0.332 (0.010) | 0.395 (0.017) | 0.407 (0.019) | 0.327 (0.015) | 0.386 (0.027) |
| BRITS | 0.637 (0.000) | 0.512 (0.000) | 1.022 (0.000) | 0.415 (0.015) | 0.334 (0.012) | 0.385 (0.018) | 0.429 (0.014) | 0.345 (0.012) | 0.415 (0.018) |
| MRNN | 0.545 (0.000) | 0.439 (0.000) | 0.588 (0.000) | 0.444 (0.022) | 0.357 (0.018) | 0.406 (0.031) | 0.455 (0.026) | 0.366 (0.021) | 0.428 (0.036) |
| GRUD | 0.539 (0.000) | 0.433 (0.000) | 0.587 (0.000) | 0.419 (0.016) | 0.337 (0.018) | 0.397 (0.021) | 0.418 (0.009) | 0.336 (0.007) | 0.395 (0.012) |
| TimesNet | 0.547 (0.000) | 0.441 (0.000) | 0.617 (0.000) | 0.416 (0.018) | 0.335 (0.015) | 0.387 (0.027) | 0.416 (0.018) | 0.334 (0.014) | 0.393 (0.032) |
| MICN | 0.646 (0.000) | 0.520 (0.000) | 0.828 (0.000) | 0.499 (0.018) | 0.401 (0.015) | 0.541 (0.031) | 0.442 (0.020) | 0.356 (0.016) | 0.441 (0.029) |
| SCINet | 0.627 (0.000) | 0.504 (0.000) | 0.807 (0.000) | 0.472 (0.007) | 0.380 (0.005) | 0.455 (0.013) | 0.485 (0.028) | 0.390 (0.022) | 0.485 (0.040) |
| StemGNN | 0.553 (0.000) | 0.445 (0.000) | 0.624 (0.000) | 0.442 (0.016) | 0.356 (0.013) | 0.429 (0.028) | 0.447 (0.016) | 0.359 (0.013) | 0.441 (0.018) |
| FreTS | 0.553 (0.000) | 0.445 (0.000) | 0.603 (0.000) | 0.426 (0.016) | 0.342 (0.013) | 0.403 (0.021) | 0.427 (0.014) | 0.343 (0.011) | 0.401 (0.019) |
| Koopa | 0.594 (0.000) | 0.478 (0.000) | 0.815 (0.000) | 0.449 (0.016) | 0.361 (0.013) | 0.429 (0.028) | 0.428 (0.014) | 0.360 (0.016) | 0.426 (0.022) |
| DLinear | 0.563 (0.000) | 0.453 (0.000) | 0.682 (0.000) | 0.433 (0.010) | 0.349 (0.008) | 0.411 (0.010) | 0.441 (0.009) | 0.354 (0.007) | 0.426 (0.031) |
| FILM | 0.577 (0.000) | 0.464 (0.000) | 0.664 (0.000) | 0.426 (0.006) | 0.343 (0.005) | 0.392 (0.008) | 0.451 (0.026) | 0.363 (0.021) | 0.432 (0.031) |
| CSDI | 0.555 (0.000) | 0.447 (0.000) | 0.650 (0.000) | 0.430 (0.011) | 0.346 (0.009) | 0.444 (0.018) | 0.421 (0.006) | 0.339 (0.005) | 0.420 (0.011) |
| US-GAN | 0.603 (0.000) | 0.485 (0.000) | 0.796 (0.000) | 0.416 (0.015) | 0.335 (0.012) | 0.383 (0.017) | 0.435 (0.012) | 0.350 (0.009) | 0.429 (0.015) |
| GP-VAE | 0.539 (0.000) | 0.433 (0.000) | 0.617 (0.000) | 0.408 (0.018) | 0.328 (0.015) | 0.379 (0.023) | 0.421 (0.006) | 0.339 (0.005) | 0.402 (0.007) |
| Mean | 0.603 (0.000) | 0.485 (0.000) | 0.721 (0.000) | 0.405 (0.015) | 0.326 (0.013) | 0.334 (0.020) | 0.399 (0.008) | 0.321 (0.006) | 0.349 (0.025) |
| Median | 0.616 (0.013) | 0.496 (0.011) | 0.747 (0.026) | 0.408 (0.012) | 0.328 (0.010) | 0.335 (0.015) | 0.395 (0.011) | 0.317 (0.009) | 0.340 (0.024) |
| LOCF | 0.613 (0.012) | 0.493 (0.010) | 0.737 (0.026) | 0.441 (0.039) | 0.354 (0.031) | 0.393 (0.084) | 0.433 (0.056) | 0.348 (0.045) | 0.399 (0.087) |
| Linear | 0.611 (0.011) | 0.492 (0.009) | 0.732 (0.024) | 0.443 (0.043) | 0.356 (0.035) | 0.431 (0.075) | 0.435 (0.049) | 0.350 (0.040) | 0.403 (0.076) |

**PeMS (subsequence 50%)** — overall XGB (wt): MAE wt XGB 0.662 (0.000), MRE wt XGB 0.533 (0.000), MSE wt XGB 0.873 (0.000)

| Model | MAE w XGB | MRE w XGB | MSE w XGB | MAE w RNN | MRE w RNN | MSE w RNN | MAE w Transformer | MRE w Transformer | MSE w Transformer |
|---|---|---|---|---|---|---|---|---|---|
| Transformer | 0.608 (0.000) | 0.489 (0.000) | 0.766 (0.000) | 0.442 (0.007) | 0.356 (0.006) | 0.420 (0.010) | 0.376 (0.008) | 0.303 (0.007) | 0.341 (0.014) |
| SAITS | 0.543 (0.000) | 0.437 (0.000) | 0.615 (0.000) | 0.423 (0.021) | 0.341 (0.017) | 0.412 (0.028) | 0.404 (0.011) | 0.325 (0.009) | 0.386 (0.016) |
| Nonstationary | 0.536 (0.000) | 0.431 (0.000) | 0.713 (0.000) | 0.471 (0.029) | 0.379 (0.023) | 0.470 (0.063) | 0.382 (0.020) | 0.307 (0.016) | 0.331 (0.027) |
| ETSformer | 0.603 (0.000) | 0.485 (0.000) | 0.729 (0.000) | 0.479 (0.012) | 0.385 (0.010) | 0.470 (0.017) | 0.489 (0.015) | 0.394 (0.012) | 0.499 (0.028) |
| PatchTST | 0.550 (0.000) | 0.443 (0.000) | 0.632 (0.000) | 0.432 (0.011) | 0.347 (0.009) | 0.412 (0.015) | 0.440 (0.014) | 0.354 (0.011) | 0.426 (0.020) |
| Crossformer | 0.546 (0.000) | 0.439 (0.000) | 0.624 (0.000) | 0.425 (0.016) | 0.342 (0.013) | 0.406 (0.024) | 0.421 (0.005) | 0.339 (0.004) | 0.405 (0.008) |
| Informer | 0.533 (0.000) | 0.429 (0.000) | 0.606 (0.000) | 0.430 (0.017) | 0.346 (0.014) | 0.421 (0.022) | 0.416 (0.005) | 0.314 (0.004) | 0.396 (0.005) |
| Autoformer | 0.608 (0.000) | 0.489 (0.000) | 0.743 (0.000) | 0.451 (0.019) | 0.363 (0.015) | 0.430 (0.023) | 0.463 (0.010) | 0.372 (0.010) | 0.450 (0.022) |
| Pyraformer | 0.550 (0.000) | 0.443 (0.000) | 0.629 (0.000) | 0.417 (0.011) | 0.335 (0.011) | 0.401 (0.018) | 0.417 (0.013) | 0.335 (0.011) | 0.404 (0.015) |
| Transformer | 0.555 (0.000) | 0.446 (0.000) | 0.643 (0.000) | 0.420 (0.016) | 0.338 (0.015) | 0.410 (0.022) | 0.402 (0.017) | 0.324 (0.014) | 0.384 (0.021) |
| BRITS | 0.567 (0.000) | 0.456 (0.000) | 0.679 (0.000) | 0.436 (0.019) | 0.351 (0.015) | 0.415 (0.027) | 0.434 (0.008) | 0.349 (0.006) | 0.416 (0.016) |
| MRNN | 0.566 (0.000) | 0.455 (0.000) | 0.629 (0.000) | 0.469 (0.009) | 0.377 (0.008) | 0.441 (0.011) | 0.420 (0.021) | 0.338 (0.017) | 0.383 (0.028) |
| GRUD | 0.528 (0.000) | 0.424 (0.000) | 0.584 (0.000) | 0.421 (0.013) | 0.339 (0.013) | 0.399 (0.015) | 0.417 (0.007) | 0.336 (0.006) | 0.399 (0.011) |
| TimesNet | 0.534 (0.000) | 0.430 (0.000) | 0.591 (0.000) | 0.432 (0.008) | 0.347 (0.015) | 0.408 (0.022) | 0.424 (0.009) | 0.341 (0.008) | 0.406 (0.010) |
| MICN | 0.658 (0.000) | 0.529 (0.000) | 0.898 (0.000) | 0.522 (0.017) | 0.420 (0.013) | 0.609 (0.040) | 0.459 (0.039) | 0.369 (0.031) | 0.481 (0.059) |
| SCINet | 0.581 (0.000) | 0.467 (0.000) | 0.681 (0.000) | 0.493 (0.000) | 0.397 (0.008) | 0.519 (0.010) | 0.446 (0.015) | 0.358 (0.010) | 0.419 (0.017) |
| StemGNN | 0.562 (0.000) | 0.452 (0.000) | 0.647 (0.000) | 0.462 (0.008) | 0.371 (0.008) | 0.456 (0.013) | 0.437 (0.017) | 0.351 (0.014) | 0.422 (0.028) |
| FreTS | 0.563 (0.000) | 0.453 (0.000) | 0.639 (0.000) | 0.431 (0.016) | 0.347 (0.013) | 0.419 (0.023) | 0.418 (0.009) | 0.336 (0.007) | 0.399 (0.010) |
| Koopa | 0.556 (0.000) | 0.447 (0.000) | 0.637 (0.000) | 0.449 (0.004) | 0.361 (0.003) | 0.429 (0.008) | 0.441 (0.013) | 0.355 (0.011) | 0.418 (0.021) |
| DLinear | 0.551 (0.000) | 0.443 (0.000) | 0.623 (0.000) | 0.451 (0.019) | 0.363 (0.015) | 0.430 (0.023) | 0.439 (0.013) | 0.353 (0.011) | 0.426 (0.020) |
| FILM | 0.610 (0.000) | 0.491 (0.000) | 0.747 (0.000) | 0.520 (0.010) | 0.418 (0.008) | 0.615 (0.024) | 0.423 (0.013) | 0.341 (0.011) | 0.405 (0.020) |
| CSDI | 0.552 (0.000) | 0.444 (0.000) | 0.639 (0.000) | 0.447 (0.014) | 0.359 (0.011) | 0.472 (0.024) | 0.409 (0.008) | 0.329 (0.014) | 0.399 (0.023) |
| US-GAN | 0.551 (0.000) | 0.444 (0.000) | 0.623 (0.000) | 0.429 (0.011) | 0.345 (0.009) | 0.399 (0.011) | 0.420 (0.020) | 0.345 (0.011) | 0.382 (0.023) |
| GP-VAE | 0.544 (0.000) | 0.438 (0.000) | 0.614 (0.000) | 0.424 (0.012) | 0.342 (0.012) | 0.403 (0.015) | 0.417 (0.021) | 0.336 (0.017) | 0.396 (0.029) |
| Mean | 0.675 (0.039) | 0.543 (0.032) | 0.901 (0.000) | 0.457 (0.018) | 0.367 (0.014) | 0.418 (0.029) | 0.454 (0.017) | 0.365 (0.014) | 0.389 (0.021) |
| Median | 0.715 (0.039) | 0.575 (0.032) | 1.078 (0.177) | 0.534 (0.080) | 0.429 (0.064) | 0.578 (0.164) | 0.438 (0.021) | 0.353 (0.017) | 0.364 (0.031) |
| LOCF | 0.709 (0.033) | 0.571 (0.026) | 1.033 (0.158) | 0.547 (0.068) | 0.440 (0.055) | 0.625 (0.150) | 0.434 (0.018) | 0.349 (0.015) | 0.382 (0.036) |
| Linear | 0.702 (0.031) | 0.565 (0.025) | 1.001 (0.148) | 0.596 (0.104) | 0.480 (0.083) | 0.701 (0.187) | 0.427 (0.023) | 0.343 (0.018) | 0.372 (0.038) |

Table 21: Performance comparison for the forecasting task on ETT_h1 datasets with 50% block missing and 50% subsequence missing.

**ETT_h1 (block_50%)**

| Model | MAE wt XGB | MRE wt XGB | MSE wt XGB | MAE w XGB | MRE w XGB | MSE w XGB | MAE w RNN | MRE w RNN | MSE w RNN | MAE w Transformer | MRE w Transformer | MSE w Transformer |
|---|---|---|---|---|---|---|---|---|---|---|---|---|
| iTransformer | | | | 1.137 (0.000) | 1.006 (0.000) | 1.488 (0.000) | 1.240 (0.073) | 1.097 (0.065) | 1.897 (0.156) | 0.347 (0.323) | 0.307 (0.286) | 0.267 (0.430) |
| SAITS | | | | 1.132 (0.000) | 1.001 (0.000) | 1.503 (0.000) | 1.224 (0.079) | 1.083 (0.070) | 1.861 (0.172) | 0.351 (0.335) | 0.311 (0.297) | 0.275 (0.452) |
| Nonstationary | | | | 1.212 (0.000) | 1.072 (0.000) | 1.702 (0.000) | 1.335 (0.076) | 1.180 (0.067) | 2.163 (0.183) | 0.366 (0.365) | 0.324 (0.323) | 0.309 (0.512) |
| ETSformer | | | | 1.011 (0.000) | 0.894 (0.000) | 1.339 (0.000) | 1.149 (0.159) | 1.017 (0.141) | 1.695 (0.365) | 0.304 (0.299) | 0.269 (0.264) | 0.220 (0.356) |
| PatchTST | | | | 1.150 (0.000) | 0.972 (0.000) | 1.591 (0.000) | 1.194 (0.091) | 1.056 (0.081) | 1.768 (0.191) | 0.317 (0.345) | 0.280 (0.305) | 0.258 (0.443) |
| Crossformer | | | | 1.099 (0.000) | 0.972 (0.000) | 1.482 (0.000) | 1.192 (0.079) | 1.055 (0.069) | 1.758 (0.165) | 0.325 (0.347) | 0.287 (0.307) | 0.264 (0.454) |
| Informer | | | | 1.163 (0.000) | 1.028 (0.000) | 1.672 (0.000) | 1.215 (0.080) | 1.075 (0.071) | 1.813 (0.169) | 0.337 (0.366) | 0.298 (0.324) | 0.284 (0.493) |
| Autoformer | | | | 1.212 (0.000) | 1.072 (0.000) | 1.812 (0.000) | 1.127 (0.058) | 0.997 (0.051) | 1.500 (0.116) | 0.395 (0.153) | 0.349 (0.135) | 0.253 (0.141) |
| Pyraformer | | | | 1.021 (0.000) | 0.903 (0.000) | 1.221 (0.000) | 1.220 (0.074) | 1.079 (0.065) | 1.827 (0.158) | 0.352 (0.353) | 0.311 (0.312) | 0.287 (0.482) |
| Transformer | | | | 1.112 (0.000) | 0.983 (0.000) | 1.459 (0.000) | 1.152 (0.093) | 1.019 (0.083) | 1.688 (0.194) | 0.328 (0.346) | 0.290 (0.306) | 0.267 (0.456) |
| BRITS | | | | 0.788 (0.000) | 0.697 (0.000) | 0.814 (0.000) | 0.975 (0.022) | 0.862 (0.020) | 1.164 (0.081) | 0.441 (0.385) | 0.390 (0.341) | 0.385 (0.480) |
| MRNN | | | | 0.750 (0.000) | 0.663 (0.000) | 0.802 (0.000) | 1.050 (0.076) | 0.929 (0.067) | 1.431 (0.156) | 0.426 (0.325) | 0.377 (0.287) | 0.339 (0.370) |
| GRUD | 1.231 (0.000) | 1.089 (0.000) | 1.766 (0.000) | 0.989 (0.000) | 0.875 (0.000) | 1.165 (0.000) | 1.199 (0.062) | 1.060 (0.054) | 1.728 (0.151) | 0.362 (0.356) | 0.320 (0.356) | 0.325 (0.562) |
| TimesNet | | | | 1.043 (0.000) | 0.922 (0.000) | 1.369 (0.000) | 1.146 (0.107) | 1.014 (0.095) | 1.667 (0.222) | 0.312 (0.333) | 0.276 (0.294) | 0.244 (0.417) |
| MICN | | | | 1.096 (0.000) | 0.969 (0.000) | 1.459 (0.000) | 1.187 (0.115) | 1.050 (0.102) | 1.772 (0.252) | 0.288 (0.331) | 0.255 (0.293) | 0.225 (0.398) |
| SCINet | | | | 1.125 (0.000) | 0.995 (0.000) | 1.520 (0.000) | 1.186 (0.083) | 1.049 (0.073) | 1.759 (0.174) | 0.325 (0.351) | 0.287 (0.310) | 0.267 (0.460) |
| StemGNN | | | | 1.094 (0.000) | 0.967 (0.000) | 1.421 (0.000) | 1.254 (0.069) | 1.109 (0.061) | 1.913 (0.143) | 0.344 (0.340) | 0.304 (0.301) | 0.276 (0.460) |
| FreTS | | | | 1.126 (0.000) | 0.996 (0.000) | 1.468 (0.000) | 1.216 (0.083) | 1.075 (0.073) | 1.809 (0.179) | 0.326 (0.338) | 0.288 (0.299) | 0.261 (0.442) |
| Koopa | | | | 1.284 (0.000) | 1.136 (0.000) | 1.990 (0.000) | 1.202 (0.076) | 1.063 (0.067) | 1.784 (0.157) | 0.343 (0.326) | 0.303 (0.288) | 0.268 (0.432) |
| DLinear | | | | 1.080 (0.000) | 0.955 (0.000) | 1.391 (0.000) | 1.194 (0.093) | 1.056 (0.082) | 1.774 (0.196) | 0.309 (0.334) | 0.273 (0.295) | 0.243 (0.417) |
| FILM | | | | 1.271 (0.000) | 1.124 (0.000) | 1.877 (0.000) | 1.269 (0.058) | 1.122 (0.052) | 1.988 (0.117) | 0.364 (0.355) | 0.322 (0.314) | 0.301 (0.493) |
| CSDI | | | | 1.191 (0.000) | 1.054 (0.000) | 1.669 (0.000) | 1.222 (0.071) | 1.081 (0.062) | 1.861 (0.157) | 0.522 (0.350) | 0.462 (0.350) | 0.500 (0.545) |
| US-GAN | | | | 0.871 (0.000) | 0.770 (0.000) | 0.953 (0.000) | 1.122 (0.033) | 0.992 (0.029) | 1.558 (0.067) | 0.161 (0.053) | 0.142 (0.047) | 0.044 (0.025) |
| GP-VAE | | | | 0.942 (0.000) | 0.833 (0.000) | 1.131 (0.000) | 1.041 (0.096) | 0.921 (0.085) | 1.393 (0.182) | 0.334 (0.319) | 0.295 (0.282) | 0.253 (0.405) |
| Mean | | | | 1.289 (0.000) | 1.140 (0.000) | 1.883 (0.000) | 1.450 (0.124) | 1.283 (0.110) | 2.538 (0.343) | 0.388 (0.325) | 0.344 (0.288) | 0.295 (0.444) |
| Median | | | | 1.418 (0.129) | 1.254 (0.114) | 2.294 (0.411) | 1.487 (0.141) | 1.315 (0.125) | 2.682 (0.417) | 0.438 (0.339) | 0.387 (0.299) | 0.355 (0.455) |
| LOCF | | | | 1.390 (0.112) | 1.230 (0.099) | 2.264 (0.338) | 1.415 (0.156) | 1.252 (0.138) | 2.449 (0.478) | 0.430 (0.328) | 0.381 (0.290) | 0.345 (0.448) |
| Linear | | | | 1.353 (0.116) | 1.197 (0.103) | 2.152 (0.352) | 1.392 (0.145) | 1.231 (0.128) | 2.372 (0.440) | 0.431 (0.331) | 0.381 (0.293) | 0.347 (0.463) |

**ETT_h1 (subseq_50%)**

| Model | MAE wt XGB | MRE wt XGB | MSE wt XGB | MAE w XGB | MRE w XGB | MSE w XGB | MAE w RNN | MRE w RNN | MSE w RNN | MAE w Transformer | MRE w Transformer | MSE w Transformer |
|---|---|---|---|---|---|---|---|---|---|---|---|---|
| iTransformer | | | | 1.138 (0.000) | 1.006 (0.000) | 1.702 (0.000) | 1.314 (0.053) | 1.162 (0.046) | 2.163 (0.116) | 0.340 (0.355) | 0.301 (0.314) | 0.284 (0.476) |
| SAITS | | | | 1.108 (0.000) | 0.980 (0.000) | 1.583 (0.000) | 1.293 (0.043) | 1.143 (0.038) | 2.093 (0.075) | 0.336 (0.375) | 0.297 (0.332) | 0.296 (0.518) |
| Nonstationary | | | | 1.228 (0.000) | 1.086 (0.000) | 1.868 (0.000) | 1.369 (0.052) | 1.211 (0.046) | 2.279 (0.118) | 0.341 (0.433) | 0.301 (0.383) | 0.342 (0.630) |
| ETSformer | | | | 1.073 (0.000) | 0.949 (0.000) | 1.489 (0.000) | 1.282 (0.047) | 1.133 (0.042) | 2.033 (0.091) | 0.335 (0.377) | 0.296 (0.334) | 0.297 (0.527) |
| PatchTST | | | | 1.301 (0.000) | 1.151 (0.000) | 2.215 (0.000) | 1.318 (0.049) | 1.166 (0.043) | 2.172 (0.101) | 0.345 (0.374) | 0.305 (0.331) | 0.303 (0.521) |
| Crossformer | | | | 1.097 (0.000) | 0.971 (0.000) | 1.519 (0.000) | 1.309 (0.048) | 1.158 (0.044) | 2.147 (0.095) | 0.342 (0.378) | 0.303 (0.335) | 0.308 (0.530) |
| Informer | | | | 1.187 (0.000) | 1.050 (0.000) | 1.749 (0.000) | 1.295 (0.050) | 1.145 (0.044) | 2.096 (0.093) | 0.329 (0.347) | 0.291 (0.306) | 0.271 (0.457) |
| Autoformer | | | | 1.359 (0.000) | 1.202 (0.000) | 2.200 (0.000) | 1.264 (0.041) | 1.118 (0.036) | 1.993 (0.085) | 0.317 (0.352) | 0.281 (0.312) | 0.264 (0.460) |
| Pyraformer | | | | 1.089 (0.000) | 0.963 (0.000) | 1.588 (0.000) | 1.278 (0.049) | 1.130 (0.044) | 2.045 (0.092) | 0.337 (0.365) | 0.298 (0.323) | 0.291 (0.499) |
| Transformer | | | | 1.033 (0.000) | 0.914 (0.000) | 1.300 (0.000) | 1.298 (0.046) | 1.148 (0.040) | 2.098 (0.083) | 0.346 (0.353) | 0.306 (0.312) | 0.292 (0.482) |
| BRITS | | | | 0.991 (0.000) | 0.876 (0.000) | 1.278 (0.000) | 1.143 (0.068) | 1.010 (0.060) | 1.695 (0.108) | 0.314 (0.334) | 0.278 (0.295) | 0.261 (0.446) |
| MRNN | | | | 1.026 (0.000) | 0.907 (0.000) | 1.423 (0.000) | 1.206 (0.050) | 1.067 (0.044) | 1.837 (0.110) | 0.319 (0.351) | 0.282 (0.310) | 0.258 (0.446) |
| GRUD | 1.398 (0.000) | 1.237 (0.000) | 2.308 (0.000) | 1.037 (0.000) | 0.917 (0.000) | 1.328 (0.000) | 1.239 (0.052) | 1.096 (0.046) | 1.927 (0.104) | 0.350 (0.357) | 0.309 (0.316) | 0.298 (0.502) |
| TimesNet | | | | 1.030 (0.000) | 0.911 (0.000) | 1.376 (0.000) | 1.290 (0.052) | 1.141 (0.046) | 2.106 (0.105) | 0.337 (0.372) | 0.298 (0.330) | 0.299 (0.517) |
| MICN | | | | 1.121 (0.000) | 0.992 (0.000) | 1.563 (0.000) | 1.302 (0.046) | 1.151 (0.041) | 2.106 (0.103) | 0.338 (0.386) | 0.298 (0.342) | 0.304 (0.535) |
| SCINet | | | | 1.129 (0.000) | 0.998 (0.000) | 1.581 (0.000) | 1.325 (0.043) | 1.172 (0.038) | 2.184 (0.082) | 0.374 (0.391) | 0.331 (0.346) | 0.339 (0.582) |
| StemGNN | | | | 1.201 (0.000) | 1.063 (0.000) | 1.741 (0.000) | 1.289 (0.048) | 1.140 (0.043) | 2.079 (0.092) | 0.357 (0.373) | 0.316 (0.330) | 0.312 (0.530) |
| FreTS | | | | 1.106 (0.000) | 0.979 (0.000) | 1.499 (0.000) | 1.306 (0.049) | 1.155 (0.043) | 2.122 (0.098) | 0.352 (0.379) | 0.311 (0.335) | 0.312 (0.533) |
| Koopa | | | | 1.282 (0.000) | 1.134 (0.000) | 2.098 (0.000) | 1.302 (0.051) | 1.151 (0.045) | 2.118 (0.100) | 0.341 (0.402) | 0.301 (0.356) | 0.323 (0.577) |
| DLinear | | | | 1.184 (0.000) | 1.047 (0.000) | 1.794 (0.000) | 1.315 (0.049) | 1.163 (0.043) | 2.162 (0.101) | 0.346 (0.382) | 0.306 (0.338) | 0.310 (0.533) |
| FILM | | | | 1.393 (0.000) | 1.232 (0.000) | 2.318 (0.000) | 1.306 (0.040) | 1.155 (0.035) | 2.130 (0.076) | 0.343 (0.390) | 0.303 (0.345) | 0.307 (0.541) |
| CSDI | | | | 1.214 (0.000) | 1.073 (0.000) | 1.722 (0.000) | 1.274 (0.048) | 1.127 (0.042) | 2.038 (0.086) | 0.352 (0.379) | 0.311 (0.335) | 0.312 (0.533) |
| US-GAN | | | | 1.011 (0.000) | 0.894 (0.000) | 1.293 (0.000) | 1.211 (0.062) | 1.071 (0.055) | 1.872 (0.107) | 0.329 (0.350) | 0.291 (0.310) | 0.278 (0.474) |
| GP-VAE | | | | 1.110 (0.000) | 0.982 (0.000) | 1.722 (0.000) | 1.246 (0.054) | 1.102 (0.048) | 1.968 (0.089) | 0.360 (0.393) | 0.319 (0.347) | 0.334 (0.577) |
| Mean | | | | 1.239 (0.000) | 1.096 (0.000) | 1.847 (0.000) | 1.540 (0.085) | 1.362 (0.075) | 2.815 (0.243) | 0.375 (0.470) | 0.332 (0.416) | 0.394 (0.724) |
| Median | | | | 1.346 (0.107) | 1.190 (0.095) | 2.167 (0.320) | 1.561 (0.097) | 1.381 (0.086) | 2.894 (0.298) | 0.384 (0.465) | 0.339 (0.411) | 0.396 (0.722) |
| LOCF | | | | 1.327 (0.091) | 1.174 (0.081) | 2.152 (0.262) | 1.474 (0.149) | 1.303 (0.132) | 2.610 (0.474) | 0.373 (0.427) | 0.330 (0.378) | 0.355 (0.644) |
| Linear | | | | 1.297 (0.095) | 1.147 (0.084) | 2.051 (0.286) | 1.436 (0.147) | 1.270 (0.130) | 2.503 (0.453) | 0.364 (0.417) | 0.322 (0.369) | 0.341 (0.616) |

Table 22: Performance comparison for the forecasting task on ETT_h1 datasets with 10% point missing and 50% point missing.

**ETT_h1 (point_10%)**

| | MAE wt XGB | MRE wt XGB | MSE wt XGB | MAE w XGB | MRE w XGB | MSE w XGB | MAE w RNN | MRE w RNN | MSE w RNN | MAE w Transformer | MRE w Transformer | MSE w Transformer |
|---|---|---|---|---|---|---|---|---|---|---|---|---|
| iTransformer | | | | 1.114 (0.000) | 0.985 (0.000) | 1.453 (0.000) | 1.266 (0.065) | 1.119 (0.057) | 2.002 (0.132) | 0.376 (0.343) | 0.332 (0.303) | 0.305 (0.478) |
| SAITS | | | | 1.146 (0.000) | 1.014 (0.000) | 1.503 (0.000) | 1.268 (0.065) | 1.121 (0.057) | 2.012 (0.131) | 0.383 (0.332) | 0.352 (0.294) | 0.315 (0.469) |
| Nonstationary | | | | 1.137 (0.000) | 1.006 (0.000) | 1.508 (0.000) | 1.287 (0.063) | 1.138 (0.056) | 2.059 (0.127) | 0.383 (0.348) | 0.339 (0.308) | 0.317 (0.490) |
| ETSformer | | | | 1.117 (0.000) | 0.988 (0.000) | 1.459 (0.000) | 1.266 (0.063) | 1.120 (0.056) | 2.002 (0.128) | 0.398 (0.332) | 0.352 (0.294) | 0.317 (0.468) |
| PatchTST | | | | 1.151 (0.000) | 1.018 (0.000) | 1.558 (0.000) | 1.272 (0.062) | 1.125 (0.055) | 2.022 (0.125) | 0.385 (0.337) | 0.340 (0.298) | 0.309 (0.473) |
| Crossformer | | | | 1.130 (0.000) | 0.999 (0.000) | 1.494 (0.000) | 1.270 (0.063) | 1.123 (0.056) | 2.016 (0.127) | 0.390 (0.335) | 0.345 (0.296) | 0.312 (0.470) |
| Informer | | | | 1.089 (0.000) | 0.963 (0.000) | 1.396 (0.000) | 1.267 (0.064) | 1.121 (0.057) | 2.008 (0.129) | 0.386 (0.341) | 0.342 (0.301) | 0.312 (0.474) |
| Autoformer | | | | 1.121 (0.000) | 0.991 (0.000) | 1.467 (0.000) | 1.270 (0.062) | 1.123 (0.055) | 2.014 (0.125) | 0.395 (0.334) | 0.349 (0.295) | 0.318 (0.470) |
| Pyraformer | | | | 1.104 (0.000) | 0.977 (0.000) | 1.439 (0.000) | 1.271 (0.063) | 1.124 (0.055) | 2.018 (0.126) | 0.390 (0.338) | 0.345 (0.299) | 0.314 (0.474) |
| Transformer | | | | 1.092 (0.000) | 0.966 (0.000) | 1.368 (0.000) | 1.268 (0.063) | 1.121 (0.056) | 2.009 (0.127) | 0.397 (0.334) | 0.351 (0.295) | 0.318 (0.469) |
| BRITS | | | | 1.063 (0.000) | 0.940 (0.000) | 1.299 (0.000) | 1.266 (0.064) | 1.119 (0.056) | 2.008 (0.129) | 0.406 (0.329) | 0.359 (0.291) | 0.322 (0.466) |
| MRNN | | | | 1.051 (0.000) | 0.930 (0.000) | 1.279 (0.000) | 1.228 (0.071) | 1.086 (0.063) | 1.894 (0.146) | 0.394 (0.334) | 0.348 (0.295) | 0.312 (0.473) |
| GRUD | | | | 1.131 (0.000) | 1.001 (0.000) | 1.467 (0.000) | 1.260 (0.064) | 1.114 (0.056) | 1.983 (0.130) | 0.388 (0.339) | 0.344 (0.300) | 0.312 (0.475) |
| TimesNet | 1.137 (0.000) | 1.005 (0.000) | 1.488 (0.000) | 1.111 (0.000) | 0.983 (0.000) | 1.425 (0.000) | 1.250 (0.063) | 1.105 (0.056) | 1.956 (0.125) | 0.396 (0.334) | 0.350 (0.295) | 0.317 (0.472) |
| MICN | | | | 1.131 (0.000) | 1.000 (0.000) | 1.509 (0.000) | 1.266 (0.065) | 1.120 (0.057) | 2.000 (0.132) | 0.400 (0.332) | 0.353 (0.294) | 0.318 (0.472) |
| SCINet | | | | 1.139 (0.000) | 1.007 (0.000) | 1.513 (0.000) | 1.269 (0.063) | 1.123 (0.056) | 2.016 (0.127) | 0.387 (0.335) | 0.342 (0.297) | 0.310 (0.471) |
| StemGNN | | | | 1.122 (0.000) | 0.992 (0.000) | 1.507 (0.000) | 1.269 (0.064) | 1.122 (0.056) | 2.013 (0.129) | 0.394 (0.332) | 0.349 (0.294) | 0.314 (0.470) |
| FreTS | | | | 1.135 (0.000) | 1.004 (0.000) | 1.507 (0.000) | 1.274 (0.063) | 1.127 (0.055) | 2.030 (0.127) | 0.385 (0.338) | 0.341 (0.299) | 0.310 (0.474) |
| Koopa | | | | 1.165 (0.000) | 1.030 (0.000) | 1.581 (0.000) | 1.263 (0.064) | 1.117 (0.055) | 1.991 (0.129) | 0.376 (0.344) | 0.333 (0.304) | 0.306 (0.479) |
| DLinear | | | | 1.135 (0.000) | 1.004 (0.000) | 1.504 (0.000) | 1.272 (0.062) | 1.125 (0.055) | 2.021 (0.126) | 0.394 (0.332) | 0.348 (0.293) | 0.313 (0.468) |
| FILM | | | | 1.147 (0.000) | 1.014 (0.000) | 1.589 (0.000) | 1.269 (0.061) | 1.122 (0.054) | 2.009 (0.122) | 0.387 (0.355) | 0.342 (0.314) | 0.321 (0.507) |
| CSDI | | | | 1.085 (0.000) | 0.960 (0.000) | 1.356 (0.000) | 1.266 (0.064) | 1.120 (0.056) | 2.009 (0.129) | 0.398 (0.331) | 0.352 (0.293) | 0.317 (0.466) |
| US-GAN | | | | 1.086 (0.000) | 0.960 (0.000) | 1.363 (0.000) | 1.272 (0.062) | 1.125 (0.055) | 2.026 (0.127) | 0.418 (0.329) | 0.370 (0.291) | 0.336 (0.471) |
| GP-VAE | | | | 1.146 (0.000) | 1.013 (0.000) | 1.522 (0.000) | 1.264 (0.066) | 1.118 (0.058) | 1.999 (0.134) | 0.386 (0.342) | 0.341 (0.303) | 0.312 (0.484) |
| Mean | | | | 1.112 (0.000) | 0.984 (0.000) | 1.423 (0.000) | 1.271 (0.070) | 1.125 (0.062) | 2.013 (0.148) | 0.420 (0.329) | 0.371 (0.291) | 0.330 (0.484) |
| Median | | | | 1.114 (0.002) | 0.986 (0.002) | 1.434 (0.011) | 1.273 (0.072) | 1.126 (0.064) | 2.018 (0.153) | 0.411 (0.333) | 0.363 (0.295) | 0.325 (0.485) |
| LOCF | | | | 1.118 (0.005) | 0.989 (0.005) | 1.452 (0.026) | 1.271 (0.070) | 1.124 (0.062) | 2.013 (0.147) | 0.404 (0.336) | 0.357 (0.297) | 0.322 (0.483) |
| Linear | | | | 1.124 (0.012) | 0.995 (0.011) | 1.466 (0.033) | 1.270 (0.069) | 1.122 (0.061) | 2.013 (0.143) | 0.404 (0.335) | 0.357 (0.296) | 0.323 (0.480) |

**ETT_h1 (point_50%)**

| | MAE wt XGB | MRE wt XGB | MSE wt XGB | MAE w XGB | MRE w XGB | MSE w XGB | MAE w RNN | MRE w RNN | MSE w RNN | MAE w Transformer | MRE w Transformer | MSE w Transformer |
|---|---|---|---|---|---|---|---|---|---|---|---|---|
| iTransformer | | | | 1.223 (0.000) | 1.082 (0.000) | 1.683 (0.000) | 1.240 (0.071) | 1.096 (0.062) | 1.937 (0.145) | 0.379 (0.337) | 0.335 (0.298) | 0.303 (0.476) |
| SAITS | | | | 1.156 (0.000) | 1.023 (0.000) | 1.616 (0.000) | 1.274 (0.062) | 1.126 (0.055) | 2.022 (0.127) | 0.375 (0.341) | 0.332 (0.301) | 0.306 (0.474) |
| Nonstationary | | | | 1.196 (0.000) | 1.058 (0.000) | 1.657 (0.000) | 1.317 (0.065) | 1.164 (0.057) | 2.119 (0.134) | 0.378 (0.369) | 0.334 (0.326) | 0.323 (0.530) |
| ETSformer | | | | 1.205 (0.000) | 1.066 (0.000) | 1.835 (0.000) | 1.235 (0.061) | 1.092 (0.054) | 1.881 (0.116) | 0.321 (0.343) | 0.284 (0.303) | 0.259 (0.449) |
| PatchTST | | | | 1.192 (0.000) | 1.054 (0.000) | 1.651 (0.000) | 1.264 (0.058) | 1.118 (0.051) | 2.002 (0.113) | 0.370 (0.331) | 0.327 (0.292) | 0.292 (0.454) |
| Crossformer | | | | 1.180 (0.000) | 1.044 (0.000) | 1.502 (0.000) | 1.268 (0.062) | 1.122 (0.052) | 2.007 (0.126) | 0.377 (0.324) | 0.334 (0.287) | 0.294 (0.449) |
| Informer | | | | 1.128 (0.000) | 0.998 (0.000) | 1.502 (0.000) | 1.269 (0.059) | 1.122 (0.052) | 1.990 (0.118) | 0.384 (0.347) | 0.340 (0.307) | 0.314 (0.490) |
| Autoformer | | | | 1.306 (0.000) | 1.155 (0.000) | 1.948 (0.000) | 1.287 (0.177) | 1.138 (0.157) | 2.069 (0.454) | 0.559 (0.102) | 0.494 (0.090) | 0.402 (0.129) |
| Pyraformer | | | | 1.120 (0.000) | 0.990 (0.000) | 1.442 (0.000) | 1.247 (0.055) | 1.103 (0.049) | 1.919 (0.106) | 0.374 (0.337) | 0.331 (0.298) | 0.297 (0.469) |
| Transformer | | | | 1.177 (0.000) | 1.041 (0.000) | 1.634 (0.000) | 1.243 (0.057) | 1.100 (0.051) | 1.923 (0.110) | 0.375 (0.334) | 0.332 (0.295) | 0.297 (0.464) |
| BRITS | | | | 1.220 (0.000) | 1.079 (0.000) | 1.764 (0.000) | 1.254 (0.059) | 1.109 (0.052) | 1.961 (0.117) | 0.399 (0.332) | 0.353 (0.293) | 0.319 (0.473) |
| MRNN | | | | 0.984 (0.000) | 0.870 (0.000) | 1.179 (0.000) | 1.081 (0.089) | 0.956 (0.079) | 1.497 (0.175) | 0.445 (0.296) | 0.394 (0.261) | 0.337 (0.347) |
| GRUD | | | | 1.210 (0.000) | 1.070 (0.000) | 1.828 (0.000) | 1.215 (0.057) | 1.075 (0.051) | 1.810 (0.115) | 0.359 (0.340) | 0.318 (0.301) | 0.286 (0.461) |
| TimesNet | 1.350 (0.000) | 1.194 (0.000) | 2.100 (0.000) | 1.143 (0.000) | 1.011 (0.000) | 1.486 (0.000) | 1.202 (0.063) | 1.063 (0.056) | 1.806 (0.119) | 0.365 (0.329) | 0.323 (0.291) | 0.290 (0.447) |
| MICN | | | | 1.130 (0.000) | 0.999 (0.000) | 1.495 (0.000) | 1.178 (0.081) | 1.042 (0.072) | 1.716 (0.163) | 0.343 (0.355) | 0.304 (0.315) | 0.283 (0.477) |
| SCINet | | | | 1.095 (0.000) | 0.968 (0.000) | 1.407 (0.000) | 1.254 (0.063) | 1.109 (0.056) | 1.975 (0.125) | 0.372 (0.339) | 0.329 (0.299) | 0.300 (0.473) |
| StemGNN | | | | 1.113 (0.000) | 0.985 (0.000) | 1.437 (0.000) | 1.259 (0.057) | 1.113 (0.052) | 1.967 (0.115) | 0.362 (0.346) | 0.320 (0.306) | 0.294 (0.477) |
| FreTS | | | | 1.173 (0.000) | 1.038 (0.000) | 1.587 (0.000) | 1.259 (0.059) | 1.114 (0.052) | 1.984 (0.118) | 0.367 (0.336) | 0.325 (0.297) | 0.293 (0.463) |
| Koopa | | | | 1.288 (0.000) | 1.139 (0.000) | 1.855 (0.000) | 1.265 (0.068) | 1.119 (0.060) | 1.973 (0.138) | 0.357 (0.356) | 0.316 (0.315) | 0.297 (0.494) |
| DLinear | | | | 1.176 (0.000) | 1.040 (0.000) | 1.616 (0.000) | 1.262 (0.054) | 1.116 (0.048) | 1.981 (0.101) | 0.368 (0.336) | 0.326 (0.297) | 0.295 (0.466) |
| FILM | | | | 1.215 (0.000) | 1.075 (0.000) | 1.686 (0.000) | 1.283 (0.062) | 1.135 (0.055) | 2.054 (0.129) | 0.376 (0.376) | 0.333 (0.332) | 0.319 (0.537) |
| CSDI | | | | 1.183 (0.000) | 1.047 (0.000) | 1.641 (0.000) | 1.269 (0.063) | 1.122 (0.056) | 1.995 (0.134) | 0.362 (0.337) | 0.321 (0.298) | 0.290 (0.459) |
| US-GAN | | | | 1.083 (0.000) | 0.958 (0.000) | 1.414 (0.000) | 1.114 (0.076) | 0.985 (0.067) | 1.603 (0.172) | 0.352 (0.267) | 0.311 (0.236) | 0.240 (0.336) |
| GP-VAE | | | | 1.170 (0.000) | 1.035 (0.000) | 1.643 (0.000) | 1.169 (0.071) | 1.034 (0.063) | 1.738 (0.139) | 0.344 (0.349) | 0.304 (0.309) | 0.282 (0.468) |
| Mean | | | | 1.472 (0.000) | 1.302 (0.000) | 2.410 (0.000) | 1.368 (0.147) | 1.210 (0.130) | 2.237 (0.368) | 0.677 (0.346) | 0.599 (0.306) | 0.645 (0.481) |
| Median | | | | 1.498 (0.026) | 1.325 (0.023) | 2.493 (0.083) | 1.380 (0.147) | 1.221 (0.130) | 2.281 (0.372) | 0.588 (0.404) | 0.520 (0.357) | 0.563 (0.523) |
| LOCF | | | | 1.411 (0.126) | 1.248 (0.112) | 2.270 (0.323) | 1.327 (0.146) | 1.174 (0.129) | 2.143 (0.368) | 0.523 (0.392) | 0.463 (0.347) | 0.480 (0.519) |
| Linear | | | | 1.356 (0.144) | 1.200 (0.127) | 2.118 (0.384) | 1.306 (0.136) | 1.155 (0.120) | 2.093 (0.337) | 0.492 (0.382) | 0.435 (0.338) | 0.440 (0.514) |

Table 23: Performance comparison for the forecasting task on PeMS datasets with 50% point missing, 50% block missing and 50% subsequence missing.

**PeMS (point 50%)**

| Model | MAE w XGB | MRE w XGB | MSE w XGB | MAE w RNN | MRE w RNN | MSE w RNN | MAE w Transformer | MRE w Transformer | MSE w Transformer |
|---|---|---|---|---|---|---|---|---|---|
| iTransformer | 1.000 (0.000) | 1.001 (0.000) | 1.309 (0.000) | 0.574 (0.020) | 0.575 (0.020) | 0.636 (0.039) | 0.647 (0.097) | 0.648 (0.097) | 0.703 (0.168) |
| SAITS | 0.998 (0.000) | 0.999 (0.000) | 1.322 (0.000) | 0.550 (0.030) | 0.551 (0.030) | 0.588 (0.049) | 0.611 (0.063) | 0.612 (0.063) | 0.622 (0.085) |
| Nonstationary | 1.019 (0.000) | 1.020 (0.000) | 1.357 (0.000) | 0.550 (0.031) | 0.550 (0.031) | 0.609 (0.055) | 0.688 (0.067) | 0.689 (0.067) | 0.765 (0.119) |
| ETSformer | 1.018 (0.000) | 1.018 (0.000) | 1.336 (0.000) | 0.580 (0.041) | 0.580 (0.041) | 0.640 (0.074) | 0.861 (0.243) | 0.862 (0.243) | 1.175 (0.698) |
| PatchTST | 0.975 (0.000) | 0.976 (0.000) | 1.254 (0.000) | 0.560 (0.029) | 0.560 (0.029) | 0.619 (0.047) | 0.845 (0.216) | 0.846 (0.217) | 1.171 (0.612) |
| Crossformer | 1.000 (0.000) | 1.001 (0.000) | 1.340 (0.000) | 0.545 (0.034) | 0.545 (0.034) | 0.598 (0.059) | 0.797 (0.285) | 0.798 (0.285) | 1.096 (0.766) |
| Informer | 1.017 (0.000) | 1.018 (0.000) | 1.345 (0.000) | 0.553 (0.023) | 0.554 (0.023) | 0.592 (0.037) | 0.671 (0.060) | 0.671 (0.060) | 0.753 (0.101) |
| Autoformer | 1.114 (0.000) | 1.115 (0.000) | 1.600 (0.000) | 0.673 (0.023) | 0.674 (0.023) | 0.745 (0.050) | 0.806 (0.255) | 0.807 (0.255) | 1.142 (0.694) |
| Pyraformer | 0.975 (0.000) | 0.976 (0.000) | 1.600 (0.000) | 0.556 (0.000) | 0.557 (0.000) | 0.654 (0.081) | 0.765 (0.353) | 0.766 (0.353) | 1.062 (0.958) |
| Transformer | 1.044 (0.000) | 1.045 (0.000) | 1.521 (0.000) | 0.537 (0.025) | 0.537 (0.025) | 0.576 (0.036) | 0.839 (0.238) | 0.840 (0.238) | 1.166 (0.628) |
| BRITS | 1.046 (0.000) | 1.047 (0.000) | 1.436 (0.000) | 0.526 (0.034) | 0.527 (0.034) | 0.564 (0.060) | 0.798 (0.147) | 0.799 (0.147) | 1.036 (0.378) |
| MRNN | 1.040 (0.000) | 1.041 (0.000) | 1.449 (0.000) | 0.545 (0.012) | 0.546 (0.012) | 0.598 (0.016) | 0.656 (0.248) | 0.657 (0.249) | 0.807 (0.606) |
| GRUD | 1.012 (0.000) | 1.013 (0.000) | 1.337 (0.000) | 0.531 (0.039) | 0.531 (0.039) | 0.566 (0.064) | 0.680 (0.074) | 0.681 (0.074) | 0.824 (0.226) |
| TimesNet | 1.025 (0.000) | 1.026 (0.000) | 1.385 (0.000) | 0.532 (0.022) | 0.532 (0.022) | 0.576 (0.046) | 0.666 (0.108) | 0.667 (0.108) | 0.729 (0.177) |
| MICN | 1.132 (0.000) | 1.133 (0.000) | 1.647 (0.000) | 0.653 (0.026) | 0.654 (0.026) | 0.743 (0.061) | 0.737 (0.125) | 0.738 (0.125) | 0.838 (0.270) |
| SCINet | 0.973 (0.000) | 0.974 (0.000) | 1.258 (0.000) | 0.622 (0.018) | 0.623 (0.018) | 0.664 (0.032) | 0.823 (0.190) | 0.824 (0.191) | 1.097 (0.490) |
| StemGNN | 1.066 (0.000) | 1.067 (0.000) | 1.629 (0.000) | 0.613 (0.029) | 0.613 (0.029) | 0.681 (0.046) | 0.942 (0.308) | 0.943 (0.309) | 1.464 (0.856) |
| FreTS | 0.998 (0.000) | 0.998 (0.000) | 1.255 (0.000) | 0.565 (0.037) | 0.565 (0.037) | 0.612 (0.067) | 0.810 (0.165) | 0.810 (0.165) | 1.036 (0.354) |
| Koopa | 1.036 (0.000) | 1.036 (0.000) | 1.407 (0.000) | 0.640 (0.024) | 0.640 (0.024) | 0.676 (0.052) | 0.632 (0.038) | 0.633 (0.038) | 0.672 (0.086) |
| DLinear | 1.029 (0.000) | 1.030 (0.000) | 1.450 (0.000) | 0.553 (0.030) | 0.554 (0.030) | 0.610 (0.037) | 0.878 (0.256) | 0.879 (0.256) | 1.325 (0.682) |
| FILM | 1.047 (0.000) | 1.048 (0.000) | 1.265 (0.000) | 0.538 (0.017) | 0.538 (0.017) | 0.512 (0.028) | 0.850 (0.351) | 0.851 (0.351) | 1.238 (0.951) |
| CSDI | 1.057 (0.000) | 1.058 (0.000) | 1.446 (0.000) | 0.553 (0.031) | 0.553 (0.031) | 0.613 (0.052) | 0.779 (0.276) | 0.780 (0.276) | 1.033 (0.747) |
| US-GAN | 1.038 (0.000) | 1.039 (0.000) | 1.507 (0.000) | 0.507 (0.030) | 0.507 (0.030) | 0.607 (0.037) | 0.972 (0.299) | 0.973 (0.299) | 1.549 (0.784) |
| GP-VAE | 1.047 (0.000) | 1.048 (0.000) | 1.488 (0.000) | 0.539 (0.042) | 0.539 (0.042) | 0.591 (0.072) | 0.792 (0.280) | 0.793 (0.280) | 1.073 (0.714) |
| Mean | 1.050 (0.000) | 1.051 (0.000) | 1.460 (0.000) | 0.501 (0.044) | 0.501 (0.044) | 0.513 (0.078) | 0.740 (0.090) | 0.741 (0.090) | 0.881 (0.193) |
| Median | 1.081 (0.031) | 1.082 (0.031) | 1.538 (0.077) | 0.519 (0.040) | 0.519 (0.040) | 0.546 (0.073) | 0.756 (0.193) | 0.757 (0.193) | 0.986 (0.482) |
| LOCF | 1.088 (0.027) | 1.089 (0.027) | 1.530 (0.064) | 0.529 (0.040) | 0.530 (0.040) | 0.562 (0.070) | 0.716 (0.178) | 0.716 (0.178) | 0.887 (0.435) |
| Linear | 1.061 (0.052) | 1.062 (0.052) | 1.465 (0.127) | 0.527 (0.036) | 0.528 (0.036) | 0.557 (0.065) | 0.724 (0.198) | 0.724 (0.199) | 0.899 (0.496) |

(MAE w/t XGB 1.084 (0.000); MRE w/t XGB 1.085 (0.000); MSE w/t XGB 1.514 (0.000))

**PeMS (block 50%)**

| Model | MAE w XGB | MRE w XGB | MSE w XGB | MAE w RNN | MRE w RNN | MSE w RNN | MAE w Transformer | MRE w Transformer | MSE w Transformer |
|---|---|---|---|---|---|---|---|---|---|
| iTransformer | 1.088 (0.000) | 1.089 (0.000) | 1.598 (0.000) | 0.598 (0.027) | 0.598 (0.027) | 0.691 (0.052) | 0.903 (0.264) | 0.904 (0.264) | 1.308 (0.724) |
| SAITS | 1.024 (0.000) | 1.024 (0.000) | 1.404 (0.000) | 0.555 (0.024) | 0.555 (0.024) | 0.610 (0.039) | 0.984 (0.335) | 0.985 (0.335) | 1.598 (0.860) |
| Nonstationary | 0.990 (0.000) | 0.991 (0.000) | 1.291 (0.000) | 0.544 (0.018) | 0.544 (0.018) | 0.592 (0.035) | 0.861 (0.260) | 0.861 (0.261) | 1.243 (0.633) |
| ETSformer | 1.039 (0.000) | 1.040 (0.000) | 1.393 (0.000) | 0.578 (0.021) | 0.578 (0.021) | 0.638 (0.042) | 0.758 (0.092) | 0.759 (0.092) | 0.894 (0.155) |
| PatchTST | 1.004 (0.000) | 1.005 (0.000) | 1.310 (0.000) | 0.548 (0.021) | 0.548 (0.021) | 0.587 (0.035) | 0.917 (0.267) | 0.918 (0.267) | 1.339 (0.796) |
| Crossformer | 1.002 (0.000) | 1.003 (0.000) | 1.333 (0.000) | 0.549 (0.011) | 0.549 (0.011) | 0.596 (0.012) | 0.735 (0.202) | 0.736 (0.202) | 0.932 (0.537) |
| Informer | 1.065 (0.000) | 1.066 (0.000) | 1.510 (0.000) | 0.551 (0.025) | 0.551 (0.025) | 0.604 (0.043) | 0.919 (0.360) | 0.920 (0.360) | 1.445 (0.340) |
| Autoformer | 1.174 (0.000) | 1.175 (0.000) | 1.856 (0.000) | 0.657 (0.016) | 0.658 (0.016) | 0.762 (0.020) | 0.912 (0.201) | 0.913 (0.201) | 1.309 (0.525) |
| Pyraformer | 1.012 (0.000) | 1.013 (0.000) | 1.338 (0.000) | 0.550 (0.021) | 0.550 (0.021) | 0.607 (0.039) | 0.921 (0.336) | 0.922 (0.336) | 1.454 (0.890) |
| Transformer | 1.135 (0.000) | 1.137 (0.000) | 1.786 (0.000) | 0.561 (0.021) | 0.562 (0.021) | 0.623 (0.044) | 0.689 (0.122) | 0.690 (0.122) | 0.775 (0.219) |
| BRITS | 1.033 (0.000) | 1.034 (0.000) | 1.403 (0.000) | 0.529 (0.022) | 0.529 (0.022) | 0.571 (0.037) | 0.796 (0.293) | 0.796 (0.293) | 1.144 (0.764) |
| MRNN | 1.012 (0.000) | 1.013 (0.000) | 1.368 (0.000) | 0.551 (0.038) | 0.551 (0.038) | 0.590 (0.054) | 0.754 (0.249) | 0.755 (0.250) | 0.985 (0.613) |
| GRUD | 1.048 (0.000) | 1.048 (0.000) | 1.402 (0.000) | 0.522 (0.014) | 0.522 (0.014) | 0.556 (0.028) | 0.926 (0.341) | 0.927 (0.342) | 1.440 (0.900) |
| TimesNet | 1.058 (0.000) | 1.059 (0.000) | 1.467 (0.000) | 0.538 (0.024) | 0.539 (0.024) | 0.585 (0.043) | 0.848 (0.300) | 0.848 (0.300) | 1.249 (0.773) |
| MICN | 1.150 (0.000) | 1.151 (0.000) | 1.731 (0.000) | 0.663 (0.030) | 0.663 (0.030) | 0.819 (0.048) | 1.066 (0.325) | 1.066 (0.326) | 1.847 (0.911) |
| SCINet | 1.048 (0.000) | 1.049 (0.000) | 1.456 (0.000) | 0.666 (0.017) | 0.666 (0.017) | 0.740 (0.032) | 0.819 (0.175) | 0.819 (0.175) | 1.112 (0.383) |
| StemGNN | 1.011 (0.000) | 1.012 (0.000) | 1.397 (0.000) | 0.616 (0.026) | 0.617 (0.026) | 0.701 (0.054) | 0.879 (0.218) | 0.880 (0.218) | 1.229 (0.591) |
| FreTS | 1.010 (0.000) | 1.011 (0.000) | 1.323 (0.000) | 0.558 (0.014) | 0.559 (0.014) | 0.599 (0.027) | 0.987 (0.330) | 0.988 (0.330) | 1.629 (0.860) |
| Koopa | 0.969 (0.000) | 0.969 (0.000) | 1.248 (0.000) | 0.590 (0.034) | 0.591 (0.034) | 0.640 (0.063) | 0.719 (0.221) | 0.720 (0.222) | 0.912 (0.560) |
| DLinear | 1.079 (0.000) | 1.079 (0.000) | 1.554 (0.000) | 0.572 (0.028) | 0.573 (0.028) | 0.638 (0.052) | 0.903 (0.294) | 0.904 (0.294) | 1.411 (0.787) |
| FILM | 1.011 (0.000) | 1.012 (0.000) | 1.349 (0.000) | 0.503 (0.014) | 0.503 (0.014) | 0.534 (0.025) | 0.641 (0.104) | 0.642 (0.104) | 0.724 (0.155) |
| CSDI | 0.989 (0.000) | 0.990 (0.000) | 1.318 (0.000) | 0.639 (0.033) | 0.639 (0.033) | 0.745 (0.066) | 0.920 (0.361) | 0.920 (0.362) | 1.414 (0.957) |
| US-GAN | 1.038 (0.000) | 1.039 (0.000) | 1.425 (0.000) | 0.508 (0.021) | 0.509 (0.021) | 0.532 (0.046) | 1.014 (0.331) | 1.015 (0.332) | 1.692 (0.877) |
| GP-VAE | 1.025 (0.000) | 1.026 (0.000) | 1.367 (0.000) | 0.528 (0.027) | 0.529 (0.027) | 0.572 (0.050) | 0.630 (0.093) | 0.630 (0.093) | 0.650 (0.148) |
| Mean | 1.040 (0.000) | 1.040 (0.000) | 1.463 (0.000) | 0.470 (0.018) | 0.470 (0.018) | 0.445 (0.032) | 0.791 (0.276) | 0.792 (0.276) | 1.134 (0.662) |
| Median | 1.057 (0.018) | 1.058 (0.018) | 1.512 (0.049) | 0.503 (0.037) | 0.503 (0.037) | 0.494 (0.058) | 0.705 (0.215) | 0.706 (0.215) | 0.916 (0.527) |
| LOCF | 1.048 (0.020) | 1.049 (0.020) | 1.471 (0.071) | 0.548 (0.073) | 0.548 (0.073) | 0.569 (0.120) | 0.759 (0.272) | 0.760 (0.272) | 1.032 (0.658) |
| Linear | 1.046 (0.017) | 1.048 (0.017) | 1.463 (0.063) | 0.541 (0.064) | 0.541 (0.064) | 0.560 (0.105) | 0.728 (0.248) | 0.728 (0.248) | 0.899 (0.597) |

(MAE w/t XGB 1.068 (0.000); MRE w/t XGB 1.069 (0.000); MSE w/t XGB 1.507 (0.000))

**PeMS (subsequence 50%)**

| Model | MAE w XGB | MRE w XGB | MSE w XGB | MAE w RNN | MRE w RNN | MSE w RNN | MAE w Transformer | MRE w Transformer | MSE w Transformer |
|---|---|---|---|---|---|---|---|---|---|
| iTransformer | 1.013 (0.000) | 1.014 (0.000) | 1.402 (0.000) | 0.546 (0.009) | 0.546 (0.009) | 0.607 (0.016) | 1.030 (0.292) | 1.031 (0.292) | 1.641 (0.805) |
| SAITS | 1.038 (0.000) | 1.039 (0.000) | 1.420 (0.000) | 0.571 (0.020) | 0.572 (0.020) | 0.635 (0.033) | 0.802 (0.286) | 0.803 (0.286) | 1.095 (0.792) |
| Nonstationary | 1.038 (0.000) | 1.039 (0.000) | 1.290 (0.000) | 0.588 (0.040) | 0.588 (0.040) | 0.684 (0.084) | 0.803 (0.289) | 0.803 (0.289) | 0.834 (0.594) |
| ETSformer | 1.086 (0.000) | 1.087 (0.000) | 1.554 (0.000) | 0.595 (0.016) | 0.596 (0.016) | 0.683 (0.027) | 0.791 (0.236) | 0.792 (0.237) | 1.085 (0.635) |
| PatchTST | 1.061 (0.000) | 1.062 (0.000) | 1.496 (0.000) | 0.573 (0.027) | 0.574 (0.027) | 0.637 (0.052) | 0.791 (0.236) | 0.792 (0.237) | 1.086 (0.669) |
| Crossformer | 1.058 (0.000) | 1.059 (0.000) | 1.480 (0.000) | 0.576 (0.023) | 0.576 (0.023) | 0.642 (0.041) | 0.655 (0.056) | 0.655 (0.056) | 0.698 (0.096) |
| Informer | 1.028 (0.000) | 1.030 (0.000) | 1.388 (0.000) | 0.383 (0.018) | 0.383 (0.018) | 0.650 (0.029) | 0.769 (0.236) | 0.770 (0.236) | 1.007 (0.612) |
| Autoformer | 1.116 (0.000) | 1.117 (0.000) | 1.319 (0.000) | 0.597 (0.024) | 0.598 (0.024) | 0.747 (0.045) | 0.872 (0.311) | 0.873 (0.311) | 1.657 (0.794) |
| Pyraformer | 0.982 (0.000) | 0.983 (0.000) | 1.301 (0.000) | 0.565 (0.016) | 0.565 (0.016) | 0.623 (0.028) | 0.825 (0.270) | 0.825 (0.271) | 1.115 (0.736) |
| Transformer | 1.005 (0.000) | 1.006 (0.000) | 1.300 (0.000) | 0.582 (0.017) | 0.582 (0.017) | 0.654 (0.026) | 0.766 (0.270) | 0.766 (0.270) | 0.998 (0.666) |
| BRITS | 1.086 (0.000) | 1.087 (0.000) | 1.541 (0.000) | 0.567 (0.015) | 0.567 (0.015) | 0.567 (0.025) | 0.664 (0.078) | 0.665 (0.078) | 0.745 (0.125) |
| MRNN | 1.028 (0.000) | 1.029 (0.000) | 1.363 (0.000) | 0.610 (0.024) | 0.611 (0.024) | 0.611 (0.037) | 0.773 (0.064) | 0.774 (0.064) | 0.956 (0.165) |
| GRUD | 1.029 (0.000) | 1.030 (0.000) | 1.389 (0.000) | 0.553 (0.022) | 0.554 (0.022) | 0.554 (0.037) | 0.835 (0.334) | 0.836 (0.334) | 1.225 (0.919) |
| TimesNet | 1.067 (0.000) | 1.068 (0.000) | 1.450 (0.000) | 0.567 (0.019) | 0.568 (0.019) | 0.629 (0.034) | 0.784 (0.271) | 0.785 (0.271) | 1.044 (0.715) |
| MICN | 1.179 (0.000) | 1.180 (0.000) | 1.881 (0.000) | 0.654 (0.015) | 0.655 (0.015) | 0.826 (0.029) | 0.983 (0.346) | 0.983 (0.346) | 1.584 (0.976) |
| SCINet | 1.017 (0.000) | 1.018 (0.000) | 1.348 (0.000) | 0.645 (0.038) | 0.645 (0.038) | 0.728 (0.079) | 0.681 (0.048) | 0.681 (0.048) | 0.737 (0.103) |
| StemGNN | 1.008 (0.000) | 1.009 (0.000) | 1.309 (0.000) | 0.604 (0.028) | 0.605 (0.028) | 0.685 (0.052) | 0.693 (0.048) | 0.693 (0.048) | 0.789 (0.085) |
| FreTS | 0.974 (0.000) | 0.974 (0.000) | 1.219 (0.000) | 0.593 (0.013) | 0.594 (0.013) | 0.668 (0.027) | 0.802 (0.207) | 0.802 (0.207) | 1.059 (0.552) |
| Koopa | 0.988 (0.000) | 0.989 (0.000) | 1.307 (0.000) | 0.606 (0.036) | 0.607 (0.036) | 0.668 (0.068) | 0.872 (0.311) | 0.873 (0.311) | 1.257 (0.841) |
| DLinear | 1.013 (0.000) | 1.014 (0.000) | 1.319 (0.000) | 0.597 (0.024) | 0.598 (0.024) | 0.677 (0.045) | 0.825 (0.270) | 0.825 (0.271) | 1.115 (0.736) |
| FILM | 1.154 (0.000) | 1.155 (0.000) | 1.737 (0.000) | 0.497 (0.022) | 0.498 (0.022) | 0.543 (0.029) | 0.513 (0.071) | 0.513 (0.071) | 0.500 (0.105) |
| CSDI | 0.995 (0.000) | 0.996 (0.000) | 1.327 (0.000) | 0.604 (0.025) | 0.605 (0.025) | 0.682 (0.033) | 0.716 (0.179) | 0.717 (0.179) | 0.851 (0.372) |
| US-GAN | 1.011 (0.000) | 1.012 (0.000) | 1.560 (0.000) | 0.537 (0.027) | 0.537 (0.027) | 0.585 (0.045) | 0.859 (0.358) | 0.860 (0.358) | 1.532 (0.909) |
| GP-VAE | 1.017 (0.000) | 1.018 (0.000) | 1.350 (0.000) | 0.545 (0.018) | 0.546 (0.018) | 0.597 (0.032) | 0.791 (0.283) | 0.792 (0.283) | 1.066 (0.765) |
| Mean | 1.133 (0.000) | 1.134 (0.000) | 1.715 (0.059) | 0.468 (0.021) | 0.468 (0.021) | 0.477 (0.314) | 0.701 (0.314) | 0.701 (0.314) | 0.931 (0.833) |
| Median | 1.156 (0.023) | 1.157 (0.023) | 1.774 (0.059) | 0.507 (0.046) | 0.507 (0.046) | 0.558 (0.089) | 0.621 (0.238) | 0.622 (0.238) | 0.725 (0.626) |
| LOCF | 1.155 (0.019) | 1.156 (0.019) | 1.775 (0.048) | 0.570 (0.097) | 0.571 (0.097) | 0.692 (0.204) | 0.734 (0.325) | 0.734 (0.326) | 1.034 (0.857) |
| Linear | 1.137 (0.034) | 1.139 (0.034) | 1.722 (0.100) | 0.568 (0.086) | 0.568 (0.086) | 0.679 (0.181) | 0.679 (0.299) | 0.679 (0.299) | 0.902 (0.780) |

(MAE w/t XGB 1.192 (0.000); MRE w/t XGB 1.193 (0.000); MSE w/t XGB 1.870 (0.000))

