# OpenReview forum: "TSI-Bench: Benchmarking Time Series Imputation"
_ICLR.cc/2025/Conference — ICLR 2025 Conference Withdrawn Submission_

### Official Review · Reviewer_Evcs · 2024-10-15

**Soundness:** 3
**Presentation:** 4
**Contribution:** 2
**Rating:** 5
**Confidence:** 4

**Summary:**

This paper proposes a benchmark dataset for time series imputation. It integrates 8 existing time-series datasets, and 28 imputation models and evaluates the influence of missingness, and missing patterns on the combination of datasets and models. Investigation of different downstream tasks is also conducted.

**Strengths:**

1. This paper thoroughly summarizes existing benchmark datasets and imputation methods.
2. Experiments are thoroughly conducted to study missingness, different methods, different missing patterns and downstream tasks.

**Weaknesses:**

1. This paper is better suited for a journal as a review paper. Nothing novel is provided. All the methods and datasets are from existing works.

2. the findings provided in this paper are also not surprising, i.e. 1) Different missing patterns and rates significantly influence the performance of imputation methods 2) Forecasting architectures demonstrate effectiveness when used as an imputation backbone 3) Imputation enhances both flexibility and effectiveness across downstream tasks.

**Questions:**

None

---

### Official Review · Reviewer_mYVZ · 2024-11-02

**Soundness:** 2
**Presentation:** 2
**Contribution:** 2
**Rating:** 3
**Confidence:** 4

**Summary:**

This paper introduces TSI-Bench, a benchmarking framework for evaluating the performance of various time series imputation (TSI) methods, particularly those using deep learning. The authors highlight the lack of standardized benchmarking in existing TSI research and aim to address this gap by providing a comprehensive platform for evaluating imputation models across diverse settings. TSI-Bench incorporates 28 different algorithms, including both imputation-specific and adapted forecasting models, and evaluates them on eight datasets spanning various domains. The benchmark considers different missingness patterns and rates, and assesses the impact of imputation on downstream tasks like forecasting, classification, and regression.

**Strengths:**

-	The paper addresses a crucial need for standardized benchmarking in TSI research, enabling more rigorous and comparable evaluations of different methods.
-	TSI-Bench includes a wide range of deep learning models, datasets, and downstream tasks.

**Weaknesses:**

-	While a wide range of time-series imputation methods are considered in this study, more recent multiple imputation methods (e.g., using generative models or probabilistic models) are missing (e.g., [A] – [C]). Including these would provide a more complete picture of the current state of time series imputation.
-	The adaptation of forecasting methods for imputation seems somewhat unnatural. It appears to involve simply replacing the forecasting output layer with a masking mechanism (similar to Masked Language Modeling) to reconstruct missing values. This doesn't truly reflect the core purpose of those forecasting models and might not be the most effective way to use them for imputation.
-	To thoroughly assess the impact of imputation, the benchmark should compare the performance of downstream models (like XGBoost) both with and without imputation, and ideally across different types of downstream models, not just XGBoost. This would offer a more robust evaluation of how imputation affects various prediction tasks.
-	The benchmark doesn't fully explore the diversity of missing data scenarios. It should include different missingness assumptions, such as Missing Not At Random (MNAR) and Missing Completely At Random (MCAR), to assess how imputation methods perform under different conditions. Additionally, various missing patterns, like periodic missingness, should be incorporated. This would help to identify which imputation methods are most effective for specific types of missingness and provide a more nuanced understanding of their strengths and weaknesses.
-	The experimental design lacks the depth needed to truly understand the strengths and weaknesses of different imputation methods.  Simply providing performance metrics across various datasets isn't enough. The benchmark needs more targeted experiments to reveal why certain methods perform better in specific situations. Without this level of analysis, the value of the benchmark is limited. It becomes a simple performance comparison rather than a tool for understanding the nuances of when and how different imputation methods should be applied.



References:
- [A] M. Choi and C. Lee, “Conditional Information Bottleneck Approach for Time Series Imputation,” ICLR 2024.
- [B] S. Kim et al, “Probabilistic Imputation for Time-series Classification with Missing Data,” ICML 2023.
- [C] J. Alcaraz and N. Strodthoff, “Diffusion-based Time Series Imputation and Forecasting with Structured State Space Models, TMLR 2024.

**Questions:**

-	Please respond to the weakness 4 and 5.
-	Some datasets used in the benchmark already contain missing values. Could you explain how these pre-existing missing values were handled during training and testing?

---

### Official Review · Reviewer_uzuS · 2024-11-04

**Soundness:** 3
**Presentation:** 2
**Contribution:** 2
**Rating:** 3
**Confidence:** 4

**Summary:**

This paper introduces a benchmark for time series imputation. This benchmark includes 8 datasets comprising air quality, traffic, electricity, and healthcare domains. Benchmark also includes 28 methods, from traditional methods to predictive and generative deep learning models. The paper compare model performances on these datasets, on diverse missing rates, patterns, and downstream performances.

**Strengths:**

- This paper provides a unified and systematic framework on evaluating time series imputation models.
- Full results tables in appendix make experiments more credible.

**Weaknesses:**

1. Models, especially generative models (VAE, GAN, Diffusion, or others) are outdated. Authors should include more recent methods such as  mTANs[1], TimeCIB[2], GRIN [3], DSPD-GP[4], TIDER[5], BiTGraph[6], and many others. (This is not a research paper but benchmark, I believe the paper should provide much more comprehensive comparison)
2. Especially for deep learning methods, authors don't provide hyperparameters and method-specific settings. For example, which GP kernels did you use? For diffusion models, how comprehensive did you tested on noise schedule / number of timesteps ? Are these models indeed have optimal hyperparameters / training settings for new datasets? If you exclude several tests, what was the reason?

[1] Satya Narayan Shukla and Benjamin Marlin. Multi-Time Attention Networks for Irregularly Sampled Time Series. In International Conference on Learning Representations, 2021.

[2] MinGyu Choi and Changhee Lee. Conditional Information Bottleneck Approach for Time series Imputation. In International Conference on Learning Representations, 2023.

[3] Cini, Andrea, Ivan Marisca, and Cesare Alippi. Filling the g_ap_s: Multivariate time series imputation by graph neural networks. In International Conference on Learning Representations, 2022.

[4] Bilos et al., Modeling Temporal Data as Continuous Functions with Stochastic Process Diffusion, ICML, 2023

[5] Shuai et al., Multivariate Time-series Imputation with Disentangled Temporal Representations, In International Conference on Learning Representations, 2023.

[6] Chen, Xiaodan, et al. "Biased temporal convolution graph network for time series forecasting with missing values." The Twelfth International Conference on Learning Representations. 2023.

**Questions:**

1. Regarding Weakness 1 - please provide much more comprehensive comparison among recent methods. Please focus on recent probabilistic generative models. If you had to choose only subset of them, please provide the reason. Again, since this paper is proposing a benchmark, I think it should include more comprehensive list of methods.
2. Regarding Weakness 2 - please provide settings for each method, each dataset, each task. You don't need to do full hyperparameter tunings, but please provide hyperparameter settings. For (recent) generative methods, I believe their performances highly depends on hyperparameter settings. I wish to know whether you tried to find (sub)optimal settings when you test on new dataset.
3. Related to 2: According to your tables, BRITS seems one of the most powerful tool yet it is quite early-published one. Do you have any hypothesis on why modern methods cannot beat them? Isn't this just because modern methods are not fully optimized for new tasks?

---

### Official Review · Reviewer_cU7Z · 2024-11-05

**Soundness:** 3
**Presentation:** 2
**Contribution:** 2
**Rating:** 3
**Confidence:** 4

**Summary:**

This article introduces a new benchmarking platform for the task of time series imputation. This platform includes, in particular, time series preprocessing, a process for masking the series according to three patterns, a substantial number of state-of-the-art algorithms, and evaluation metrics. The second part of the article presents the benchmarking results of 28 algorithms on 8 datasets from various perspectives: type and amount of imputed data, type of deep architecture, performance analysis on downstream tasks. Analyses and conclusions are drawn from all these experiments.

**Strengths:**

* The article is well-written, well-presented, and provides a comprehensive overview of the state of the art.
* The idea of a benchmarking platform for time series imputation is excellent (and indeed desirable for other tasks as well like forecasting); it would be a substantial help to the community and a fair way to evaluate both new and existing algorithms.
* In the description provides by the article, the platform seems to contain the necessary features for such a benchmark.
* The presentation of the results and the analysis conducted are insightful and demonstrate the value of such a platform.

**Weaknesses:**

The weaknesses mainly concern the platform's capabilities and its ease of use :
* It is really unfortunate that there was no plan to include static information (such as spatial coordinates for sensor networks, or other types of descriptive attributes). Although many state-of-the-art algorithms for imputation are generic and do not use static information, in many use cases such information is available, and it would be beneficial to allow its use.
*Looking at the code, the platform does not appear to be user-friendly. The provided code seems to manage all the experiments listed by the authors, but the code's organization makes it difficult to use outside of the intended framework:
 - There is no documentation (though the code is commented).
 -  The missing rate settings are pre-set, and one must manually edit the code to adjust these settings.
 - The list of algorithms is hard-coded, requiring modification of several files to add a new algorithm.
 - There is no clear specification of the expected input/output for the algorithms.
 * It is very challenging to imagine how to integrate a new dataset into the platform.

In short, while the idea and approach are excellent, the current result is not satisfactory for broad dissemination and use of the platform, except to replicate the experiments already conducted by the authors. Since I consider this a main contribution of the paper, this explains the rating I have given.

* The submission is not fully anonymized; an author's name appears in the code provided in the supplementary material (e.g. global_config.py)

**Questions:**

I think the authors should take the time to refine the platform for a more widely usable version, incorporating additional features.

---

### Note · Authors · 2024-11-19

I have read and agree with the venue's withdrawal policy on behalf of myself and my co-authors.